# HIGHER-ORDER GRAPHON NEURAL NETWORKS: APPROXIMATION AND CUT DISTANCE

**Daniel Herbst**[1] **and Stefanie Jegelka**[1,2,3]
[1]TUM, School of CIT   [2]TUM, MCML and MDSI   [3]MIT, Department of EECS and CSAIL
{daniel.herbst, stefanie.jegelka}@tum.de

## ABSTRACT

Graph limit models, like *graphons* for limits of dense graphs, have recently been used to study size transferability of graph neural networks (GNNs). While most literature focuses on message passing GNNs (MPNNs), in this work we attend to the more powerful *higher-order* GNNs. First, we extend the $k$-WL test for graphons (Böker, 2023) to the graphon-signal space and introduce *signal-weighted homomorphism densities* as a key tool. As an exemplary focus, we generalize *Invariant Graph Networks* (IGNs) to graphons, proposing *Invariant Graphon Networks* (IWNs) defined via a subset of the IGN basis corresponding to bounded linear operators. Even with this restricted basis, we show that IWNs of order $k$ are at least as powerful as the $k$-WL test, and we establish universal approximation results for graphon-signals in $L^p$ distances. This significantly extends the prior work of Cai & Wang (2022), showing that IWNs—a subset of their *IGN-small*—retain effectively the same expressivity as the full IGN basis in the limit. In contrast to their approach, our blueprint of IWNs also aligns better with the geometry of graphon space, for example facilitating comparability to MPNNs. We highlight that, while typical higher-order GNNs are discontinuous w.r.t. cut distance—which causes their lack of convergence and is inherently tied to the definition of $k$-WL—transferability remains achievable.

## 1 INTRODUCTION

Graph Neural Networks (GNNs) have emerged as a powerful tool for machine learning on complex graph-structured data, driving advances in fields like social network analysis (Fan et al., 2019), weather prediction (Lam et al., 2023) or materials discovery (Merchant et al., 2023). Message Passing GNNs (MPNNs) (Gilmer et al., 2017; Kipf & Welling, 2017; Veličković et al., 2018; Xu et al., 2019), which update node features by neighborhood aggregations, are a popular paradigm.

The question of *size transferability*—whether an MPNN generalizes to larger graphs than those in the training set—has recently gained attention. Unlike *extrapolation* (Xu et al., 2021; Yehudai et al., 2021; Jegelka, 2022), where generalization to arbitrary graph topologies is considered, size transferability typically assumes structural similarities between the training and evaluation graphs, such as them being sampled from the same random graph model (Keriven et al., 2020), topological space (Levie et al., 2021), or graph limit model (Ruiz et al., 2020; 2023; 2021b; Maskey et al., 2024; Le & Jegelka, 2023). For limits of dense graphs, *graphons* (Lovász & Szegedy, 2006; Lovász, 2012), which extend graphs to node sets on the unit interval and have been used to study extremal graph theory with analytic techniques, have also become a popular choice for studying transferability. In contrast to sparse graph limits (Lovász, 2012; Backhausz & Szegedy, 2022), they offer an established framework with powerful tools, such as embedding the set of all graphs into a compact space and having favorable spectral properties (Ruiz et al., 2021a). In such transferability analyses, a GNN is extended to a function of graphons, and regularity properties of the GNN are then used to bound the difference between outputs of the GNN applied to samples of different sizes from a graphon.

Existing works on transferability have been almost exclusively limited to MPNNs. However, their expressive power is constrained by the 1-*dimensional Weisfeiler-Leman graph isomorphism test* (1-WL), also known as the *color refinement algorithm* (Xu et al., 2019; Morris et al., 2019). Hence, standard MPNNs even fail at straightforward tasks such as counting simple patterns like cycles.

This motivates extending generalization analyses to more powerful architectures. Most prominent among these are higher-order extensions of MPNNs that are as powerful as the $k$-WL test, $k > 1$, which iteratively colors $k$-tuples of nodes (Morris et al., 2019). *Invariant* and *Equivariant Graph Networks* (IGNs/EGNs) (Maron et al., 2019b), which serve as an exemplary focus in this work, are a popular architectural choice, in which adjacency matrices and node signals are processed through higher-order tensor operations that maintain permutation equivariance. IGNs/EGNs universally approximate any permutation in-/equivariant graph function, and are as powerful as $k$-WL with orders $\leq k$ (Maron et al., 2019c; Keriven & Peyré, 2019; Maehara & NT, 2019; Azizian & Lelarge, 2021).

The expressive power of a GNN can also be judged via its *homomorphism expressivity*, i.e., its ability to count the number of homomorphisms from fixed graphs into the input graph. E.g., 1-WL corresponds to counting homomorphisms w.r.t. trees, and its higher-order extensions are related to counting homomorphisms w.r.t. graphs of bounded treewidth (Dvořák, 2010; Dell et al., 2018). In the graphon case, similar results exist for *homomorphism densities* (Böker et al., 2023; Böker, 2023).

In this work, we study expressivity, continuity, and transferability properties of higher-order GNN extensions to graphons. Maehara & NT (2019) note that their IGN/EGN universality proof—using a parametrization relying on explicit *simple* homomorphism densities—extends to graphons. They regard the use of graphons for such analyses as promising. Keriven et al. (2021) study the convergence of a universal class of GNNs based on node IDs and identify the analysis of higher-order GNNs as an open direction. The closest related work by Cai & Wang (2022) investigates the convergence of IGNs to a limit graphon via a *partition norm*—a vector of norms over all diagonals of a graphon. They observe that IGNs on graphon-sampled graphs do not always converge. They propose a reduced model class *IGN-small*, which enables convergence after estimating edge probabilities under certain regularity conditions. They also show that IGN-small retains sufficient expressiveness to approximate spectral GNNs. We argue that considering diagonals of graphons (which would lead to a limit object of self-looped graphs) is somewhat nonstandard in the larger body of work in graphon theory, limiting its applicability to their version of IGN limits. Furthermore, the expressivity analysis of IGN-small is rather limited, given that IGNs are typically universal GNN architectures.

**Contributions.** A first focus of this work is to extend the $k$-WL test (Böker, 2023) and homomorphism densities from graphons to graphon-signals (Levie, 2023), i.e., node-attributed graphons. For the extension of homomorphism densities, we introduce signal weighting and show that *signal-weighted homomorphism densities* inherit most topological properties from their graphon equivalent.

We generalize IGNs to graphon-signals, introducing *Invariant Graphon Networks (IWNs)*. In contrast to Cai & Wang (2022), we restrict linear equivariant layers to *bounded operators*, and, thus, our IWNs can be analyzed using $L^p$ and cut distances, enhancing comparability to the existing graphon literature. Using only this reduced basis, we show that IWNs of order up to $k$ are as powerful as the $k$-WL test for graphon-signals and we establish universal approximation results in $L^p$ distances. As IWNs are a subset of IGN-small, this significantly extends the work of Cai & Wang (2022), resolving the open questions posed in their conclusion: We show that the restriction to IGN-small comes at no cost in terms of expressivity. We carry out expressivity analyses using our notion of homomorphism expressivity via signal-weighted homomorphism densities. IWNs are discontinuous w.r.t. the cut distance and only continuous in the finer topologies induced by $L^p$ distances, which do not represent our intuitive notion of graph similarity (Levie, 2023). This discontinuity is not unique to IWNs, but inherently tied to the way in which $k$-WL processes edge weights, and results in a large class of higher-order GNNs defined via color refinement exhibiting an absence of convergence under sampling simple graphs. Yet, despite this discontinuity, transferability is still possible.

To the best of our knowledge, this work is the first to extend higher-order GNNs to graphons in a way that facilitates a systematic study of *continuity*, *expressivity*, and *transferability* in comparison to MPNNs, addressing the aforementioned challenges of Cai & Wang (2022) while building on the foundational work of Böker (2023). In summary, we make the following contributions:

- We define *signal-weighted homomorphism densities*, link them to a natural extension of the $k$-WL test to graphon-signals, and show how they capture graphon-signal topology.
- We introduce *Invariant Graphon Networks (IWNs)*, restricting linear equivariant layers to bounded operators. We show that IWNs of order $k$ are at least as powerful as $k$-WL for graphon-signals, and establish universal approximation, extending Cai & Wang (2022).
- We point out the cut distance discontinuity of typical higher-order GNNs and demonstrate that such models are still transferable despite not converging to their graphon limit.

## 2 BACKGROUND

In this section, we provide background on graphon theory, homomorphism expressivity, and the $k$-WL test for graphons, as well as on how to extend graphons to incorporate node signals. Contents of § 2.1 and § 2.2 are mostly drawn from Lovász (2012); Janson (2013); Zhao (2023), while in § 2.2 we also refer to Böker (2023). In § 2.3 we summarize key results of Levie (2023).

For $n \in \mathbb{N}$, write $[n] := \{1, \dots, n\}$. Unless stated otherwise, a graph always refers to a *simple graph*, meaning an undirected graph $G = (V, E)$ with a finite node set $V(G) = V$ and edge set $E(G) = E \subseteq \binom{V}{2}$. Define also $v(G) := |V(G)|$, $e(G) := |E(G)|$. We will also consider *multigraphs*, for which the edges are a *multiset*. Write $\lambda^k$ for the $k$-dimensional Lebesgue measure; $\lambda := \lambda^1$. See § A for a table of notation and § B.1 for background on topology and measure theory.

### 2.1 GENERAL BACKGROUND ON GRAPHON THEORY

**Graphons.** Informally, a graphon can be seen as a graph with node set $[0, 1]$, and the adjacency matrix being represented by a function on the unit square. Intuitively, graphons can be obtained by taking the limit of adjacency matrices of *dense* graph sequences as the number of nodes grows. Formally, we first define a **kernel** as a bounded symmetric measurable function $W : [0, 1]^2 \to \mathbb{R}$. A **graphon** is a kernel mapping to $[0, 1]$. Write $\mathcal{W}$ for the space of kernels; $\mathcal{W}_0$ for graphons. Define the **cut norm** of a kernel as

$$\|W\|_\square := \sup_{S, T \subseteq [0,1]} \left| \int_{S \times T} W \, \mathrm{d}\lambda^2 \right|, \tag{1}$$

where $S, T$ are tacitly assumed measurable. Let $S_{[0,1]}$ be the set of measure preserving maps $\varphi : [0, 1] \to [0, 1]$ that are bijective almost everywhere; that is, for every measurable $A \subseteq [0, 1]$, we have $\lambda(\varphi^{-1}(A)) = \lambda(A)$. Write $\overline{S}_{[0,1]}$ for the set of *all* measure preserving functions and define $W^\varphi(x, y) := W(\varphi(x), \varphi(y))$ for $\varphi \in \overline{S}_{[0,1]}$. Since the specific ordering of the graphon values does not matter, we work with the **cut distance** (Lovász, 2012, § 8.2) between two graphons, defined as

$$\delta_\square(W, V) := \inf_{\varphi \in S_{[0,1]}} \|W - V^\varphi\|_\square = \min_{\varphi, \psi \in \overline{S}_{[0,1]}} \|W^\varphi - V^\psi\|_\square, \tag{2}$$

where the infimum is only guaranteed to be attained in the latter expression. Analogously, we can define distances $\delta_p$ on graphons based on $L^p$ norms, $p \in [1, \infty]$. Note that $\delta_\square \leq \delta_1$ (which follows from moving $|\cdot|$ into the integral in (1)) and $\delta_p \leq \delta_q$ for $p \leq q$. $\delta_\square$ and $\{\delta_p\}_p$, $p < \infty$, vanish simultaneously. However, the topology induced by $\delta_\square$ is strictly coarser than that induced by $\{\delta_p\}_{p \in [1,\infty)}$ (all of which coincide), which is in turn coarser than the topology induced by $\delta_\infty$. The most commonly used among $\{\delta_p\}_p$ is $\delta_1$, which corresponds (up to an absolute constant; see Lovász (2012, § 9)) to the edit distance on graphs. We identify **weakly isomorphic** graphons of distance 0 to form the space $\widetilde{\mathcal{W}}_0$ of *unlabeled graphons*. The stricter concept of (strong) isomorphism, namely that the minimum over $S_{[0,1]}$ in (2) is attained and zero, is less practical. The usefulness of $\delta_\square$ over any $\delta_p$ lies in the fact that $(\widetilde{\mathcal{W}}_0, \delta_\square)$ forms a *compact space* (Lovász, 2012, Theorem 9.23).

**Discretization and sampling.** Any labeled graph $G$ with adjacency $\boldsymbol{A} \in \mathbb{R}^{n \times n}$ can be identified with its *induced step graphon* $W_G := \sum_{j=1}^n \sum_{k=1}^n A_{jk} \mathbb{1}_{I_j \times I_k}$ for a regular partition $\{I_j\}_{j=1}^n$ of the unit interval, and finite graphs are dense in the graphon space (Zhao, 2023, Theorem 4.2.8). Graphons can also be seen as random graph models: Draw $\boldsymbol{X} \sim U(0, 1)^n$, and let $W(\boldsymbol{X})$ be a graph with edge weights $W_{ij} = W(X_i, X_j)$. If $e_{ij} \sim \mathrm{Bernoulli}(W_{ij})$ is further sampled, we obtain an unweighted graph $\mathbb{G}(W, \boldsymbol{X})$. Write $\mathbb{H}_n(W)$ and $\mathbb{G}_n(W)$ for the respective distributions. We have $\delta_\square(\mathbb{H}_n(W), W) \leq \delta_1(\mathbb{H}_n(W), W) \to 0$ (Zhao, 2023, Lemma 4.9.4). Also $\delta_\square(\mathbb{G}_n(W), W) \to 0$ (Lovász, 2012, Proposition 11.32), but this does not hold for $\delta_1$: Take, e.g., $W \equiv 1/2$, then $\delta_1(W, \mathbb{G}_n(W)) = 1/2$ for all $n \in \mathbb{N}$.

**Homomorphism densities.** Let $\mathrm{hom}(F, G)$ denote the number of homomorphisms from a graph $F$ into a graph $G$. The corresponding **homomorphism density** is defined as $t(F, G) := \mathrm{hom}(F, G)/v(G)^{v(F)}$, i.e., the proportion of homomorphisms among all maps $V(F) \to V(G)$. We define the homomorphism density of a (multi)graph $F$ with $V(F) = [k]$ to a graphon $W$ by

$$t(F, W) := \int_{[0,1]^k} \prod_{\{i,j\} \in E(F)} W(x_i, x_j) \, \mathrm{d}\lambda^k(\boldsymbol{x}), \tag{3}$$

where factors may appear multiple times for a multigraph. This generalizes the discrete concept in the sense that $t(F, G) = t(F, W_G)$. For a sequence $(W_n)_n$ of graphons, $\delta_\square(W_n, W) \to 0$ iff $t(F, W_n) \to t(F, W)$ for all *simple* graphs $F$, and thus two graphons $W, V$ are weakly isomorphic iff $t(F, W) = t(F, V)$ for all simple graphs $F$. Hence, homomorphism densities can be seen as a counterpart of moments of a real random variable for $W$-random graphs, as they fix the distribution $\mathbb{G}_n(W)$ similarly as the moments would for a well-behaved real random variable (Zhao, 2023).

## 2.2 $k$-WL and Homomorphism Expressivity

In the discrete setting, the *1-dimensional Weisfeiler-Leman (1-WL) graph isomorphism test* (or *color refinement algorithm*) and its multidimensional extensions are widely used to judge the expressive power of a GNN model. Alternatively, the model's **homomorphism expressivity**, i.e., its ability to count the number of homomorphisms from smaller graphs, called *patterns*, into the input graph, can be considered. Through homomorphic images (Lovász, 2012, § 6.1), this also relates to subgraph counting (Chen et al., 2020; Tahmasebi et al., 2023; Jin et al., 2024). 1-WL expressivity corresponds precisely to distinguishing graphs for which the values of $\hom(T, \cdot)$ differ if $T$ are trees. More generally, $k$-WL can be precisely characterized as being able to compute $\{\hom(F, \cdot)\}_F$, with $F$ ranging over all simple graphs of treewidth bounded by $k$ (Dvořák, 2010; Dell et al., 2018). See also § B.4 for more information on treewidth and the tree decomposition of a graph. A finer characterization for various GNN architectural choices was recently shown by Zhang et al. (2024).

The 1-WL test has been extended to graphons by Grebík & Rocha (2022) through the concept of **iterated degree measures (IDMs)**. These serve as the continuous counterpart of the color space used in the color refinement algorithm for graphons and are represented by sequences $(\alpha_n)_n$ of colors after $n$ refinement rounds. The *distribution* of such colors, akin to the multiset of all assigned colors per node, represents the result of the isomorphism test. As in the discrete case, two graphons are 1-WL indistinguishable iff their tree homomorphism densities match.

Recently, Böker (2023) developed a $k$-WL test for graphons using distributions of $k$-WL *measures*. Intuitively, for a given graphon $W$, the mapping $\mathfrak{C}_W^{k\text{-WL}}$ assigns a color to every $k$-tuple of nodes in $[0,1]^k$; these colors are elements of a topological space $\mathbb{M}^k$ called $k$-**WL measures**. The resulting $k$-**WL distribution**, defined as the pushforward $\nu_W^{k\text{-WL}} := (\mathfrak{C}_W^{k\text{-WL}})_* \lambda^k \in \mathcal{P}(\mathbb{M}^k)$, is a Borel probability measure on $\mathbb{M}^k$ that captures the test's output. Notably, the homomorphism characterization of this natural $k$-WL test is given in terms of *multigraph* homomorphism densities w.r.t. patterns of bounded treewidth (rather than *simple* graphs as for 1-WL). See also § D. Note that in this work, $k$-WL always refers to the *oblivious* $k$-WL instead of the *Folklore* $k$-WL test, which also processes $k$-tuples but is $(k+1)$-WL expressive (Cai et al., 1992; Grohe & Otto, 2015; Jegelka, 2022).

## 2.3 Extension to Graphon-Signals

Most common GNNs take a graph-signal $(G, \boldsymbol{f})$ as inputs, i.e., a graph $G$ with node set $[n] := \{1, \dots, n\}$ and a signal $\boldsymbol{f} \in \mathbb{R}^{n \times k}$, with $k$ being the number of features. Levie (2023) extends this definition to graphons with one-dimensional node signals. They fix $r > 0$, consider signals in $L_r^\infty[0,1] := \{f \in L^\infty[0,1] \mid \|f\|_\infty \leq r\}$, and set $\|f\|_\square := \sup_{S \subseteq [0,1]} |\int_S f \, d\lambda|$ with $S$ measurable. Note that this is equivalent to the signal $L^1$ norm. They then let $\mathcal{WS}_r := \mathcal{W}_0 \times L_r^\infty[0,1]$ and define the *cut norm* $\|(W, f)\|_\square := \|W\|_\square + \|f\|_\square$. Define $\delta_\square$ and $\delta_p$, step graphon-signals, and sampling from graphon-signals analogously to the standard case. E.g., write $\mathbb{G}_n(W, f)$ for the distribution of $(\mathbb{G}(W, \boldsymbol{X}), f(\boldsymbol{X}))$, $\boldsymbol{X} \sim U(0,1)^n$. Also, identify weakly isomorphic graphon-signals of cut distance zero to obtain the space $\widetilde{\mathcal{WS}}_r$ of *unlabeled* graphon-signals. Central to their contribution, Levie (2023) establishes the compactness of $(\widetilde{\mathcal{WS}}_r, \delta_\square)$ and bounds its covering number (cf. § B.5). They also derive a sampling lemma: For $r > 1$, $(W, f) \in \mathcal{WS}_r$, and large $n \in \mathbb{N}$,

$$\mathbb{E}\left[\delta_\square\big((W, f), \mathbb{G}_n(W, f)\big)\right] \leq 15 \, (\log n)^{-1/2}. \tag{4}$$

## 3 Signal-Weighted Homomorphism Densities

It is important to note that Böker et al. (2023); Böker (2023) focus exclusively on graphons and do not consider graphon-signals, and Levie (2023) does not introduce a notion of homomorphism den-

sities for graphon-signals either. However, since most GNN architectures in the literature operate on node-featured graphs, we need a concept of homomorphism densities that reflects the properties of the graphon-signal space well. This could then, e.g., be applied to characterize the homomorphism expressivity of GNN models on graphon-signals, similar to the approach for graphons outlined in § 2.2. As in Lovász (2012, § 5.2) for finite graphs, we introduce weighting by signals.

**Definition 3.1** (Signal-weighted homomorphism density). *Let $F$ be a multigraph with $V(F) = [k]$, $\boldsymbol{d} \in \mathbb{N}_0^k$, and let $(W, f) \in \mathcal{WS}_r$. We set*

$$t((F, \boldsymbol{d}), (W, f)) := \int_{[0,1]^k} \Big( \prod_{i \in V(F)} f(x_i)^{d_i} \Big) \Big( \prod_{\{i,j\} \in E(F)} W(x_i, x_j) \Big) \, \mathrm{d}\lambda^k(\boldsymbol{x}), \qquad (5)$$

*calling the functions $t((F, \boldsymbol{d}), \cdot)$ **signal-weighted homomorphism densities**.*

Note that setting $\boldsymbol{d} = \boldsymbol{0} \in \mathbb{N}_0^k$ recovers the graphon homomorphism densities $t((F, \boldsymbol{0}), (W, f)) = t(F, W)$. $\boldsymbol{d} \neq \boldsymbol{0}$ will allow us to consider moments of the signal, which could alternatively be viewed as a multiset of *nodes*, similarly to homomorphism densities of multigraphs. This enables us to capture the distribution of the signal, coupled with the graph structure, which will be crucial for (5) to separate non-weakly isomorphic graphon-signals. In contrast to common approaches in the GNN literature, only considering $\boldsymbol{d} = \boldsymbol{1}$ does not suffice in our case, as this only distinguishes graphs under twin reduction. Restricting the exponents to be the same across all nodes as in Nguyen & Maehara (2020) results in $\{t((F, \boldsymbol{d}), \cdot)\}_{F, \boldsymbol{d}}$ not being closed under multiplication, which would later pose challenges. The finite-graph approach of enforcing homomorphisms to respect signal values does not extend to graphon-signals—the level sets $\{f = t\}$ may all have measure zero, making homomorphism densities degenerate. A possible workaround is incorporating "similarity kernels" in (5) for approximate matches, but this also appears to be less practical.

The definition of signal weighting assumes a scalar-valued signal $f$, with integer node features $\boldsymbol{d}$ acting via $x \mapsto x^{d_i}$. This could be straightforwardly extended to signals mapping into a compact space $K$, where we define signal-weighted homomorphism densities by replacing pattern node features with continuous functions $\boldsymbol{\omega} \in C(K)^{v(F)}$ applied pointwise to $f(x_i)$. To uniquely determine a graphon-signal, it suffices to use a dense subset of the continuous functions $C(K)$, such as polynomials for $K = [-r, r]$, which form algebras and enable the use of the Stone-Weierstrass theorem.

As a first step, we derive a counting lemma (cf. § C.1) similar to the standard graphon case (Lovász, 2012, Lemma 10.23), which shows that signal-weighted homomorphism densities from *simple* graphs into a graphon-signal are Lipschitz continuous w.r.t. cut distance. A similar statement can also be shown for all *multigraphs* using $\delta_1$. However, the main justification for Definition 3.1 is Theorem 3.2 (akin to Theorem 8.10 from Janson (2013)), as well as Corollary 3.3, demonstrating how signal-weighted homomorphism densities capture weak isomorphism and the topological structure of the graphon-signal space in a similar way as do homomorphism densities for graphons:

**Theorem 3.2** (Characterizations of weak isomorphism for graphon-signals). *Fix $r > 1$ and let $(W, f), (V, g) \in \mathcal{WS}_r$. Then, the following statements are equivalent:*

*(1) $\delta_p((W, f), (V, g)) = 0$ for any $p \in [1, \infty)$;*
*(2) $\delta_\square((W, f), (V, g)) = 0$;*
*(3) $t((F, \boldsymbol{d}), (W, f)) = t((F, \boldsymbol{d}), (V, g))$ for all multigraphs $F$, $\boldsymbol{d} \in \mathbb{N}_0^{v(F)}$;*
*(4) $t((F, \boldsymbol{d}), (W, f)) = t((F, \boldsymbol{d}), (V, g))$ for all simple graphs $F$, $\boldsymbol{d} \in \mathbb{N}_0^{v(F)}$;*
*(5) $\mathbb{H}_k(W, f) \overset{\mathcal{D}}{=} \mathbb{H}_k(V, g)$ for all $k \in \mathbb{N}$;*
*(6) $\mathbb{G}_k(W, f) \overset{\mathcal{D}}{=} \mathbb{G}_k(V, g)$ for all $k \in \mathbb{N}$.*

See § C.3 for the proof. The equivalence **(1)** $\Leftrightarrow$ **(2)** in Theorem 3.2, which we show in § C.2 by extending the argument of Lovász (2012, Theorem 8.13), reveals that any $\{\delta_p\}_{p \in [1, \infty)}$ distance could be alternatively used to define weak isomorphism of two graphon-signals. Thus, any $\delta_p$ can also be seen as a metric on the space of unlabeled graphon-signals. The other equivalences show that weak isomorphism of two graphon-signals can be alternatively characterized by them having the same signal-weighted homomorphism densities, and the same random graph distributions. Specifically, $\{t((F, \boldsymbol{d}), \cdot)\}_{F, \boldsymbol{d}}$ fixes the distribution of $(W, f)$-random graphs similarly as do homomorphism densities for $W$-random graphs or moments for real-valued random variables. We also remark that the condition $r > 1$ stems from the fact that we use the graphon-signal sampling lemma (Levie, 2023,

Theorem 3.7). The following corollary shows that signal-weighted homomorphism densities of *simple* graphs precisely characterize cut distance convergence (refer to § C.4 for the proof):

**Corollary 3.3** (Convergence in graphon-signal space)**.** *For $(W_n, f_n)_n$, $(W, f) \in \mathcal{WS}_r$ and $r > 1$,*

$$\delta_\square((W_n, f_n), (W, f)) \to 0 \quad \Leftrightarrow \quad t((F, \boldsymbol{d}), (W_n, f_n)) \to t((F, \boldsymbol{d}), (W, f)) \ \ \forall F, \boldsymbol{d} \in \mathbb{N}_0^{v(F)} \quad (6)$$

*as $n \to \infty$, with $F$ ranging over all simple graphs.*

Finally, we show that signal-weighted homomorphism densities also make sense on the granularity level of the $k$-WL hierarchy, in the way that their indistinguishability is equivalent to the equality of a natural generalization of $k$-WL distributions as defined in Böker (2023) to graphon-signals.

**Theorem 3.4** ($k$-WL for graphon-signals, informal)**.** *Two graphon-signals $(W, f)$ and $(V, g)$ are $k$-WL indistinguishable if and only if $t((F, \boldsymbol{d}), (W, f)) = t((F, \boldsymbol{d}), (V, g))$ for all multigraphs $F$ of treewidth $\leq k - 1$, $\boldsymbol{d} \in \mathbb{N}_0^{v(F)}$.*

Due to their technical nature, all details, including the formal definition of the $k$-WL color space for graphon-signals as well as a formal statement of Theorem 3.4, are deferred to § D.

## 4    INVARIANT GRAPHON NETWORKS

In this section, we introduce *Invariant Graphon Networks (IWNs)* as an exemplary higher-order architecture on the graphon-signal space. The key components of IWNs are the *linear equivariant layers*, which we extend from the original framework of Maron et al. (2019b) to arbitrary measure spaces. We also determine the dimension of these layers and establish a canonical basis. We then proceed to define multilayer IWNs, highlight connections to the work of Cai & Wang (2022), and analyze the expressivity of IWNs.

### 4.1    LINEAR EQUIVARIANT LAYERS

We start with generalizing the building blocks of IGNs—namely, the linear equivariant layers. For IGNs, these are linear functions $T : \mathbb{R}^{n^k} \to \mathbb{R}^{n^\ell}$, such that $T$ is equivariant w.r.t. all *permutations* acting on the $n$ coordinates. We now extend this notion from the set $[n]$ to arbitrary measure spaces. The suitable generalization of permutations will be measure preserving maps:

**Definition 4.1** (Linear equivariant layer)**.** *Let $(\mathcal{X}, \mathcal{A}, \mu)$ be a measure space, simply denoted by $\mathcal{X}$, and let $\overline{S}_\mathcal{X}$ be the set of measure-preserving functions $\varphi : \mathcal{X} \to \mathcal{X}$. Let $k, \ell \in \mathbb{N}_0$. Write $\mathcal{X}^k$ for $(\mathcal{X}^k, \mathcal{A}^{\otimes k}, \mu^{\otimes k})$ and note that $L^2(\mathcal{X})^{\otimes k} \cong L^2(\mathcal{X}^k)$. Define the **linear equivariant layers**

$$\mathsf{LE}_{k \to \ell}^\mathcal{X} := \left\{ T \in \mathcal{L}(L^2(\mathcal{X}^k), L^2(\mathcal{X}^\ell)) \mid \forall \varphi \in \overline{S}_\mathcal{X} : T(U^\varphi) = T(U)^\varphi \ a.e. \right\} \quad (7)$$

*as the space of all bounded linear operators that are equivariant w.r.t. all measure preserving functions on $\mathcal{X}$, i.e., all relabelings of $\mathcal{X}$. Here, $U^\varphi(x_1, \ldots, x_k) := U(\varphi(x_1), \ldots, \varphi(x_k))$, and $\mathcal{L}(\cdot, \cdot)$ denotes bounded linear operators.*

For $\mathcal{X} := [n]$ with a uniform probability measure (or counting measure), we obtain $L^2([n]) \cong \mathbb{R}^n$, and $\mathsf{LE}_{k \to \ell}^{[n]}$ can be identified with the space of linear permutation equivariant functions $\mathbb{R}^{n^k} \to \mathbb{R}^{n^\ell}$, as measure preserving functions $[n] \to [n]$ are just the permutations $S_n$. This yields precisely the linear equivariant layers that are building blocks of IGNs, which were studied by Maron et al. (2019b). One of their results is that $\dim \mathsf{LE}_{k \to \ell}^{[n]} = \mathrm{bell}(k + \ell)$, with $\mathrm{bell}(m)$ denoting the number of partitions $\Gamma_m$ of $[m]$, independent of $n$. There exists a canonical basis in which every basis element $T_\gamma^{(n)} \in \mathsf{LE}_{k \to \ell}^{[n]}$ corresponds to a partition $\gamma \in \Gamma_{k+\ell}$, with basis elements being simple operations such as extracting diagonals, summing/averaging over axes, and replication (see § B.2).

For graphons, we are interested in $\mathsf{LE}_{k \to \ell} := \mathsf{LE}_{k \to \ell}^{[0,1]}$ as building blocks of IWNs, where $[0, 1]$ is equipped with its Borel $\sigma$-algebra and Lebesgue measure. The immediate question is how this space compares to $\mathsf{LE}_{k \to \ell}^{[n]}$, i.e., what its dimension is and if there exists a canonical basis we can use to parameterize IWNs later on. It turns out that this space can be seen as just implementing a subset of the possibilities in the discrete setting, which is essentially a consequence of $[0, 1]$ being atomless.

**Theorem 4.2.** *Let $k, \ell \in \mathbb{N}_0$. Then, $\mathsf{LE}_{k \to \ell}$ is a finite-dimensional vector space of dimension*

$$\dim \mathsf{LE}_{k \to \ell} = \sum_{s=0}^{\min\{k,\ell\}} s! \binom{k}{s}\binom{\ell}{s} \leq \mathrm{bell}(k + \ell). \tag{8}$$

The proof can be found in § E.1. Central to the argument is the observation that we can consider the action of any $T \in \mathsf{LE}_{k \to \ell}$ on step functions, and apply the characterization from Maron et al. (2019b) to a sequence of nested subspaces, which fixes the operator on the entire space $L^2[0,1]^k$. In fact, (8) is precisely the dimension of the *Rook* subalgebra of the partition algebra (see, e.g., Grood (2006); Halverson & Jacobson (2020)).

**A canonical basis of $\mathsf{LE}_{k \to \ell}$.** The proof of Theorem 4.2 also provides insight into constructing a canonical basis of $\mathsf{LE}_{k \to \ell}$, which is indexed by the following subset of the partitions $\Gamma_{k+\ell}$ of $[k+\ell]$:

$$\widetilde{\Gamma}_{k,\ell} := \left\{ \gamma \in \Gamma_{k+\ell} \mid \forall A \in \gamma : |A \cap [k]| \leq 1, \ |A \cap (k + [\ell])| \leq 1 \right\}. \tag{9}$$

For a partition $\gamma \in \widetilde{\Gamma}_{k,\ell}$, suppose that $\gamma$ contains $s$ sets of size 2 $\{i_1, j_1\}, \ldots, \{i_s, j_s\}$ with $i_1, \ldots, i_s \in [k]$, $j_1, \ldots, j_s \in k + [\ell]$, and let $\boldsymbol{a} = (i_1, \ldots, i_s)$, $\boldsymbol{b} = (j_1, \ldots, j_s)$. Then, we can write the corresponding basis element $T_\gamma \in \mathsf{LE}_{k \to \ell}$ as

$$T_\gamma(U) := \left[ [0,1]^\ell \ni \boldsymbol{y} \mapsto \int_{[0,1]^{k-s}} U(\boldsymbol{x_a}, \boldsymbol{x}_{[k]\backslash \boldsymbol{a}}) \, \mathrm{d}\lambda^{k-s}(\boldsymbol{x}_{[k]\backslash \boldsymbol{a}}) \Big|_{\boldsymbol{x_a} = \boldsymbol{y}_{\boldsymbol{b}-k}} \right]. \tag{10}$$

In comparison to the basis of Maron et al. (2019b), this corresponds precisely to the basis elements for which no diagonals of the input are selected, and the output is always replicated on the entire space. We also note that the choice $p = 2$ in Definition 4.1 is somewhat arbitrary, and $T_\gamma$ can indeed be seen as an operator $L^p \to L^p$ for any $p \in [1, \infty]$, with $\|T_\gamma\|_{p \to p} = 1$ (see § E.2). We also briefly analyze the asymptotic dimension of $\mathsf{LE}_{k \to \ell}$ compared to the discrete case in § E.3.

## 4.2 Definition of Invariant Graphon Networks

Using $\mathsf{LE}_{k \to \ell}$ as building blocks, we extend the definitions of IGNs from Maron et al. (2019b) to graphons. This also corresponds to the definition used by Cai & Wang (2022), with the restriction that linear equivariant layers are limited to $\mathsf{LE}_{k \to \ell}$.

**Definition 4.3** (In- and equivariant graphon networks). *Let $\varrho : \mathbb{R} \to \mathbb{R}$ be an activation. Let $S \in \mathbb{N}$, and for each $s \in \{0, \ldots, S\}$, let $k_s \in \mathbb{N}_0$, $d_s \in \mathbb{N}$. Set $(d_0, k_0) := (2, 2)$. An **Equivariant Graphon Network (EWN)** is a function that maps a graphon-signal $(W, f) \in \mathcal{WS}_r$ to*

$$\mathcal{N}^{\mathrm{EWN}}(W, f) := \left( \mathfrak{T}^{(S)} \circ \varrho \circ \cdots \circ \varrho \circ \mathfrak{T}^{(1)} \right)(W, f), \tag{11}$$

*where for each $s \in [S]$,*

$$\mathfrak{T}^{(s)} : \left( L^2[0,1]^{k_{s-1}} \right)^{d_{s-1}} \to \left( L^2[0,1]^{k_s} \right)^{d_s}, \quad \boldsymbol{U} \mapsto \boldsymbol{T}^{(s)}(\boldsymbol{U}) + \boldsymbol{b}^{(s)}, \tag{12}$$

*with $\boldsymbol{T}^{(s)} \in \left( \mathsf{LE}_{k_{s-1} \to k_s} \right)^{d_s \times d_{s-1}}$, $\boldsymbol{b}^{(s)} \in \left( \mathsf{LE}_{0 \to k_s} \right)^{d_s}$, and $\boldsymbol{T}^{(s)}(\boldsymbol{U})_i := \sum_{j=1}^{d_{s-1}} T_{ij}^{(s)}(\boldsymbol{U}_j)$ for $i \in [d_s]$. Here, the addition of the bias terms $\boldsymbol{b}^{(s)}$ and application of $\varrho$ are understood elementwise. $(W, f)$ is identified with $\left[ (x, y) \mapsto (W(x,y), f(x)) \right]$ in the first layer. An **Invariant Graphon Network (IWN)** is an EWN with $(d_S, k_S) = (1, 0)$, i.e., mapping to scalars.*

We call $\max_{s \in [S]} k_s$ the **order** of an EWN, and $k_s$ the orders of the individual layers. For notational convenience, we defined the individual biases as elements of $\mathsf{LE}_{0 \to k_s}$ (noting that the input space of such a function is a singleton). In the discrete setting, individual weights are assigned to biases that are constant over the entire tensor and all its diagonals. Here, however, this issue does not arise because $\dim \mathsf{LE}_{0 \to k}$ is always 1, so the bias is merely a scalar. Note that any IWN can be seen as a function $\mathcal{N}^{\mathrm{IWN}} : \widetilde{\mathcal{WS}}_r \to \mathbb{R}$, as it is invariant w.r.t. all $\varphi \in \overline{S}_{[0,1]}$ by the definition of $\mathsf{LE}_{k \to \ell}$.

We also immediately observe that IWNs yield a parametrization that is closely related to IGN-small proposed by Cai & Wang (2022), a subset of IGNs with more favorable convergence properties under regularity assumptions on the graphon; see Cai & Wang (2022, Theorem 4) and § B.3. In their work, they define IGN-small as continuous IGNs (i.e., defined on graphons with signals) for which grid-sampling commutes with application of the discrete/continuous version of the IGN.

**Proposition 4.4.** *Any IWN from (11) is an instance of IGN-small (Cai & Wang, 2022).*

The proof of Proposition 4.4 (see § E.4) is a direct application of invariance under discretization (Lemma E.2) and representation stability of the basis elements. While the IGN-small constraint applies to the entire multilayer network, we impose our boundedness condition on each linear equivariant layer *individually*. Consequently, it does *not* follow that every linear equivariant layer used in an IGN-small model must lie in $\mathsf{LE}_{k\to\ell}$. The crux of this discrepancy is that a graphon can, for example, be mapped into a higher-dimensional diagonal, from which the network might then compute its output by integrating over that diagonal. Although this overall procedure meets the IGN-small consistency requirement, each single layer involved may fail to be a bounded linear operator. Moreover, while IWNs only utilize a subset of the basis employed by IGNs, the following section will show that IWNs can still attain strong expressivity on par with the discrete setting. By Proposition 4.4, these expressivity results extend to IGN-small as well.

### 4.3 EXPRESSIVITY OF INVARIANT GRAPHON NETWORKS

We prove expressivity results for IWNs, namely that IWNs up to order $k$ are at least as powerful as $k$-WL for graphon-signals, and that they act as universal approximators in the $\delta_p$ distances on any compact *subset* of graphon-signals. The analysis relies on the signal-weighted homomorphism expressivity of IWNs (§ 3). Clearly, IWNs are continuous in all $\delta_p$ distances (see § E.5 for a proof):

**Lemma 4.5.** *Let $\mathcal{N}^{\text{IWN}} : \mathcal{WS}_r \to \mathbb{R}$ be an IWN with Lipschitz continuous nonlinearity $\varrho$. Then, $\mathcal{N}^{\text{IWN}}$ is Lipschitz continuous w.r.t. $\delta_p$ for each $p \in [1, \infty]$.*

As a first step towards analyzing expressivity, we show that IWNs can approximate signal-weighted homomorphism densities w.r.t. graphs of size up to their order. Inspired by Keriven & Peyré (2019), we explicitly model the product in the homomorphism densities and track the employed linear equivariant layers. Finally, the result follows via a tree decomposition of the graph:

**Theorem 4.6** (Approximation of signal-weighted homomorphism densities). *Let $r > 0$, $1 < k \in \mathbb{N}$, $\varrho : \mathbb{R} \to \mathbb{R}$ Lipschitz continuous and non-polynomial, and $F$ be a multigraph of treewidth $k - 1$, $\boldsymbol{d} \in \mathbb{N}_0^{v(F)}$. Fix $\varepsilon > 0$. Then there exists an IWN $\mathcal{N}^{\text{IWN}}$ of order $k$ such that for all $(W, f) \in \mathcal{WS}_r$*

$$\left| t((F, \boldsymbol{d}), (W, f)) - \mathcal{N}^{\text{IWN}}(W, f) \right| \leq \varepsilon. \tag{13}$$

The proof (see § E.6) is by induction on the tree decomposition of a graph. As we traverse the tree, we introduce IWN layers that add new nodes and marginalize over processed ones. We write $\mathfrak{F}_\varrho^{\text{IWN}}$ for the set of all IWNs w.r.t. nonlinearity $\varrho$, and $\mathfrak{F}_\varrho^{k\text{-IWN}}$ for the restriction to IWNs of order up to $k$. Note that as an immediate consequence of Theorem 4.6 we can see that IWNs are $k$-WL-expressive (refer to § E.7):

**Corollary 4.7** ($k$-WL expressivity). *$\mathfrak{F}_\varrho^{k\text{-IWN}}$ is at least as expressive as the $k$-WL test at distinguishing graphon-signals.*

As we know from Theorem 3.2 that two graphon-signals are weakly isomorphic if and only if $\{t((F, \boldsymbol{d}), \cdot)\}_{F,\boldsymbol{d}}$ agree for all simple graphs or multigraphs $F$, Theorem 4.6 gives us an immediate way to prove universal approximation when not restricting the tensor order.

**Corollary 4.8** ($\delta_p$-Universality of IWNs). *Let $r > 1$, $p \in [1, \infty)$, $\varrho : \mathbb{R} \to \mathbb{R}$ Lipschitz continuous and non-polynomial. For any compact $K \subset (\widetilde{\mathcal{WS}_r}, \delta_p)$, $\mathfrak{F}_\varrho^{\text{IWN}}$ is dense in the continuous functions $C(K)$ w.r.t. $\|\cdot\|_\infty$.*

The proof (see § E.8) is a straightforward application of the Stone-Weierstrass theorem: The span of the signal-weighted homomorphism densities forms a subalgebra that, by Theorem 3.2, is point separating. This result also crucially implies that IWNs can distinguish any two graphon-signals that are not weakly isomorphic. We also want to mention that while IWNs are continuous w.r.t. $\delta_\infty$, the proof of Theorem 3.2 does not extend to this case as $\|\cdot\|_\infty$ is not a smooth norm on $[0, 1]^2$.

## 5 CUT DISTANCE AND TRANSFERABILITY OF HIGHER-ORDER WNNS

In this section, we discuss the relation of IWNs and more general higher-order *graphon* neural networks (referred to as "WNNs") to the cut distance and their transferability. One example besides IWNs are *refinement-based* networks that emulate the $k$-WL test, such as the ones we define in § F.

## 5.1 Cut Distance Discontinuity and Convergence

We first note that all nontrivial IWNs are discontinuous w.r.t. cut distance, in the following sense:

**Proposition 5.1.** *Let $\varrho : [0, 1] \to \mathbb{R}$. Then, the assignment $\mathcal{W}_0 \ni W \mapsto \varrho(W) \in \mathcal{W}$, where $\varrho$ is applied pointwise, is continuous w.r.t. $\|\cdot\|_\square$ if and only if $\varrho$ is linear.*

See § G.1 for a proof. This is evident in that node-featured graphs obtained from sampling do not converge to their underlying graphon-signal, a phenomenon first observed by Cai & Wang (2022) in a related setting. This discontinuity is inherent to $k$-WL (Böker, 2023) because it uses multigraph homomorphism densities, which are discontinuous in the cut distance. As such, *any* $k$-WL expressive function defined on graphon-signals would exhibit this discontinuity. The consideration of multigraphs arises from a fundamental difference in how $k$-WL and 1-WL handle edges. For 1-WL, weighted edges are treated simply as *weights*, i.e., function values of a graphon only act through its shift operator and, thus, carry precisely the meaning of edge probabilities. For intuition, note that most operator norms of the shift operator are topologically equivalent to the cut norm (Janson, 2013, Lemma E.6). In contrast, typical $k$-WL expressive models capture the full *distribution* of these edge weights, rather regarding them as *edge features*. For IWNs (§ 4), this manifests through pointwise application of the nonlinearity on the graphon-signal. Although the parametrization of Maehara & NT (2019) as a linear combination of *simple* homomorphism densities is cut distance continuous, its purely conceptual formulation provides no clear guideline for selecting pattern graphs.

Note that, a priori, this might constitute a disadvantage of such higher-order models compared to MPNNs. In particular, while the space $(\widetilde{\mathcal{WS}}_r, \delta_\square)$ is compact—allowing for the direct application of the Stone-Weierstrass theorem and the derivation of generalization bounds as in Levie (2023)— no similar results hold for $(\widetilde{\mathcal{WS}}_r, \delta_p)$ with $p \in [1, \infty)$. Although one can of course restrict the domain to compact subsets, as done for Corollary 4.8, it is doubtful if real-world distributions of node-featured graphs (or graphon-signals) would be confined to these. One example of subsets that are indeed compact w.r.t. $\delta_p$ distances—though of limited utility, particularly in the context of graph limits—is the set of regular step graphons bounded by some maximum size. This is somewhat similar to the restriction of Keriven & Peyré (2019) in their universality proof for IGNs/EGNs.

## 5.2 Transferability

Often, one may analyze the convergence of a graph ML model $\mathcal{N}$ to an underlying limit to study the question of its *transferability*, i.e., if

$$\mathcal{N}(G_n, \boldsymbol{f}_n) \approx \mathcal{N}(G_m, \boldsymbol{f}_m) \tag{14}$$

holds when $(G_n, \boldsymbol{f}_n), (G_m, \boldsymbol{f}_m) \sim \mathbb{G}_n(W, f), \mathbb{G}_m(W, f)$ as $n, m \in \mathbb{N}$ grow. In the theory of graph limits, the convergence of (14) for any graph parameter $\mathcal{N}$ is also known as *estimability* (Lovász, 2012, § 15). For the *continuous* MPNNs converging under their respective graph limits, transferability is usually shown simply by invoking the triangle inequality (see, e.g., Ruiz et al. (2023); Le & Jegelka (2023)), which is not possible for typical higher-order models as pointed out in § 5.1. Even worse, it is not even guaranteed that random graphs sampled from a graphon-signal become close in the $\delta_p$ distances as they grow in size: For example, take $G_n^{(1)}, G_n^{(2)}$ to be independent Erdős–Rényi graphs of size $n \in \mathbb{N}$. By Lavrov (2023) and Lovász (2012, Theorem 9.30),

$$\liminf_{n \to \infty} \mathbb{E}\left[\delta_1(G_n^{(1)}, G_n^{(2)})\right] \geq \tfrac{1}{12} \tag{15}$$

(note that the expectation in (15) would tend to zero in $\delta_\square$). As, for example, we have seen in Corollary 4.8 that IWNs are universal on compact subsets w.r.t. $\delta_1$, one might expect there to be an "adversarial" IWN which does not converge to the same value for all such random graphs. However, it turns out that this discontinuity can be "fixed" for a large class of functions:

**Theorem 5.2** (Transferability). *Let $r > 1$. Let $\mathcal{N} : \widetilde{\mathcal{WS}}_r \to \mathbb{R}$ such that $\mathcal{N}$ is contained in the closure of*

$$\mathrm{span}\left\{t((F, \boldsymbol{d}), \cdot)\right\}_{F \text{ multigraph}, \boldsymbol{d} \in \mathbb{N}_0^{v(F)}} \subseteq C_b(\widetilde{\mathcal{WS}}_r, \delta_1) \tag{16}$$

*w.r.t. uniform convergence. Then, for any $(W, f) \in \mathcal{WS}_r$ and $(G_n, \boldsymbol{f}_n), (G_m, \boldsymbol{f}_m) \sim \mathbb{G}_n(W, f), \mathbb{G}_m(W, f)$,*

$$\mathbb{E}\left|\mathcal{N}(G_n, \boldsymbol{f}_n) - \mathcal{N}(G_m, \boldsymbol{f}_m)\right| \to 0, \qquad n, m \to \infty. \tag{17}$$

The proof (see § G.2) is immediate and consists of replacing a multigraph $F$ in any $t((F, \boldsymbol{d}), \cdot)$ by its corresponding simple graph $F^{\text{simple}}$, resulting in a function of $(W, f) \in \mathcal{WS}_r$ which is cut distance continuous. We can further see that the assumptions of Theorem 5.2 are fulfilled for IWNs, as well as more general functions which factorize over the color space of the $k$-WL test (see also § G.3):

**Corollary 5.3** (Transferability of higher-order WNNs). *The assumption of Theorem 5.2 holds for*

*(1) any IWN with continuous nonlinearity $\varrho$,*

*(2) any $\mathcal{N} : \widetilde{\mathcal{WS}}_r \to \mathbb{R}$ for which $\mathcal{N}(W, f) = \widetilde{\mathcal{N}}(\nu_{(W,f)}^{k\text{-WL}})$ for a continuous $\widetilde{\mathcal{N}} : \mathcal{P}(\mathbb{M}^k) \to \mathbb{R}$.*

Note that, while Theorem 5.2 guarantees convergence, it does not make any statement about the rate. For this, one could show that $\mathcal{N}$ restricted to finite graph-signals extends to a $\delta_\square$-Lipschitz continuous function on $\widetilde{\mathcal{WS}}_r$. The challenge here is that a simple and general closed form is not immediate for this extension. Yet, the following can be shown about a subclass of IWNs:

**Theorem 5.4** (Quantitative transferability of IWNs, informal). *Let $r > 1$. For any $\mathcal{N}^{\text{IWN}}$ in a universal class of 2-layer IWNs with real-analytic nonlinearity $\varrho$, there exists a constant $M_{\mathcal{N}^{\text{IWN}}} > 0$ such that for $(W, f) \in \mathcal{WS}_r$ and $(G_n, \boldsymbol{f}_n), (G_m, \boldsymbol{f}_m) \sim \mathbb{G}_n(W, f), \mathbb{G}_m(W, f)$ for large $n, m$,*

$$\mathbb{E} \left| \mathcal{N}^{\text{IWN}}(G_n, \boldsymbol{f}_n) - \mathcal{N}^{\text{IWN}}(G_m, \boldsymbol{f}_m) \right| \leq M_{\mathcal{N}^{\text{IWN}}} \left( (\log n)^{-1/2} + (\log m)^{-1/2} \right). \quad (18)$$

See § G.4 for more details. Note that the rate $(\log n)^{-1/2}$ in (18) comes from the sampling lemma, and can be substantially better under additional assumptions or different discretization techniques.

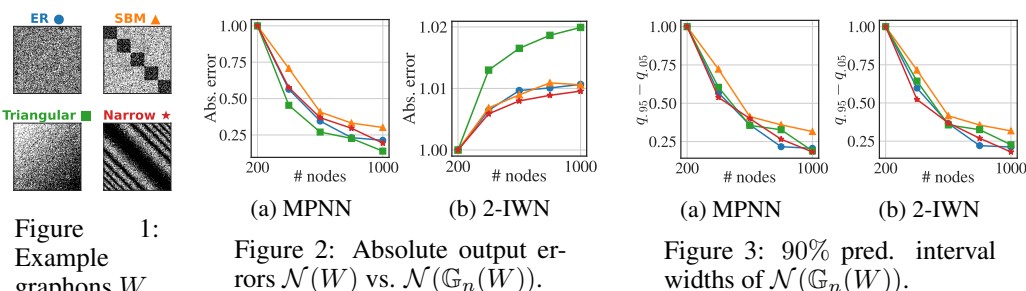

Figure 1: Example graphons $W$.

Figure 2: Absolute output errors $\mathcal{N}(W)$ vs. $\mathcal{N}(\mathbb{G}_n(W))$.

Figure 3: 90% pred. interval widths of $\mathcal{N}(\mathbb{G}_n(W))$.

We also validate our theoretical findings for IWNs with a proof-of-concept experiment on the graphons from Figure 1. For *continuity/convergence*, we plot the absolute errors of the model outputs for the sampled simple graphs in comparison to their graphon limits in Figure 2. Due to the $\delta_\square$-continuity of MPNNs, their errors decrease as the graph size grows. For the IWN, however, this does not hold. Yet, the errors for the IWN stabilize with increasing sizes, suggesting that the outputs converge (just *not* to their graphon limit). For *transferability*, we further plot prediction interval widths of the output distributions on simple graphs for each of the sizes in Figure 3. Here, the widths contract for both models and there are only minor differences visible between the MPNN and the IWN. This validates Theorem 5.2 and suggests that IWNs can indeed have similar transferability properties as MPNNs. For more details, see § H.

## 6 CONCLUSION

In this work, we study the expressivity, continuity, and transferability of graphon-based higher-order GNNs on the graphon-signal space (Levie, 2023) via *signal-weighted homomorphism densities*. We introduce Invariant Graphon Networks (IWNs) and analyze them through $L^p$ and cut distances on graphons. Significantly extending Cai & Wang (2022), we demonstrate that IWNs, as a subset of their IGN-small, retain the same expressive power as their discrete counterparts. Unlike MPNNs, IWNs are discontinuous w.r.t. cut distance, so standard transferability arguments (e.g., Ruiz et al. (2023); Levie (2023); Le & Jegelka (2023)) do not generalize. This stems from $k$-WL (Böker, 2023), so many $k$-WL expressive models on graphons would have the same limitations. Yet, we show that this discontinuity can be overcome in the sense that higher-order GNNs are still transferable.

One potential direction for future research could be to derive general, explicit, and tight bounds for Theorem 5.2, beyond the restricted setting of Theorem 5.4. Furthermore, one could analyze expressive spectral methods (Lim et al., 2023b;a; Huang et al., 2024). More broadly, future work could consider sparse graph limits (Le & Jegelka, 2023; Ruiz et al., 2024) or inductive biases through training, data distribution, or the specific task.

ACKNOWLEDGMENTS

The authors would like to thank Thien Le, Manish Krishan Lal, Dominik Fuchsgruber, and Levi Rauchwerger for insightful discussions at various stages of this work, and Andreas Bergmeister and Eduardo Santos Escriche for careful proofreading. This research was funded by an Alexander von Humboldt professorship.

REPRODUCIBILITY STATEMENT

We provide rigorous proofs of all our statements in the appendix, along with detailed explanations and the underlying assumptions. A comprehensive overview of our notation is listed in § A. Details for the toy experiment are provided in § H.

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

APPENDICES

# A  NOTATION

Table 1: We list the most important symbols used in this work.

| | |
|---|---|
| $\mathbb{N}; \mathbb{N}_0; \mathbb{Q}; \mathbb{R}$ | Natural, non-negative integer, rational, real numbers. |
| $[n]$ | Set $\{1, \ldots, n\}$ for $n \in \mathbb{N}$. |
| $\mathbb{1}_A$ | Indicator function in a set $A$. |
| $\{\cdot\}$ | Explicit list of elements in a set. |
| $\{\!\!\{\cdot\}\!\!\}$ | Explicit list of elements in a multiset. |
| $\varnothing$ | Empty set and empty tuple. |
| $V(G); v(G)$ | Node set of a graph; number of nodes of a graph $G$. |
| $E(G); e(G)$ | Edge (multi)set of a (multi)graph; number of edges of a (multi)graph $G$. |
| $\deg(u)$ | Degree of a node $u$ in a graph. |
| $\mathcal{O}(\cdot)$ | "Big-O" notation for asymptotic growth of a function. |
| $o(\cdot)$ | "Little-o" notation, indicating that one function is dominated by another. |
| $\mathcal{N}$ | Generic variable for a neural network (MLP or GNN). |
| $\mathfrak{F}$ | Generic variable for a function class of neural networks. |
| $\mathrm{tw}(F)$ | Treewidth of a (multi)graph $F$. |
| $\hom(F, G)$ | Number of homomorphisms from graph $F$ to $G$. |
| $\overline{A}$ | Closure of a subset $A$ of a topological space $\mathcal{X}$. |
| $\mathcal{B}(\mathcal{X})$ | Borel $\sigma$-algebra of a topological space $\mathcal{X}$. |
| $\sigma(\cdot)$ | Generated $\sigma$-algebra. |
| $\mathbb{P}$ | Probability measure. |
| $\mathbb{E}$ | Expected value. |
| $\lambda; \lambda^k$ | 1-dimensional Lebesgue measure; $k$-dimensional Lebesgue measure. |
| $L^p(\mathcal{X})$ | Space of $p$-integrable functions on a measure space $\mathcal{X}$, for $p \in [1, \infty]$. |
| $L_r^p(\mathcal{X})$ | Space of $p$-integrable functions, with norm bounded by $r$. |
| $\|\cdot\|_\square$ | Cut norm. |
| $\|\cdot\|_p$ | $L^p$ norm of functions on a measure space, for $p \in [1, \infty]$. |
| $\|\cdot\|_{p,\mathcal{X}}$ | $L^p$ norm, with emphasis on the underlying measure space $\mathcal{X}$. |
| $\|\cdot\|_{\mathrm{Lip}}$ | Lipschitz norm of a continuous function on a metric space. |
| $\mathcal{L}(V_1, V_2)$ | Space of bounded linear operators from normed vector space $V_1$ to $V_2$. |
| $\|T\|_{p \to q}$ | Operator norm of $T \in \mathcal{L}(L^p(\mathcal{X}), L^q(\mathcal{Y}))$. |
| $C(K)$ | Space of continuous functions from compact topological space $K$ into $\mathbb{R}$, with uniform norm $\|\cdot\|_\infty$. |
| $C_b(K)$ | Space of bounded continuous functions from topological space $K$ into $\mathbb{R}$, with uniform norm $\|\cdot\|_\infty$. |
| $\mathcal{W}$ | Space of kernels. |
| $\mathcal{W}_0$ | Space of graphons. |
| $\mathcal{WS}_r$ | Space of graphon-signals $\mathcal{W}_0 \times L_r^\infty[0, 1]$. |
| $\widetilde{\mathcal{W}}_0$ | Space of unlabeled graphons. |
| $\widetilde{\mathcal{WS}}_r$ | Space of unlabeled graphon-signals. |
| $T_W$ | Shift operator of a graphon $W$. |
| $S_{[0,1]}$ | Measure preserving (almost) bijections of $[0, 1]$. |
| $\overline{S}_{\mathcal{X}}$ | Measure preserving functions $\mathcal{X} \to \mathcal{X}$, for a measure space $\mathcal{X}$. |
| $\delta_\square$ | Cut distance. |
| $\delta_p$ | (Unlabeled) $L^p$ distance for graphons/kernels. |
| $\delta_N$ | (Unlabeled) distance w.r.t. smooth invariant norms $N = (N_1, N_2)$. |
| $\xrightarrow{w}$ | Weak convergence of probability measures. |
| $f_*\mu$ | Pushforward of measure $\mu$ under $f$. |
| $W_G$ | Step graphon of a graph $G$. |
| $\mathbb{H}_k(W); \mathbb{H}_k(W, f)$ | Distribution of weighted graphs/graph-signals of size $k$ sampled from a graphon $W$/graphon-signal $(W, f)$. |
| $\mathbb{G}_k(W); \mathbb{G}_k(W, f)$ | Distribution of unweighted graphs/graph-signals of size $k$ sampled from a graphon $W$/graphon-signal $(W, f)$. |
| $U(0, 1)$ | Uniform distribution on the interval $[0, 1]$. |
| $t(F, W)$ | Homomorphism density from a (multi)graph $F$ into graphon $W$. |

| | |
|---|---|
| $t((F, \boldsymbol{d}), (W, f))$ | Signal-weighted homomorphism density from a (multi)graph $F$, $\boldsymbol{d} \in \mathbb{N}_0^{v(F)}$, into graphon-signal $(W, f)$. |
| $\boldsymbol{F} = (F, \boldsymbol{a}, \boldsymbol{b}, \boldsymbol{d})$ | Tri-labeled graph. |
| $\mathcal{M}^{k,\ell}$ | Set of all tri-labeled graphs with $k$ input, $\ell$ output vertices. |
| $\boldsymbol{F}_1 \circ \boldsymbol{F}_2$ | Composition of tri-labeled graphs. |
| $\boldsymbol{F}_1 \cdot \boldsymbol{F}_2$ | Schur product of tri-labeled graphs. |
| $\mathcal{F}^k$ | Set of atomic tri-labeled graphs. |
| $\langle \mathcal{F} \rangle$ | Set of $\mathcal{F}$-terms. |
| $\llbracket \mathbb{F} \rrbracket$ | Evaluation of a term $\mathbb{F} \in \langle \mathcal{F} \rangle$. |
| $h(\mathbb{F})$ | Height of a term $\mathbb{F}$. |
| $T_{\boldsymbol{F} \to (W, f)}$ | Graphon-signal operator associated with tri-labeled graph $\boldsymbol{F}$. |
| $\mathbb{M}_s^k$ | Space of $k$-WL colors up to step $s$. |
| $\mathbb{M}^k$ | Space of $k$-WL colors. |
| $p_{t \to s}$; $p_{\infty \to s}$ | Canonical projections $\mathbb{M}_t^k \to \mathbb{M}_s^k$, $\mathbb{M}^k \to \mathbb{M}_s^k$. |
| $\mathbb{P}^k$ | Space of $k$-WL refinements. |
| $F^{\mathbb{F}}$ | Realizing functions on $\mathbb{M}^k$ of a term $\mathbb{F}$. |
| $t^{\mathbb{F}}$ | Realizing function on $\mathbb{P}^k$ of a term $\mathbb{F}$. |
| $\mathfrak{C}_{(W,f)}^{k\text{-WL},(s)}$ | $k$-WL measure of $(W, f) \in \mathcal{WS}_r$. |
| $\nu_{(W,f)}^{k\text{-WL}}$ | $k$-WL distribution of $(W, f) \in \mathcal{WS}_r$. |
| $\mathsf{LE}_{k \to \ell}^{\mathcal{X}}$ | Linear equivariant layers on measure space $\mathcal{X}$. |
| $\mathsf{LE}_{k \to \ell}$ | Linear equivariant layers on $[0, 1]$. |
| $\mathcal{F}_k^{(n)}$ | Regular step functions in $L^2[0, 1]^k$ at resolution $n$. |
| $\Gamma_m$ | Set of partitions of $[m]$, $m \in \mathbb{N}_0$. |
| $\mathsf{bell}(m)$ | $|\Gamma_m|$, i.e., number of partitions of $[m]$. |
| $\widetilde{\Gamma}_{k,\ell}$ | Set of partitions of $[k + \ell]$ that index a basis of $\mathsf{LE}_{k \to \ell}$. |
| $T_\gamma$ | Linear operator w.r.t. $\gamma \in \Gamma_{k+\ell}$. |

## B EXTENDED BACKGROUND

### B.1 TOPOLOGY AND MEASURE THEORY

We briefly recall fundamental definitions and results from topology, measure theory, and the theory of measures on Polish spaces that are used in this work. See, for example, Simon (2015) or Elstrodt (2018) for comprehensive primers.

#### B.1.1 TOPOLOGY

A **topological space** is a pair $(\mathcal{X}, \mathcal{O})$, where $\mathcal{X}$ is a set and the **topology** $\mathcal{O} \subseteq 2^{\mathcal{X}}$ is a collection satisfying that $\mathcal{X}, \varnothing \in \mathcal{O}$, $\bigcup_{\iota \in I} U_\iota \in \mathcal{O}$ for any family $\{U_\iota\}_{\iota \in I}$ of $U_\iota \in \mathcal{O}$, and $U_1 \cap \cdots \cap U_n \in \mathcal{O}$ for any $U_1, \ldots, U_n \in \mathcal{O}$. I.e., a topology is closed under arbitrary unions and finite intersections. A set $U \in \mathcal{O}$ is called **open**, and a set $A \in 2^{\mathcal{X}}$ is **closed** if its complement $A^{\mathsf{c}} := \mathcal{X} \setminus A$ is open. The **closure** $\overline{A}$ of a set $A \subseteq \mathcal{X}$ is the smallest closed set containing $A$. A subset $A \subset \mathcal{X}$ is **dense** if its closure $\overline{A} = \mathcal{X}$. A topological space is **separable** if it has a countable dense subset. A **neighborhood** of $x \in \mathcal{X}$ is a set $A \subseteq \mathcal{X}$ such that there is an open set $U$ with $x \in U \subseteq A$. If the topology is clear from context, it is often left implicit.

A **metric space** is a pair $(\mathcal{X}, d)$ with metric $d \colon \mathcal{X} \times \mathcal{X} \to [0, \infty)$ satisfying positive definiteness $d(x, y) = 0$ iff $x = y$, symmetry $d(x, y) = d(y, x)$, and the triangle inequality $d(x, z) \leq d(x, y) + d(y, z)$ (here, $x, y, z \in \mathcal{X}$). A **pseudometric** is a function $d$ that satisfies all of the previous requirements except positive definiteness. A metric $d$ on $\mathcal{X}$ induces a topology by choosing the coarsest topology on $\mathcal{X}$ under which all balls $B_\varepsilon(x) := \{y \in \mathcal{X} \,|\, d(x, y) < \varepsilon\}$ for $x \in \mathcal{X}$, $\varepsilon > 0$, are open. A topological space is **metrizable** if its topology can be induced by a metric. A topological space is **Hausdorff** if for any two points $x \neq y$ there exists a pair of disjoint neighborhoods. Metric spaces are trivially Hausdorff by positive definiteness. A metric space is **complete** if every Cauchy sequence in it converges to a point in the space. Note that this is a property of the metric itself and not of the induced topology. Spaces like $\mathbb{R}$ or $\mathbb{R}^n$ are typically considered with their **standard topology**, i.e. the one induced by the Euclidean norm/distance (which is the same for all norms).

A function $f \colon \mathcal{X} \to \mathcal{Y}$ between topological spaces $(\mathcal{X}, \mathcal{O}_{\mathcal{X}})$ and $(\mathcal{Y}, \mathcal{O}_{\mathcal{Y}})$ is **continuous** if for every open set $V \in \mathcal{O}_{\mathcal{Y}}$, one has $f^{-1}(V) \in \mathcal{O}_{\mathcal{X}}$. For metric spaces with their induced topology, this is equivalent to the standard definitions of continuity (via $\varepsilon$-$\delta$ or sequences).

An open cover of a topological space $\mathcal{X}$ is a family of open sets whose union is all of $\mathcal{X}$. A topological space $\mathcal{X}$ is **compact** if every open cover of $\mathcal{X}$ has a finite subcover; in metric spaces this is equivalent to every sequence in $\mathcal{X}$ having a convergent subsequence.

The **product topology** on $\prod_{\iota \in I} \mathcal{X}_\iota$ is the coarsest topology making all projections $p_j \colon \prod_{\iota \in I} \mathcal{X}_\iota \to X_j$ continuous, and if $A \subseteq \mathcal{X}$, the **subspace topology** on $A$ is $\{U \cap A \,|\, U \text{ open in } \mathcal{X}\}$. The convergence of a sequence in a topological product space is equivalent to convergence of all of its components, i.e., images under the projections $p_\iota$. A subset $A \subseteq \mathcal{X}$ is **relatively compact** if its closure is compact. By the **Heine-Borel theorem**, subsets of $\mathbb{R}^n$ are compact iff they are closed and bounded (in any norm).

A closed subset of a compact space is compact w.r.t. the subspace topology. The converse also holds if the space is Hausdorff. **Tychonoff's theorem** states that *any* product $\prod_{\iota \in I} \mathcal{X}_\iota$ of compact topological spaces is again compact (regardless of the cardinality of $I$). A topological space is **normal** if any two disjoint closed subsets have disjoint open neighborhoods (i.e., open sets containing them). Every metrizable space is normal. The **Tietze extension theorem** states that if $\mathcal{X}$ is normal and $A \subseteq \mathcal{X}$ is closed, then any continuous function $f \colon A \to \mathbb{R}$ can be extended to a continuous function $\widetilde{f} \colon \mathcal{X} \to \mathbb{R}$.

Let $K$ be a compact Hausdorff space. Write $C(K, \mathbb{R})$ for the space of all continuous functions on $K$ (which are all bounded), equipped with the topology of uniform convergence, i.e., $\|\cdot\|_\infty$. With pointwise addition and multiplication, this space becomes an algebra. The **Stone-Weierstrass theorem** states that if $A \subset C(K, \mathbb{R})$ is a subalgebra that separates points in $K$ and contains the constant functions, then $A$ is dense in $C(K, \mathbb{R})$.

### B.1.2 Measure Theory

A **measurable space** $(\mathcal{X}, \mathcal{A})$ is a pair where $\mathcal{X}$ is an underlying set, and $\mathcal{A} \subseteq 2^{\mathcal{X}}$ is a $\sigma$**-algebra**, which fulfills $\varnothing \in \mathcal{A}$, and is stable under complements as well as *countable* unions. Note that this also implies stability under countable intersections. A **measure space** $(\mathcal{X}, \mathcal{A}, \mu)$ is a tuple, where $(\mathcal{X}, \mathcal{A})$ is a measurable space and $\mu$ is a **measure**, i.e., a function from $\mathcal{A}$ to $[0, \infty]$ which satisfies $\mu(\varnothing) = 0$, and $\sigma$-*additivity* $\mu(\bigcup_{n \in \mathbb{N}} A_n) = \sum_{n \in \mathbb{N}} \mu(A_n)$ for any disjoint $\{A_n\}_n \subseteq \mathcal{A}$. If the $\sigma$-algebra and/or measure is clear from context, we omit it. For any $\mathcal{G} \subseteq \mathcal{X}$, define its **generated** $\sigma$**-algebra** $\sigma(\mathcal{G})$ as the smallest $\sigma$-algebra containing $\mathcal{G}$ (this is well-defined as it is simply the intersection of all $\sigma$-algebras containing $\mathcal{G}$). A property of $\mathcal{X}$ holds **almost everywhere** (a.e.) if the set $N \in \mathcal{A}$ on which this property does not hold is a **null set**, i.e., $\mu(N) = 0$. A **probability space** $(\mathcal{X}, \mathcal{A}, \mathbb{P})$ is a measure space with $\mathbb{P}(\mathcal{X}) = 1$.

The **Borel** $\sigma$**-algebra** $\mathcal{B}(\mathcal{X})$ on a topological space $\mathcal{X}$ is the $\sigma$-algebra generated by its open sets. Any continuous function on such a space is also measurable. The **Lebesgue measure** $\lambda$ on $\mathbb{R}$ is the unique measure on $\mathcal{B}(\mathbb{R})$ assigning intervals to its length. Analogously, $\lambda^n$ on $\mathcal{B}(\mathbb{R}^n)$ is the unique measure assigning $n$-dimensional cuboids to its volume. These assignments determine the Lebesgue measure uniquely under translation invariance and regularity conditions. Often, the Lebesgue measure is considered on the *Lebesgue $\sigma$-algebra*, which is the completion of the Borel $\sigma$-algebra, containing all subsets of null sets. For this work, the distinction between both is not important, and we will work just with Borel sets. The **counting measure** on a set $\mathcal{X}$ is defined by mapping each subsets to its cardinality. Similarly to topologies, we can define **subspace** and **product** $\sigma$**-algebras**.

A function $f : (\mathcal{X}, \mathcal{A}_{\mathcal{X}}) \to (\mathcal{Y}, \mathcal{A}_{\mathcal{Y}})$ between two measurable spaces is **measurable** if $f^{-1}(A) \in \mathcal{A}_{\mathcal{X}}$ for every $A \in \mathcal{A}_{\mathcal{Y}}$. The **Lebesgue integral** of a measurable function $f : (\mathcal{X}, \mathcal{A}, \mu) \to (\mathbb{R}, \mathcal{B}(\mathbb{R}))$ is denoted by $\int_{\mathcal{X}} f \, \mathrm{d}\mu$, and is defined via taking a.e. limits of indicator and step functions.

A linear operator $T \colon (V, \|\cdot\|_V) \to (W, \|\cdot\|_W)$ between two normed spaces is **bounded** iff $T$ is continuous w.r.t. their induced metrics, which is equivalent to $\|T\|_{V \to W} := \sup_{\|v\|_V = 1} \|Tv\|_W < \infty$. For a measure space $(\mathcal{X}, \mathcal{A}, \mu)$ and $1 \leq p < \infty$, its $L^p$ space is defined as $L^p(\mathcal{X}) = \{f \colon \mathcal{X} \to \mathbb{R} \mid f \text{ is measurable}, \|f\|_p = (\int_{\mathcal{X}} |f|^p \, \mathrm{d}\mu)^{1/p} < \infty\}$. $L^\infty(\mathcal{X})$ is defined via the essential supremum, $\|f\|_\infty = \mathrm{ess\,sup} \, f := \inf\{M \geq 0 \mid |f(x)| \leq M \text{ a.e.}\}$. Functions that agree a.e. are identified.

In $L^p$ spaces, the following inequalities hold: **Minkowski's inequality** states that $\|f + g\|_p \leq \|f\|_p + \|g\|_p$. **Jensen's inequality** states that in a probability space $\mathcal{X}$ and for a convex function $\phi \colon \mathbb{R} \to \mathbb{R}$, $\phi\left(\int_{\mathcal{X}} f \, d\mu\right) \leq \int_{\mathcal{X}} \phi(f) \, \mathrm{d}\mu$. **Hölder's inequality** states that for *dual coefficients* $p, q \in [1, \infty]$ with $1/p + 1/q = 1$, we have $\int_{\mathcal{X}} |fg| \, d\mu \leq \|f\|_p \|g\|_q$.

If $f \colon \mathcal{X} \to \mathcal{Y}$ is measurable and $\mu$ is a measure on $\mathcal{X}$, the **pushforward** measure $f_* \mu$ on $\mathcal{Y}$ is defined by $f_* \mu(A) := \mu\left(f^{-1}(A)\right)$ for all measurable $A \subseteq \mathcal{Y}$. If $g : \mathcal{Y} \to \mathbb{R}$ is measurable, then $\int_{\mathcal{Y}} g \, \mathrm{d}f_* \mu = \int_{\mathcal{X}} g \circ f \, \mathrm{d}\mu$ holds for the Lebesgue integral. A **measure preserving function** between two measure spaces $(\mathcal{X}, \mu)$ and $(\mathcal{Y}, \nu)$ is a measurable function $\varphi : \mathcal{X} \to \mathcal{Y}$ such that $\varphi_* \mu = \nu$. Two measure spaces $\mathcal{X}$ and $\mathcal{Y}$ are **isomorphic** if there exists a measure preserving *bijection* between them whose inverse is also measure preserving. The spaces are **almost isomorphic** if the former holds for some subsets of full measure of $\mathcal{X}, \mathcal{Y}$.

### B.1.3 Measures on Polish Spaces

A **Polish space** $\mathcal{X}$ is a topological space that is separable and *completely* metrizable, meaning that its topology is induced by a metric $d$ w.r.t. which $(\mathcal{X}, d)$ is complete. We typically consider a Polish space with its Borel $\sigma$-algebra. A **standard Borel probability space** is a probability space defined on the Borel $\sigma$-algebra of a Polish space. Notably, by the **isomorphism theorem**, every *nonatomic* standard Borel probability space (meaning there are no points of positive measure) is almost isomorphic to the unit interval $([0, 1], \mathcal{B}([0, 1]), \lambda)$. By renormalization, a similar result holds for all *finite* measures.

Let $\mathcal{P}(\mathcal{X})$ be the set of all Borel probability measures on a Polish space $\mathcal{X}$. We equip this set with a topology as well: The **weak topology** of $\mathcal{P}(\mathcal{X})$ is the coarsest topology making all the maps $\mu \mapsto \int_{\mathcal{X}} f \, \mathrm{d}\mu$ continuous, where $f \in C_b(\mathcal{X})$ is considered over the *bounded* continuous functions

on $\mathcal{X}$. This corresponds to the weak-$*$-topology on the dual $C_b(\mathcal{X})^*$, i.e., bounded linear functionals on $(C_b(\mathcal{X}), \|\cdot\|_\infty)$. A sequence $(\mu_n)_n$ of probability measures on $\mathcal{X}$ is said to **converge weakly** to $\mu$ (denoted $\mu_n \xrightarrow{w} \mu$) if $\int_\mathcal{X} f \, \mathrm{d}\mu_n \to \int_\mathcal{X} f \, \mathrm{d}\mu$ for every $f \in C_b(\mathcal{X})$.

The **Portmanteau theorem** gives several equivalent formulations of weak convergence (for example, convergence of the measures evaluated on continuity sets, or convergence of the integrals only for a dense subset of $C_b(\mathcal{X})$). On $\mathcal{X} = \mathbb{R}$, this is precisely convergence of random variables *in distribution*. Notably, if $\mathcal{X}$ is Polish, then so is $\mathcal{P}(\mathcal{X})$, and if $\mathcal{X}$ is metrizable and compact, this also carries over to $\mathcal{P}(\mathcal{X})$.

By **Prokhorov's theorem**, a family of probability measures on a Polish space $\mathcal{X}$ is relatively compact (with respect to the weak topology) iff it is tight, meaning that the total probability mass can be approximated arbitrarily well by compact subsets $K_\varepsilon \subseteq \mathcal{X}$ *uniformly* on the family of measures.

### B.2 CHARACTERIZATION OF THE IGN BASIS

We restate the characterization of the IGN basis introduced by Cai & Wang (2022). As described by Maron et al. (2019b), $\dim \mathsf{LE}^{[n]}_{k \to \ell} = \mathrm{bell}(k + \ell)$, i.e., the number of partitions $\Gamma_{k+\ell}$ of the set $[k + \ell]$. In the basis of Cai & Wang (2022), each basis element $L_\gamma^{(n)}$ associated with a partition $\gamma \in \Gamma_{k+\ell}$ can be characterized as a sequence of basic operations.

Given $\gamma \in \Gamma_{k+\ell}$, divide $\gamma$ into 3 subsets $\gamma_1 := \{A \in \gamma \mid A \subseteq [k]\}$, $\gamma_2 := \{A \in \gamma \mid A \subseteq k + [\ell]\}$, $\gamma_3 := \gamma \setminus (\gamma_1 \cup \gamma_2)$. Here, the numbers $1, \ldots, k$ are associated with the input axes and $k+1, \ldots, k+\ell$ with the output axes respectively.

① (*Selection*: $\boldsymbol{H} \mapsto \boldsymbol{H}_\gamma$). In a first step, we specify which part of the input tensor $\boldsymbol{H} \in \mathbb{R}^{n^k}$ is under consideration. Take $\gamma\big|_{[k]} := \{A \cap [k] \mid A \in \gamma, A \cap [k] \neq \varnothing\}$ and construct a new $|\gamma_1| + |\gamma_3| = |\gamma\big|_{[k]}|$-tensor $\boldsymbol{H}_\gamma$ by selecting the diagonal of the $k$-tensor $\boldsymbol{H}$ corresponding with the partition $\gamma\big|_{[k]}$.

② (*Reduction*: $\boldsymbol{H}_\gamma \mapsto \boldsymbol{H}_{\gamma,\mathbf{red}}$). We average $\boldsymbol{H}_\gamma$ over the axes $\gamma_1 \subseteq \gamma\big|_{[k]}$, resulting in a tensor $\boldsymbol{H}_{\gamma,\mathbf{red}}$ of order $|\gamma_3|$, indexed by $\gamma_3\big|_{[k]}$.

③ (*Alignment*: $\boldsymbol{H}_{\gamma,\mathbf{red}} \mapsto \boldsymbol{H}_{\gamma,\mathbf{align}}$). We align $\boldsymbol{H}_{\gamma,\mathbf{red}}$ with a $|\gamma_3|$-tensor $\boldsymbol{H}_{\gamma,\mathbf{align}}$ indexed by $\gamma_3\big|_{k+[\ell]}$, sending for $A \in \gamma_3$ the axis $A \cap [k]$ to $A \cap [\ell]$.

④ (*Replication*: $\boldsymbol{H}_{\gamma,\mathbf{align}} \mapsto \boldsymbol{H}_{\gamma,\mathbf{rep}}$). Replicate the $|\gamma_3|$-tensor $\boldsymbol{H}_{\gamma,\mathbf{align}}$ indexed by $\gamma_3\big|_{k+[\ell]}$ along the axes in $\gamma_2$. Note that if $\gamma_3\big|_{k+[\ell]} \cup \gamma_2$ contains non-singleton sets, the output tensor is supported on some diagonal.

Aggregations in this procedure can either be normalized (as described here) or simple sums. The basis element $T_\gamma^{(n)} : \mathbb{R}^{n^k} \to \mathbb{R}^{n^\ell}$ can now be described by the assignment $T_\gamma^{(n)}(\boldsymbol{H}) := \boldsymbol{H}_{\gamma,\mathbf{rep}}$, and

$$\mathsf{LE}^{[n]}_{k \to \ell} = \mathrm{span}\left\{ T_\gamma^{(n)} \,\Big|\, \gamma \in \Gamma_{k+\ell} \right\}. \tag{19}$$

### B.3 IGN-SMALL (CAI & WANG, 2022)

Cai & Wang (2022) study the convergence of discrete IGNs applied to graphs sampled from a graphon to a continuous version of the IGN defined on graphons. For this, they use the full IGN basis and a *partition norm*, which is for $U \in L^2[0,1]^k$ a $\mathrm{bell}(k)$-dimensional vector consisting of $L^2$ norms of $U$ on all possible diagonals. While they show that convergence of a discrete IGN on weighted graphs sampled from a graphon to its continuous counterpart holds, they also demonstrate that this is not the case for unweighted graphs with $\{0, 1\}$-valued adjacency matrix.

As a remedy, Cai & Wang (2022) constrain the IGN space to *IGN-small*, which consists of IGNs for which applying the discrete version to a grid-sampled step graphon yields the same output as applying the continuous version and grid-sampling afterwards. In the following definition, we will formalize this. Here, $\mathcal{F}_k^{(n)}$ denotes the regular *$k$-dimensional step kernels* on $[0,1]$, $k \in \mathbb{N}_0$.

**Definition B.1** (IGN-small (Cai & Wang, 2022)). *Let $\mathcal{N}$ be defined as in Definition 4.3, with the only difference that $\mathsf{LE}_{k\to\ell}$ is replaced by the full IGN basis, where averaging steps should be understood as integration. Cai & Wang (2022) call such $\mathcal{N}$ a **continuous IGN**. For any basis element $T_\gamma$, $\gamma \in \Gamma_{k+\ell}$, denote its discrete version at resolution $n \in \mathbb{N}$ by $T_\gamma^{(n)}$ and the network obtained by discretizing all equivariant linear layers by $\mathcal{N}^{(n)}$. Let*

$$S^{(n)} : \mathbb{R}^{[0,1]^k \times d} \to \mathbb{R}^{n^k \times d}, \; \boldsymbol{U} \mapsto (U(i/n, j/n))_{i,j=1}^n \tag{20}$$

*be the grid-sampling operator. Then, $\mathcal{N}$ is contained in **IGN-small** if*

$$(S^{(n)} \circ \mathcal{N})(W, f) \;=\; (\mathcal{N}^{(n)} \circ S^{(n)})(W, f) \tag{21}$$

*for any $(W, f) \in \mathcal{WS}_r$ such that $W \in \mathcal{F}_2^{(n)}$, $f \in \mathcal{F}_1^{(n)}$. In this case, the input to such an IGN is $[(x, y) \mapsto (W(x, y), f(x))]$.*

Cai & Wang (2022) show that convergence of IGN-small can be achieved in a model where a $\{0, 1\}$-valued adjacency matrix is sampled from the graphon, provided that certain assumptions on the graphon and the signal—such as Lipschitz continuity—and a prior estimation of an edge probability are satisfied (see Theorem 4). It is important to note, however, that assuming the graphon is continuous is a rather strong condition, as it implies a topological structure on the node set corresponding to the unit interval. In contrast, similarly to Levie (2023), we treat $[0, 1]$ solely as a *measure space*, which, being almost isomorphic to any nonatomic standard Borel probability space, is much more general. Regarding the expressivity of IGN-small, Cai & Wang (2022) establish that this model class can approximate spectral GNNs with arbitrary precision (cf. Theorem 5).

### B.4 Tree Decomposition and Treewidth

In this section, we will recall the tree decomposition of a graph and the related notion of *treewidth*, which essentially captures how "far" a graph is from being a tree. See for example Diestel (2017, § 12.3) for a more in-depth discussion of this fundamental graph theoretic concept. We use the specific notation of Böker (2023).

**Definition B.2** (Tree Decomposition of a Graph). *Let $G$ be a graph. A **tree decomposition** of $G$ is a pair $(T, \beta)$, where $T$ is a tree and $\beta : V(T) \to 2^{V(G)}$ such that*

*(1) for every $v \in V(G)$, the set $\{t \,|\, v \in \beta(t)\}$ is nonempty and connected in $T$,*
*(2) for every $e \in E(G)$, there is a node $t \in V(T)$ such that $e \subseteq \beta(t)$.*

For $t \in V(T)$, the sets $\beta(t) \subseteq V(G)$ are commonly referred to as *bags* of the tree decomposition. Note that every graph $G$ has a trivial tree decomposition, given by a tree consisting of one node, with the bag being the entire node set $V(G)$. However, we are generally interested in finding tree decompositions with smaller bags. This leads us to the concept of *treewidth*:

**Definition B.3** (Treewidth of a Graph). *Let $G$ be a graph. For any tree decomposition $(T, \beta)$ of $G$, define its **width** as*

$$\max\{|\beta(t)| \,|\, t \in V(T)\} - 1. \tag{22}$$

*The **treewidth** of a graph $G$ is then the minimum width of all tree decompositions of $G$.*

Note that, the edge graph of a tree $G$ can be seen as a tree decomposition of $G$, with each edge being a bag. Hence, the treewidth of a tree is 1. It can also be shown that, e.g., the treewidth of a circle of size at least 3 is 2. The definition can be extended to multigraphs by simply ignoring the edge multiplicities, i.e., considering the *set* of edges instead of the multiset.

### B.5 Graphon-Signal Space (Levie, 2023)

Without reintroducing the graphon-signal space (see § 2.3 for the basic definitions), we formally restate two of the main results of Levie (2023) relevant to this work. Central to their contribution, Levie (2023) proves compactness of the graphon-signal space and provides a bound on its covering number. Note that something similar does *not* hold for any of the $\delta_p$ distances.

**Theorem B.4** (Levie (2023), Theorem 3.6). *The space $(\widetilde{\mathcal{WS}}_r, \delta_\square)$ is compact. Moreover, given $r > 0$ and $c > 1$, for every sufficiently small $\varepsilon > 0$, the space can be covered by $2^{k^2}$ balls of radius $\varepsilon$, where $k = \lceil 2^{\frac{9c}{4\varepsilon^2}} \rceil$.*

The proof follows an approach analogous to that used for establishing the compactness of the space of unlabeled graphons and relies on a graphon-signal adaptation of the weak regularity lemma (Levie, 2023, Theorem B.6). The graphon-signal weak regularity lemma can also be used to derive a sampling lemma:

**Theorem B.5** (Levie (2023), Theorem 3.7). *Let $r > 1$. There exists a constant $N_0 > 0$ depending on $r$, such that for every $n \geq N_0$ and $(W, f) \in \mathcal{WS}_r$ we have*

$$\mathbb{E}\left[\delta_\square\big((W, f), \mathbb{G}_n(W, f)\big)\right] \leq \frac{15}{\sqrt{\log n}}. \tag{23}$$

Although the above results were obtained for one-dimensional signals, they readily extend to $d$-dimensional signals taking values in compact sets $K \subset \mathbb{R}^d$ (say, a hypercube $[-r, r]^d$) based on a multidimensional version of the signal cut norm normalized by $d$ (see also Rauchwerger & Levie (2025)). In this case, the exact statements of Theorem B.4 and Theorem B.5 can be recovered. By norm equivalence, qualitative statements of the theorems remain valid when using the $L^1$ norm $\|\boldsymbol{f}\|_1$ as signal norm. The same reasoning extends to any $L^1$ norm defined from a vector norm $\|\cdot\|_{\mathbb{R}^s}$; that is, if one sets

$$\|\boldsymbol{f}\| := \int_{[0,1]} \|\boldsymbol{f}(x)\|_{\mathbb{R}^s} \, \mathrm{d}\lambda(x), \tag{24}$$

as well as to other $L^p$ norms of $\boldsymbol{f}$: If for all signals $\boldsymbol{f}$, $x \mapsto \|\boldsymbol{f}(x)\|_p$ is uniformly bounded by $r$ on $[0, 1]$, then

$$\int_{[0,1]} \|\boldsymbol{f}(x)\|_p \, \mathrm{d}\lambda(x) \leq \underbrace{\left(\int_{[0,1]} \|\boldsymbol{f}(x)\|_p^p \, \mathrm{d}\lambda(x)\right)^{1/p}}_{=\|\boldsymbol{f}\|_p} \leq r^{(p-1)/p} \left(\int_{[0,1]} \|\boldsymbol{f}(x)\|_p \, \mathrm{d}\lambda(x)\right)^{1/p}, \tag{25}$$

where we used Jensen's inequality for the first part.

## C   SIGNAL-WEIGHTED HOMOMORPHISM DENSITIES

### C.1   A COUNTING LEMMA FOR GRAPHON-SIGNALS

We derive a counting lemma similar to the standard graphon case (Lovász, 2012, Lemma 10.23), which shows that signal-weighted homomorphism densities from *simple* graphs into a graphon-signal are Lipschitz continuous w.r.t. cut distance.

**Proposition C.1** (Counting lemma for graphon-signals). *Let* $(W, f), (V, g) \in \mathcal{WS}_r$ *and* $F$ *be a simple graph,* $\boldsymbol{d} \in \mathbb{N}_0^{v(F)}$. *Then, writing* $D := \sum_{i \in V(F)} d_i$,

$$\left| t((F, \boldsymbol{d}), (W, f)) - t((F, \boldsymbol{d}), (V, g)) \right| \leq 2r^{D-1} \Big( 2r \cdot e(F) \left\| W - V \right\|_{\square} + D \left\| f - g \right\|_{\square} \Big). \quad (26)$$

As $t((F, \boldsymbol{d}), \cdot)$ is clearly invariant w.r.t. measure preserving functions acting on the graphon-signal, the bound of (26) can be easily extended to $\delta_{\square}$. The proof is relatively straightforward, with the only detail requiring a little extra consideration being that the signals can take negative values and interact with the graphon. A similar statement can be shown for all *multigraphs* $F$ using $\left\| \cdot \right\|_1$.

*Proof.* We split the l.h.s., bounding the difference of the graphons and the signals separately:

$$\left| t((F, \boldsymbol{d}), (W, f)) - t((F, \boldsymbol{d}), (V, g)) \right| \leq \quad (27)$$

$$\underbrace{\left| \int_{[0,1]^k} \left( \prod_{i \in V(F)} f(x_i)^{d_i} \right) \left( \prod_{\{i,j\} \in E(F)} W(x_i, x_j) - \prod_{\{i,j\} \in E(F)} V(x_i, x_j) \right) \mathrm{d}\lambda^k(\boldsymbol{x}) \right|}_{\textcircled{1}} \quad (28)$$

$$+ \underbrace{\left| \int_{[0,1]^k} \left( \prod_{\{i,j\} \in E(F)} V(x_i, x_j) \right) \left( \prod_{i \in V(F)} f(x_i)^{d_i} - \prod_{i \in V(F)} g(x_i)^{d_i} \right) \mathrm{d}\lambda^k(\boldsymbol{x}) \right|}_{\textcircled{2}}. \quad (29)$$

For term $\textcircled{1}$, we set $D := \sum_i d_i$ and observe that for all $\boldsymbol{x} \in [0,1]^k$

$$\frac{1}{r^D} \prod_{i \in V(F)} f(x_i)^{d_i} \in [-1, 1], \quad (30)$$

and hence similarly to the proof of the classical counting lemma (see, e.g., Zhao (2023)) we bound

$$\textcircled{1} \leq r^D e(F) \left\| W - V \right\|_{\square,2} \leq 4r^D e(F) \left\| W - V \right\|_{\square}. \quad (31)$$

In comparison to the standard proof, the usage of $\left\| \cdot \right\|_{\square,2}$, an alternative definition of the cut norm, stems from the fact that function values appearing in the integral in $\textcircled{1}$ (renormalizing by $r^D$) are not necessarily in $[0, 1]$, but $[-1, 1]$. See also equations (4.3), (4.4) in Janson (2013). For $\textcircled{2}$, we bound the $L^1$ difference of the terms involving $f$ and $g$:

$$\left| \int_{[0,1]^k} \left( \prod_{\{i,j\} \in E(F)} V(x_i, x_j) \right) \left( \prod_{i \in V(F)} f(x_i)^{d_i} - \prod_{i \in V(F)} g(x_i)^{d_i} \right) \mathrm{d}\lambda^k(\boldsymbol{x}) \right| \quad (32)$$

$$\leq \int_{[0,1]^k} \left| \prod_{\{i,j\} \in E(F)} V(x_i, x_j) \right| \left| \prod_{i \in V(F)} f(x_i)^{d_i} - \prod_{i \in V(F)} g(x_i)^{d_i} \right| \mathrm{d}\lambda^k(\boldsymbol{x}) \quad (33)$$

$$\leq \sum_{i \in V(F)} \int_{[0,1]^k} \left| f(x_i)^{d_i} - g(x_i)^{d_i} \right| \left| \prod_{j < i} f(x_j)^{d_j} \prod_{j > i} g(x_j)^{d_j} \right| \mathrm{d}\lambda^k(\boldsymbol{x}) \quad (34)$$

$$\leq \sum_{i \in V(F)} r^{\sum_{j \neq i} d_i} \int_{[0,1]} \left| f(x_i)^{d_i} - g(x_i)^{d_i} \right| \mathrm{d}\lambda(x) \quad (35)$$

$$\overset{(*)}{\leq} \sum_{i \in V(F)} r^{\sum_{j \neq i} d_i} \cdot d_i r^{d_i - 1} \left\| f - g \right\|_1 = Dr^{D-1} \left\| f - g \right\|_1 \leq 2Dr^{D-1} \left\| f - g \right\|_{\square}, \quad (36)$$

where $(*)$ uses $\|f\|_\infty, \|g\|_\infty \le r$ and hence the Lipschitz constant of $x \mapsto x^{d_i}$ is bounded by the maximum of its derivative $d_i r^{d_i-1}$, and the last inequality uses $\|\cdot\|_1 \le 2 \|\cdot\|_\square$ in one dimension. Combining the two bounds for ① from (31) and ② from (36), we obtain

$$\left| t((F, \boldsymbol{d}), (W, f)) - t((F, \boldsymbol{d}), (V, g)) \right| \le 4r^D e(F) \|W - V\|_\square + 2Dr^{D-1} \|f - g\|_\square, \quad (37)$$

which yields the claim. $\qquad\square$

## C.2    Smooth and Invariant Norms for Graphon-Signals

In this work, we consider not only the *cut norm*, but also $L^p$ norms (and distances) of graphon-signals. The purpose of this section is to show that all of the derived unlabeled distances we consider on the graphon-signal space yield the same notion of weak isomorphism, i.e., vanish simultaneously. This can be shown for *smooth invariant norms* on the graphon-signal space (cf. Lovász (2012, § 8.2.5)):

**Definition C.2** (Smooth and invariant norms). *Two norms $N = (N_1, N_2)$, where $N_1$ is a norm on $L^\infty[0,1]$ and $N_2$ on $\mathcal{W}$, are called **smooth** if the two conditions*

*(1) $W_n \to W \in \mathcal{W}$, $f_n \to f \in L^\infty[0,1]$ almost everywhere,*
*(2) $\sup_{n\in\mathbb{N}} \|W_n\|_\infty \le \infty$, $\sup_{n\in\mathbb{N}} \|f_n\|_\infty \le \infty$,*

*imply that*

$$N_1(f_n) \to N_1(f), \qquad N_2(W_n) \to N_2(W). \quad (38)$$

*They are **invariant** if*

$$N_1(f^\varphi) = N_1(f), \quad N_2(W^\varphi) = N_2(W) \qquad \forall \varphi \in S_{[0,1]}, \quad (39)$$

*where $(W, f) \in \mathcal{W} \times L^\infty[0,1]$ and $f^\varphi(x) := f(\varphi(x))$ for $x \in [0,1]$. We may sometimes also write $(W, f)^\varphi := (W^\varphi, f^\varphi)$.*

The conditions clearly apply to $\|\cdot\|_\square$ and $\|\cdot\|_p$ for $p \in [1, \infty)$ (acting either as $N_1$ and $N_2$), but do *not* hold for $p = \infty$ (take for example $W_n = \mathbb{1}_{[0,1/n]^2}$, $f_n = \mathbb{1}_{[0,1/n]}$). For any smooth and invariant $N$, we can obtain a derived unlabeled distance as done for the cut distance:

**Definition C.3** (Derived unlabeled distance). *Let $N = (N_1, N_2)$ be smooth and invariant norms. Define its **derived unlabeled distance** on the graphon-signal space as*

$$\delta_N((W, f), (V, g)) := \inf_{\varphi \in S_{[0,1]}} \big( N_2(W - V^\varphi) + N_1(f - g^\varphi) \big). \quad (40)$$

Just as in the standard graphon case, for *all* such smooth and invariant norms, the infimum in (40) is attained when minimizing over *all* measure preserving functions. This turns out to be a generalization of Lovász (2012, Theorem 8.13):

**Theorem C.4** (Minima vs. infima for smooth invariant norms). *Let $N$ be a smooth invariant norm on $\mathcal{W}$ and $L^\infty[0,1]$. Then, we have the following alternate expressions for $\delta_N$:*

$$\delta_N((W, f), (V, g)) = \inf_{\varphi \in \overline{S}_{[0,1]}} \big( N_2(W - V^\varphi) + N_1(f - g^\varphi) \big) \quad (41)$$

$$= \min_{\varphi, \psi \in \overline{S}_{[0,1]}} \big( N_2(W^\varphi - V^\psi) + N_1(f^\varphi - g^\psi) \big). \quad (42)$$

*Sketch of proof.* We follow the proof of Theorem 8.13 by Lovász (2012), briefly highlighting the necessary adjustments to the argument. To establish the first equality, approximations by step graphons that converge a.e. are considered, and the crucial point is that any $\varphi \in \overline{S}_{[0,1]}$ can be realized by a suitable $\widetilde{\varphi} \in S_{[0,1]}$ for such step graphons. For graphon-signals, the argument can be transferred if one simply considers partitions respecting each step graphon and step signal simultaneously when constructing the corresponding $\widetilde{\varphi} \in S_{[0,1]}$. For the second equality, which is proven in greater generality with coupling measures over $[0,1]^2$ by Lovász (2012), note that the lower semi-continuity in (8.24) is just shown for kernels (i.e., $L^\infty[0,1]^2$), but the argument extends verbatim to $L^\infty[0,1]$, and the sum of two lower semicontinuous functions is still lower semicontinuous. The rest of the argument applies without modification. $\qquad\square$

Note that our definition of $\delta_\square$ coincides with the one by Levie (2023), as they also considered measure preserving bijections of co-null sets in $[0, 1]$ (writing $S'_{[0,1]}$ for this set). As an immediate corollary, we can obtain a simple characterization of weak isomorphism for graphon-signals:

**Corollary C.5** (Weak isomorphism). *Two graphon-signals $(W, f), (V, g) \in \mathcal{WS}_r$ are **weakly isomorphic**, i.e., $\delta_N((W, f), (V, g)) = 0$ for any smooth and invariant norm $N = (N_1, N_2)$, if and only if there are $\varphi, \psi \in \overline{S}_{[0,1]}$ such that $W^\varphi = V^\psi$ and $f^\varphi = g^\psi$ almost surely.*

Here, Theorem C.4 ensures that this does not depend on the specific $N$. Similar to standard graphons, we identify weakly isomorphic graphon-signals to obtain the space of *unlabeled* graphon-signals $\widetilde{\mathcal{WS}}_r$. We mainly work with the cut distance $\delta_\square$ as well as the $L^p$ distances $\delta_p$ for $p \in [1, \infty)$ (where we choose $N = (\|\cdot\|_p, \|\cdot\|_p)$). Among these, of special interest are typically $\delta_1$ as this corresponds to the *edit distance* on graphs, as well as $\delta_2$, which gives rise to a geodesic space in the standard graphon case (Oh et al., 2024).

## C.3 PROOF OF THEOREM 3.2

**Theorem 3.2** (Characterizations of weak isomorphism for graphon-signals). *Fix $r > 1$ and let $(W, f), (V, g) \in \mathcal{WS}_r$. Then, the following statements are equivalent:*

*(1) $\delta_p((W, f), (V, g)) = 0$ for any $p \in [1, \infty)$;*
*(2) $\delta_\square((W, f), (V, g)) = 0$;*
*(3) $t((F, \boldsymbol{d}), (W, f)) = t((F, \boldsymbol{d}), (V, g))$ for all multigraphs $F$, $\boldsymbol{d} \in \mathbb{N}_0^{v(F)}$;*
*(4) $t((F, \boldsymbol{d}), (W, f)) = t((F, \boldsymbol{d}), (V, g))$ for all simple graphs $F$, $\boldsymbol{d} \in \mathbb{N}_0^{v(F)}$;*
*(5) $\mathbb{H}_k(W, f) \stackrel{\mathcal{D}}{=} \mathbb{H}_k(V, g)$ for all $k \in \mathbb{N}$;*
*(6) $\mathbb{G}_k(W, f) \stackrel{\mathcal{D}}{=} \mathbb{G}_k(V, g)$ for all $k \in \mathbb{N}$.*

The following elementary lemma will be useful on several occasions when comparing two probability distributions.

**Lemma C.6** (Moments of Random Vectors). *Let $m \in \mathbb{N}$ and let $\boldsymbol{X}, \boldsymbol{Y}$ be $m$-dimensional random vectors which are almost surely bounded, i.e., there is some $R > 0$ such that $\mathbb{P}(\|\boldsymbol{X}\| \leq R) = \mathbb{P}(\|\boldsymbol{Y}\| \leq R) = 1$, for any norm $\|\cdot\|$ on $\mathbb{R}^m$. Suppose that for all $\boldsymbol{d} \in \mathbb{N}_0^m$ the moments of $\boldsymbol{X}, \boldsymbol{Y}$ agree:*

$$\mathbb{E}\left[\prod_{i=1}^m X_i^{d_i}\right] = \mathbb{E}\left[\prod_{i=1}^m Y_i^{d_i}\right]. \tag{43}$$

*Then, $\boldsymbol{X}$ and $\boldsymbol{Y}$ are identically distributed, i.e., $\boldsymbol{X} \stackrel{\mathcal{D}}{=} \boldsymbol{Y}$.*

*Proof.* In the case of more general random variables/vectors, this is known in the literature as the *moment problem* (see, e.g., Schmüdgen (2017)). Under boundedness, however, this is trivial and can for example be proven via the characteristic functions of the random vectors. $\square$

We are now ready to prove Theorem 3.2. To this end, we will show the implications in Figure 4.

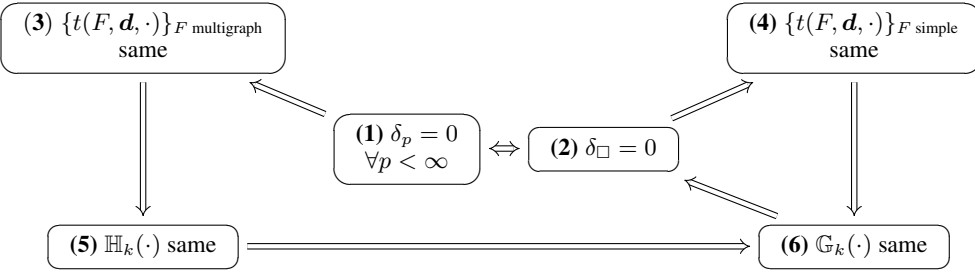

Figure 4: Equivalence chain for the proof of Theorem 3.2.

*Proof of Theorem 3.2.*

**(1)** $\Leftrightarrow$ **(2):** Theorem C.4 implies that for any $p \in [1, \infty)$

$$\delta_\square((W, f), (V, g)) = 0 \Leftrightarrow \exists \varphi, \psi \in \overline{S}_{[0,1]} : (W, f)^\varphi = (V, g)^\psi \Leftrightarrow \delta_p((W, f), (V, g)) = 0, \quad (44)$$

where equality in the middle holds in an $\lambda^2$-a.e. sense.

**(2)** $\Rightarrow$ **(4):** This follows immediately from the graphon-signal counting lemma (§ C.1).

**(4)** $\Rightarrow$ **(6):** Let $(W, f), (V, g) \in \mathcal{WS}_r$ such that $t((F, \boldsymbol{d}), (W, f)) = t((F, \boldsymbol{d}), (V, g))$ for all simple graphs $F, \boldsymbol{d} \in \mathbb{N}_0^{v(F)}$. Fix some $k \in \mathbb{N}$. Clearly, the distribution of $\mathbb{G}_k(W, f)$ is uniquely determined by

$$\mathbb{P}_{(G, \boldsymbol{f}) \sim \mathbb{G}_k(W, f)}(G \cong F), \quad (45)$$

$$\mathbb{P}_{(G, \boldsymbol{f}) \sim \mathbb{G}_k(W, f)}(\boldsymbol{f} \in \cdot | G \cong F), \quad (46)$$

i.e., the discrete distribution of the (labeled) random graph $G$ and the conditional distribution of the node features given the graph structure, for every simple graph $F$ of size $k$. For (45), we remark that the standard homomorphism densities w.r.t. just the graphons $W, V$ can be recovered by taking $\boldsymbol{d} = 0$. Thus, the inclusion-exclusion argument from the proof of Theorem 4.9.1 in (Zhao, 2023) can be used verbatim to reconstruct the probabilities from (45). With a similar inclusion-exclusion argument, we see that for any $F$

$$\mathbb{1}\{\mathbb{G}_k(W) \cong F\} = \sum_{F' \supseteq F} (-1)^{e(F') - e(F)} \mathbb{1}\{\mathbb{G}_k(W) \supseteq F'\} \quad (47)$$

and therefore

$$\mathbb{E}_{(G, \boldsymbol{f}) \sim \mathbb{G}_k(W, f)} \left[ \prod_{i \in V(F)} f_i^{d_i} \Big| G \cong F \right] = \frac{\sum_{F' \supseteq F} (-1)^{e(F') - e(F)} t(F', \boldsymbol{d}, (W, f))}{\mathbb{P}_{(G, \boldsymbol{f}) \sim \mathbb{G}_k(W, f)}(G \cong F)} \quad (48)$$

as long as the denominator is positive (otherwise, the corresponding conditional distribution is arbitrary). Since $(\boldsymbol{f}|G \cong F)$ is a bounded random vector ($\|\boldsymbol{f}\|_\infty \leq r$ a.s.), its distribution is uniquely determined by its multidimensional moments, i.e., precisely the expressions from (48) (see Lemma C.6). Thus, we can conclude $\mathbb{G}_k(W, f) \overset{\mathcal{D}}{=} \mathbb{G}_k(V, g)$ for all $k \in \mathbb{N}$.

**(6)** $\Rightarrow$ **(2):** This implication follows from applying the graphon-signal sampling lemma (Levie (2023, Theorem 3.7) and Theorem B.5): If **(6)** holds, we can bound

$$\delta_\square((W, f), (V, g)) \leq \mathbb{E}\left[\delta_\square((W, f), \mathbb{G}_k(W, f))\right] + \mathbb{E}\left[\delta_\square((V, g), \mathbb{G}_k(W, f))\right] \quad (49)$$

$$= \mathbb{E}\left[\delta_\square((W, f), \mathbb{G}_k(W, f))\right] + \mathbb{E}\left[\delta_\square((V, g), \mathbb{G}_k(V, g))\right] \to 0 \quad (50)$$

as $k \to \infty$.

**(1)** $\Rightarrow$ **(3):** With a technique as in § C.1, bounding the individual graphon terms in a similar way as the signal terms, it is straightforward to show that the signal-weighted homomorphism density $t((F, \boldsymbol{d}), \cdot)$ from any *multigraph* $F$ is also Lipschitz continuous w.r.t. $\delta_1$. Thus, statement **(3)** follows immediately.

**(3)** $\Rightarrow$ **(5):** Let $(W, f), (V, g) \in \mathcal{WS}_r$ such that $t((F, \boldsymbol{d}), (W, f)) = t((F, \boldsymbol{d}), (V, g))$ for all multigraphs $F, \boldsymbol{d} \in \mathbb{N}_0^{v(F)}$. Fix $k \in \mathbb{N}$. Then, $\mathbb{H}_k(W, f)$ and $\mathbb{H}_k(V, g)$ can be seen as $(k^2 + k)$-dimensional random vectors which are clearly bounded, since all graphon entries are in $[0, 1]$ and all signal entries in $[-r, r]$. We observe that $\{t((F, \boldsymbol{d}), (W, f))\}_{F, \boldsymbol{d}}$ and $\{t((F, \boldsymbol{d}), (V, g))\}_{F, \boldsymbol{d}}$, with $F$ ranging over multigraphs of size $k$, are precisely the multidimensional moments of these random vectors. Lemma C.6 yields statement **(5)**.

**(5)** $\Rightarrow$ **(6):** This is immediate, as $\mathbb{G}_k(\cdot)$ is a function of $\mathbb{H}_k(\cdot)$. $\qquad \square$

## C.4 PROOF OF COROLLARY 3.3

**Corollary 3.3** (Convergence in graphon-signal space). *For $(W_n, f_n)_n, (W, f) \in \mathcal{WS}_r$ and $r > 1$,*

$$\delta_\square((W_n, f_n), (W, f)) \to 0 \quad \Leftrightarrow \quad t((F, \boldsymbol{d}), (W_n, f_n)) \to t((F, \boldsymbol{d}), (W, f)) \ \forall F, \boldsymbol{d} \in \mathbb{N}_0^{v(F)} \quad (6)$$

*as $n \to \infty$, with $F$ ranging over all simple graphs.*

*Proof.* The proof idea is essentially the same as in the classical graphon case (for example, see § 4.9 in Zhao (2023)): An application of Theorem 3.2 that uses compactness of the graphon-signal space. For the sake of completeness, we restate the argument.

"$\Rightarrow$" follows immediately from the counting lemma (§ C.1). For "$\Leftarrow$", let $(W_n, f_n)_n$ be a sequence of graphon-signals that left-converges to $(W, f) \in \mathcal{WS}_r$. By compactness (Levie, 2023, Theorem 3.6), there exists a subsequence $(W_{n_i}, f_{n_i})_i$ converging to some limit $(V, g)$ in cut distance. But then also all signal-weighted homomorphism densities of the subsequence converge, and hence

$$t(F, \boldsymbol{d}, (W, f)) = t(F, \boldsymbol{d}, (V, g)) \quad \forall F, \boldsymbol{d} \in \mathbb{N}_0^{v(F)}. \tag{51}$$

Theorem 3.2 yields $\delta_\square((W, f), (V, g)) = 0$, i.e., also $(W_n, f_n) \to (W, f)$ in cut distance. $\qquad\square$

# D $k$-WL FOR GRAPHON-SIGNALS

In this section, we demonstrate that signal-weighted homomorphism densities also make sense on the granularity level of the $k$-WL hierarchy. Specifically, we show that their indistinguishability aligns with a natural formulation of the $k$-WL test for graphon-signals:

**Theorem 3.4** ($k$-WL for graphon-signals, informal). *Two graphon-signals $(W, f)$ and $(V, g)$ are $k$-WL indistinguishable if and only if $t((F, \boldsymbol{d}), (W, f)) = t((F, \boldsymbol{d}), (V, g))$ for all multigraphs $F$ of treewidth $\leq k - 1$, $\boldsymbol{d} \in \mathbb{N}_0^{v(F)}$.*

Recently, Böker (2023) established a characterization of the $k$-WL test for graphons, linking it to *multigraph* homomorphism densities w.r.t. patterns of bounded treewidth. Building on this, we introduce a natural generalization of the $k$**-WL measure** and **distribution** by Böker (2023) to graphon-signals and highlight that their homomorphism expressivity can be captured through signal-weighted homomorphism densities. This serves as the final step in demonstrating the suitability of this extension for our analyses. Since, compared to the discrete case, the formulation for graphons and graphon-signals is significantly more technical, we directly state the graphon-signal versions of the concepts. Our goal in this section is not a fully detailed exposition, but rather to give intuition and highlight the key modifications required in the definitions and arguments of Böker (2023) to make them work for graphon-signals.

## D.1 TRI-LABELED GRAPHS AND GRAPHON-SIGNAL OPERATORS

To relate the $k$-WL measure to homomorphism densities on standard graphons, Böker (2023, § 3.1) introduces *bi-labeled graphs* to define a set of operators that generalize the single shift operator $T_W$ used by Grebík & Rocha (2022) for 1-WL. Using *bi-labeled graphs*, they define operations that generate all multigraphs of treewidth $\leq k - 1$ by taking the closure of a set of *atomic* graphs under these operations. Intuitively, decomposing a bi-labeled graph into atomic parts corresponds to a tree decomposition with a certain special structure. This effectively allows one to express homomorphism densities via elementary operators associated with atomic graphs. This concept can be generalized to *tri-labeled graphs* for signal-weighted homomorphism densities:

**Definition D.1** (Tri-labeled graphs). *A **tri-labeled graph** is a tuple $\boldsymbol{G} = (G, \boldsymbol{a}, \boldsymbol{b}, \boldsymbol{d})$, where $G$ is a multigraph, $\boldsymbol{a} \in V(G)^k$, $\boldsymbol{b} \in V(G)^\ell$ with $k, \ell \geq 0$, and $\boldsymbol{d} \in \mathbb{N}_0^{v(G)}$. Here, the entries of $\boldsymbol{a}, \boldsymbol{b}$ each must be pairwise distinct. $(G, \boldsymbol{d})$ is called the **underlying graph** of $\boldsymbol{G}$, and $\boldsymbol{a}, \boldsymbol{b}$ its **input** and **output vertices**. Write $\mathcal{M}^{k,\ell}$ for the set of all such graphs.*

One can define some elementary operations with tri-labeled graphs: For $\boldsymbol{F}_1 = (F_1, \boldsymbol{a}_1, \boldsymbol{b}_1, \boldsymbol{d}_1) \in \mathcal{M}^{k,\ell}$ and $\boldsymbol{F}_2 = (F_2, \boldsymbol{a}_2, \boldsymbol{b}_2, \boldsymbol{d}_2) \in \mathcal{M}^{\ell,m}$, define the **composition** of $\boldsymbol{F}_1$ and $\boldsymbol{F}_2$ as

$$\boldsymbol{F}_1 \circ \boldsymbol{F}_2 := (F, \boldsymbol{a}_1, \boldsymbol{b}_2, \boldsymbol{d}_1 + \boldsymbol{d}_2) \in \mathcal{M}^{k,m}, \tag{52}$$

where $F$ is obtained from $F_1 \sqcup F_2$ by identifying the vertices in the tuples $\boldsymbol{b}_1$ and $\boldsymbol{a}_2$ (whose dimensions we required to be aligned), and the sum $\boldsymbol{d}_1 + \boldsymbol{d}_2$ should be seen w.r.t. this identification. This identification also potentially introduces multiedges. The **Schur product** of $\boldsymbol{F}_1 = (F_1, \boldsymbol{a}_1, \varnothing, \boldsymbol{d}_1)$ and $\boldsymbol{F}_2 = (F_2, \boldsymbol{a}_2, \varnothing, \boldsymbol{d}_2) \in \mathcal{M}^{k,0}$ is defined as

$$\boldsymbol{F}_1 \cdot \boldsymbol{F}_2 := (F, \boldsymbol{a}_1, \varnothing, \boldsymbol{d}_1 + \boldsymbol{d}_2) \in \mathcal{M}^{k,0}, \tag{53}$$

where we abuse notation by writing $\varnothing$ for the empty tuple, and $F = F_1 \sqcup F_2$, with the nodes in $\boldsymbol{a}_1$ and $\boldsymbol{a}_2$ identified. Now, also consider some *atomic* tri-labeled graphs which can be glued together using the above operations:

**Definition D.2** (cf. Böker (2023, Definition 12)). *Let $k \geq 1$. Define the following specific tri-labeled graphs:*

- ***Adjacency graph:*** $\boldsymbol{A}_{\{i,j\}}^{(k)} := (([k], \{i, j\}), (1, \ldots, k), (1, \ldots, k), \boldsymbol{0}) \in \mathcal{M}^{k,k}$ *for $i \neq j$,*
- ***Introduce graph:*** $\boldsymbol{I}_j^{(k)} := (([k], \varnothing), (1, \ldots, k), (1, \ldots, j-1, j+1, \ldots, k), \boldsymbol{0}) \in \mathcal{M}^{k,k-1}$,
- ***Forget graph:*** $\boldsymbol{F}_j^{(k)} := (([k], \varnothing), (1, \ldots, j-1, j+1, \ldots, k), (1, \ldots, k), \boldsymbol{0}) \in \mathcal{M}^{k-1,k}$,
- ***Neighbor graph:*** $\boldsymbol{N}_j^{(k)} := \boldsymbol{I}_j^{(k)} \circ \boldsymbol{F}_j^{(k)} \in \mathcal{M}^{k,k}$,

- **Signal graph:** $S_d^{(k)} := (([k], \varnothing), (1, \ldots, k), (1, \ldots, k), d) \in \mathcal{M}^{k,k}$ *for* $d \in \mathbb{N}_0^k$,
- **All-one graph:** $\mathbf{1}^{(k)} := (([k], \varnothing), (1, \ldots, k), \varnothing, \mathbf{0}) \in \mathcal{M}^{k,0}$.

*Define the set of all **atomic tri-labeled graphs** as*

$$\mathcal{F}^k := \left\{ A_{\{i,j\}}^{(k)} \,\middle|\, i \neq j \in [k] \right\} \cup \left\{ N_j^{(k)} \,\middle|\, j \in [k] \right\} \cup \left\{ S_d^{(k)} \,\middle|\, d \in \mathbb{N}_0^k \right\} \subset \mathcal{M}^{k,k}. \quad (54)$$

These atomic graphs can now be joined by the elementary operations composition and Schur product to *terms* (Böker, 2023, Definition 13): For $k \geq 1$ and $\mathcal{F} \subseteq \mathcal{M}^{k,k}$, the set $\langle \mathcal{F} \rangle$ of $\mathcal{F}$-**terms** is the smallest set of expressions such that

$$\mathbf{1}^{(k)} \in \langle \mathcal{F} \rangle, \quad (55)$$

$$F \circ \mathbb{F}_1 \in \langle \mathcal{F} \rangle \qquad\qquad \forall F \in \mathcal{F}, \mathbb{F} \in \langle \mathcal{F} \rangle, \quad (56)$$

$$\mathbb{F}_1 \cdot \mathbb{F}_2 \in \langle \mathcal{F} \rangle \qquad\qquad \forall \mathbb{F}_1, \mathbb{F}_2 \in \langle \mathcal{F} \rangle. \quad (57)$$

For such a term $\mathbb{F}$, write $[\![\mathbb{F}]\!] \in \mathcal{M}^{k,0}$ for its evaluation. Notably, Böker (2023, Definition 13) shows that the underlying graphs when evaluating the terms in their version of $\langle \mathcal{F}^k \rangle$ are precisely the multigraphs of treewidth bounded by $k - 1$. In our case, we obtain

**Theorem D.3** (cf. Böker (2023, Theorem 14)). *The underlying graphs of the tri-labeled graphs obtained by evaluating the terms in $\langle \mathcal{F}^k \rangle$ are precisely the node-featured multigraphs $(F, d)$ with* $\mathrm{tw}(F) \leq k - 1$, *up to isolated vertices.*

We omit the proof here; it is elementary and relies on the existence of certain *nice tree decompositions* (Kloks, 1994; Böker, 2023). In these (rooted) tree decompositions, roughly speaking, each edge where the bag changes corresponds to the addition or removal of a single node in a bag, and each bag has at most two children. Consequently, the tree can be traversed so that every edge is associated with one of the operations from Definition D.2.

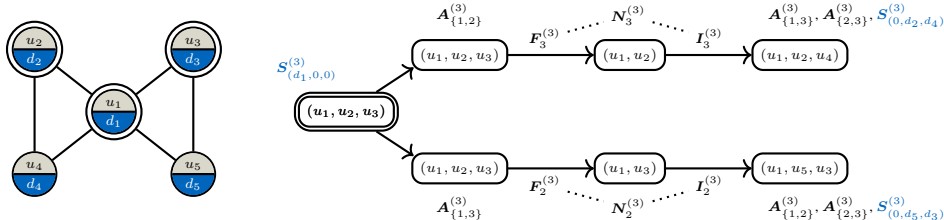

Figure 5: A *nice* tree decomposition for a node-featured graph of treewidth 2, along with corresponding atomic tri-labeled graphs.

Consider, for example, the tri-labeled graph $\mathbf{F} = (F, a, \varnothing, d) \in \mathcal{M}^{3,0}$ from Figure 5 (the input vertices are highlighted), as well as a corresponding nice tree decomposition. $\mathbf{F}$ can be seen as the evaluation of the following term in $\mathcal{F}^3$:

$$\mathbf{F} = S_{(d_1,0,0)}^{(3)} \circ \left( \left( A_{\{1,2\}}^3 \circ N_3^{(3)} \circ A_{\{1,3\}}^{(3)} \circ A_{\{2,3\}}^{(3)} \circ S_{(0,d_2,d_4)}^{(3)} \circ \mathbf{1}^{(k)} \right) \right. \quad (58)$$

$$\left. \cdot \left( A_{\{1,3\}}^3 \circ N_2^{(3)} \circ A_{\{1,2\}}^{(3)} \circ A_{\{2,3\}}^{(3)} \circ S_{(0,d_5,d_3)}^{(3)} \circ \mathbf{1}^{(k)} \right) \right). \quad (59)$$

To later connect these terms in $\langle \mathcal{F}^k \rangle$ to the $k$-WL test, Böker (2023) defines the concept of *height* $h(\mathbb{F})$ for a term $\mathbb{F}$. Intuitively, this means that the homomorphism density w.r.t. the underlying graph of $[\![\mathbb{F}]\!]$ can be computed after $h(\mathbb{F})$ rounds of color refinement.

**Definition D.4** (Height of a term). *We define the **height** $h(\mathbb{F})$ of a term $\mathbb{F} \in \langle \mathcal{F}^k \rangle$ inductively by setting $h(\mathbf{1}^{(k)}) := 0$, $h(N \circ \mathbb{F}) := h(\mathbb{F}) + 1$ for any neighbor graph $N$, $h(A \circ \mathbb{F}) := h(\mathbb{F})$ and $h(S \circ \mathbb{F}) := h(\mathbb{F})$ for any adjacency and signal graphs $A$ and $S$ respectively, and $h(\mathbb{F}_1 \cdot \mathbb{F}_2) := \max\{h(\mathbb{F}_1), h(\mathbb{F}_2)\}$.*

We can define **graphon-signal operators** using tri-labeled graphs. Let $\mathbf{F} = (F, a, b, d) \in \mathcal{M}^{k,\ell}$ be a tri-labeled graph. Let $r > 0$. Define the associated $\mathbf{F}$-operator of $(W, f) \in \mathcal{WS}_r$ as $T_{\mathbf{F} \to (W,f)}$ :

$L^2[0,1]^\ell \to L^2[0,1]^k$, which acts on $U \in L^2[0,1]^\ell$ as

$$\left(T_{\boldsymbol{F} \to (W,f)} U\right)(\boldsymbol{x_a})$$

$$:= \int_{[0,1]^{v(F)-k}} \left(\prod_{i \in V(F)} f(x_i)^{d_i}\right)\left(\prod_{\{i,j\} \in E(F)} W(x_i, x_j)\right) U(\boldsymbol{x_b}) \, \mathrm{d}\lambda^{v(F)-k}(\boldsymbol{x}_{V(F) \setminus \boldsymbol{a}}). \quad (60)$$

Importantly, composition (52) and Schur product (53) of tri-labeled graphs correspond to the composition and pointwise product of their associated operators, i.e.,

$$T_{\boldsymbol{F_1} \circ \boldsymbol{F_2} \to (W,f)} = T_{\boldsymbol{F_1} \to (W,f)} \circ T_{\boldsymbol{F_2} \to (W,f)} \quad \forall \boldsymbol{F_1} \in \mathcal{M}^{k,m}, \ \boldsymbol{F_2} \in \mathcal{M}^{m,\ell}, \ k,l,m \geq 0, \quad (61)$$

$$T_{\boldsymbol{F_1} \cdot \boldsymbol{F_2} \to (W,f)} = T_{\boldsymbol{F_1} \to (W,f)} \cdot T_{\boldsymbol{F_2} \to (W,f)} \quad \forall \boldsymbol{F_1}, \boldsymbol{F_2} \in \mathcal{M}^{k,0}, \ k \geq 0. \quad (62)$$

This allows us to gradually build up the operator $T_{[\![\mathbb{F}]\!] \to (W,f)}$ for any term $\mathbb{F} \in \langle \mathcal{F}^k \rangle$ from the operators w.r.t. its atomic graphs. It is also straightforward to see that just integrating over the output of such an operator applied to a constant function recovers the signal-weighted homomorphism densities. Formally, for any $(W,f) \in \mathcal{WS}_r$ and $\boldsymbol{F} = (F, \boldsymbol{a}, \boldsymbol{b}, \boldsymbol{d}) \in \mathcal{M}^{k,\ell}$, we have

$$\int_{[0,1]^k} T_{\boldsymbol{F} \to (W,f)} \mathbb{1}_{[0,1]^\ell} \, \mathrm{d}\lambda^k = t((F, \boldsymbol{d}), (W, f)). \quad (63)$$

## D.2 WEISFEILER-LEMAN MEASURES

One can now define the space of colors used by the $k$-WL test for graphon-signals. In the discrete case—at least as long as node labels come from a countable alphabet—this space is countable and can be left implicit. However, for graphons and graphon-signals, the situation is more nuanced as the color space requires careful consideration of its topology, i.e., a suitable notion of convergence/distance.

**Definition D.5** (Weisfeiler-Leman Measure, cf. Böker (2023, Definition 26)). *Let $k \geq 1$ and fix $r > 0$. Let*

$$P_0^k := [0,1]^{\binom{[k]}{2}} \times [-r, r]^k \quad (64)$$

*be the space of initial colors. Inductively define*

$$\mathbb{M}_s^k := \prod_{i \leq s} P_i^k, \qquad P_{s+1}^k := \mathcal{P}(\mathbb{M}_s^k)^k \quad (65)$$

*for every $s \in \mathbb{N}$, where we consider $P_0^k$ with its standard topology, and $\mathcal{P}(\cdot)$ denotes the set of all Borel probability measures, equipped with the weak topology. Let $\mathbb{M}^k := \prod_{s \in \mathbb{N}} P_s^k$ (equipped with the product topology) and define $p_{t \to s} : \mathbb{M}_t^k \to \mathbb{M}_s^k$ to be the natural projection for $s \leq t$. Set*

$$\mathbb{P}^k := \left\{\alpha \in \mathbb{M}^k \mid (\alpha_{s+1})_j = (p_{s+1 \to s})_*(\alpha_{s+2})_j \text{ for all } j \in [k], s \in \mathbb{N}\right\}, \quad (66)$$

*where $(p_{s \to s+1})_*$ denotes the pushforward of measures.*

We tacitly equip all spaces in Definition D.5 with their Borel $\sigma$-algebra. Intuitively, one can think of $\mathbb{P}^k$ as the space of all colors (potentially) used by the $k$-WL test, where for $\alpha \in \mathbb{P}^k$ each entry $\alpha_s$, $s \in \mathbb{N}$, is a *refinement* of the previous one. For any $\alpha \in \mathbb{P}^k$ and $j \in [k]$, there is a unique measure $\mu_j^\alpha \in \mathcal{P}(\mathbb{M}^k)$ such that

$$(p_{\infty \to s})_* \mu_j^\alpha = (\alpha_{s+1})_j, \qquad \forall s \in \mathbb{N}, \quad (67)$$

by the Kolmogorov consistency theorem. A few basic statements about the topology of the spaces $\mathbb{M}^k$ and $\mathbb{P}^k$ are collected in the following lemma:

**Lemma D.6** (cf. Böker (2023, p. 32 and Lemma 27)). *Consider $\mathbb{M}^k$ and $\mathbb{P}^k$ as in Definition D.5. The following holds:*

*(1) The space $\mathbb{M}^k$ is metrizable and compact, and $\mathbb{P}^k \subseteq \mathbb{M}^k$ is closed.*
*(2) The set*

$$\bigcup_{s \in \mathbb{N}_0} C(\mathbb{M}_s^k) \circ p_{\infty \to s} \subset C(\mathbb{M}^k) \quad (68)$$

*is dense.*

*(3) For any $j \in [k]$, the assignment $\mathbb{P}^k \ni \alpha \mapsto \mu_j^\alpha \in \mathcal{P}(\mathbb{M}^k)$ is continuous.*

*Proof.* Statement **(1)** follows immediately: $P_0^k$ is clearly metrizable and compact, and we can inductively conclude the same for $P_s^k$, $s \in \mathbb{N}$, as this is preserved by $\mathcal{P}(\cdot)$ and finite products. Also, as a countable product of metrizable and compact spaces, $\mathbb{M}^k$ is compact (by Tychonoff's theorem) as well as metrizable. Now, let $\alpha^{(n)} \in \mathbb{P}^k$ such that $\alpha^{(n)} \to \alpha \in \mathbb{M}^k$. Clearly, for any $s \in \mathbb{N}$, $j \in [k]$, we have $(\alpha_{s+1}^{(n)})_j \xrightarrow{w} (\alpha_{s+1})_j$ by the definition of product topology, and for any $f \in C(\mathbb{M}_s^k)$

$$\int_{\mathbb{M}_s^k} f \, \mathrm{d}(\alpha_{s+1}^{(n)})_j = \int_{\mathbb{M}_{s+1}^k} f \circ p_{s+1\to s} \, \mathrm{d}(\alpha_{s+2}^{(n)})_j \to \int_{\mathbb{M}_{s+1}^k} f \circ p_{s+1\to s} \, \mathrm{d}(\alpha_{s+2})_j = \int_{\mathbb{M}_s^k} f \, \mathrm{d}(\alpha_{s+1})_j \tag{69}$$

by the definition of the pushforward, and as $\alpha^{(n)} \in \mathbb{P}^k$. This implies also $(\alpha_{s+1}^{(n)})_j \xrightarrow{w} (p_{s+1\to s})_*(\alpha_{s+2})_j$, and, thus, $(\alpha_{s+1})_j = (p_{s+1\to s})_*(\alpha_{s+2})_j$ (as $\mathcal{P}(\mathbb{M}_s^k)$ is metrizable and, hence, Hausdorff). We conclude $\alpha \in \mathbb{P}^k$, and $\mathbb{P}^k$ is closed. Particularly, this means that $\mathbb{P}^k$ is compact as well. For **(2)**, note that the set is an algebra, clearly contains a constant function, and separates points of $\mathbb{M}^k$ (again, by the definition of product topology). The Stone-Weierstrass theorem yields the claim. Statement **(3)** follows similarly easily from **(2)**. $\qquad\square$

One can now define a set of "simple" functions on $\mathbb{M}^k$ which will turn out to be related to homomorphism densities. For a term $\mathbb{F} \in \langle \mathcal{F}^k \rangle$ of height $h(\mathbb{F}) \le s$, define a set of **realizing functions** $F_s^{\mathbb{F}}$ from $\mathbb{M}_s^k$ to $\mathbb{R}$ inductively as the smallest one fulfilling

$$F_s^{\mathbf{1}^{(k)}} \ni \mathbb{1}_{\mathbb{M}_s^k}, \tag{70}$$

$$F_s^{\mathbf{A}_{\{i,j\}}^{(k)} \circ \mathbb{F}} \ni \left[\alpha \mapsto ((\alpha_0)_1)_{\{i,j\}} \cdot t(\alpha)\right] \qquad \forall t \in F_s^{\mathbb{F}}, \, i \neq j \in [k], \tag{71}$$

$$F_s^{\mathbf{S}_{\mathbf{d}}^{(k)} \circ \mathbb{F}} \ni \left[\alpha \mapsto \prod_{i \in [k]} ((\alpha_0)_2)_i^{d_i} \cdot t(\alpha)\right] \qquad \forall t \in F_s^{\mathbb{F}}, \, \mathbf{d} \in \mathbb{N}_0^k, \tag{72}$$

$$F_{s+1}^{\mathbf{N}_j^k \circ \mathbb{F}} \ni \left[\alpha \mapsto \int_{\mathbb{M}_s^k} t \, \mathrm{d}(\alpha_{s+1})_j\right] \qquad \forall t \in F_s^{\mathbb{F}}, \, j \in [k], \tag{73}$$

$$F_s^{\mathbb{F}_1 \cdot \mathbb{F}_2} \ni t_1 \cdot t_2 \qquad \forall t_1 \in F_s^{\mathbb{F}_1}, t_2 \in F_s^{\mathbb{F}_2}, \tag{74}$$

$$F_s^{\mathbb{F}} \ni t \circ p_{s\to t} \qquad \forall t \in F_t^{\mathbb{F}}, t \le s \le h(\mathbb{F}). \tag{75}$$

Also, set

$$F^{\mathbb{F}} := \bigcup_{s \in \mathbb{N}_0} \bigcup_{h(\mathbb{F}) \le s} F_s^{\mathbb{F}} \circ p_{\infty \to s} \subseteq L^\infty(\mathbb{M}^k). \tag{76}$$

One can check that each $t \in F^{\mathbb{F}}$ is continuous. By the consistency requirement on $\mathbb{P}^k$, the restriction of $F^{\mathbb{F}}$ to this subspace collapses to a singleton. Abusing notation, we define this **realizing function** of a term $\mathbb{F} \in \langle \mathcal{F}^k \rangle$ as

$$t^{\mathbb{F}} := F^{\mathbb{F}}\big|_{\mathbb{P}^k} \in C(\mathbb{P}^k). \tag{77}$$

This leads to the following result:

**Theorem D.7.** *The sets*

$$\mathrm{span} \bigcup_{\mathbb{F} \in \langle \mathcal{F}^k \rangle} F^{\mathbb{F}} \subset C(\mathbb{M}^k) \qquad and \qquad \mathrm{span}\{t^{\mathbb{F}} \mid \mathbb{F} \in \mathcal{F}^k\} \subset C(\mathbb{P}^k) \tag{78}$$

*are dense w.r.t. $\|\cdot\|_\infty$.*

The proof for $\mathbb{M}^k$ is tedious but conceptually simple, and relies on an inductive application of the Stone-Weierstrass theorem on $\mathbb{M}_s^k$ for $s \in \mathbb{N}$. Via Lemma D.6, (68), one can then arrive at $\mathbb{M}^k$ (this works, again, by a basic property of the product topology). The reader is referred to Böker (2023, Lemma 44) for the detailed argument. The statement for $\mathbb{P}^k$ follows immediately by its compactness.

### D.3 COLOR REFINEMENT FOR GRAPHON-SIGNALS

One can now define the equivalent of the color refinement for a specific graphon-signal. Observe the close resemblance of the following definition to discrete $k$-WL test (cf. Huang & Villar (2021)).

**Definition D.8** ($k$-WL, cf. Böker (2023, Definition 29))**.** *Let $k \geq 1$, $r > 0$, and let $(W, f) \in \mathcal{WS}_r$. Define $\mathfrak{C}_{(W,f)}^{k\text{-WL},(0)} : [0,1]^k \to \mathbb{M}_0^k$ by*

$$\mathfrak{C}_{(W,f)}^{k\text{-WL},(0)}(\boldsymbol{x}) := \left( \left( W(x_i, x_j) \right)_{\{i,j\} \in \binom{[k]}{2}}, \left( f(x_j) \right)_{j \in [k]} \right) \tag{79}$$

*for every $\boldsymbol{x} \in [0,1]^k$. We then inductively define $\mathfrak{C}_{(W,f)}^{k\text{-WL},(s+1)} \to \mathbb{M}_{s+1}^k$ by*

$$\mathfrak{C}_{(W,f)}^{k\text{-WL},(s)}(\boldsymbol{x}) := \left( \mathfrak{C}_{(W,f)}^{k\text{-WL},(s-1)}(\boldsymbol{x}), \left( \left( \mathfrak{C}_{(W,f)}^{k\text{-WL},(s-1)} \circ \boldsymbol{x}[\cdot/j] \right)_* \lambda \right)_{j \in [k]} \right) \tag{80}$$

*for every $\boldsymbol{x} \in [0,1]^k$, where $\lambda$ is the Lebesgue measure and $\boldsymbol{x}[\cdot/j] := (\dots, x_{j-1}, \cdot, x_{j+1}, \dots) \in [0,1]^k$. Let $\mathfrak{C}_{(W,f)}^{k\text{-WL}} : [0,1]^k \to \mathbb{M}^k$ such that $(\mathfrak{C}_{(W,f)}^{k\text{-WL}})_s = (\mathfrak{C}_{(W,f)}^{k\text{-WL},(s)})_s$ for all $s \in \mathbb{N}$. Call*

$$\nu_{(W,f)}^{k\text{-WL}} := \left( \mathfrak{C}_{(W,f)}^{k\text{-WL}} \right)_* \lambda^k \in \mathcal{P}(\mathbb{M}^k) \tag{81}$$

*the $k$-**dimensional Weisfeiler-Leman distribution** of the graphon-signal $(W, f)$.*

Intuitively, this corresponds with the multiset of colors that is obtained by the $k$-WL test for graphs. Consequentially, we call two graphon-signals $k$-WL-**indistinguishable** if and only if their $k$-WL distributions coincide. It is straightforward to show that $\nu_{(W,f)}^{k\text{-WL}}(\mathbb{P}^k) = 1$, i.e., $k$-WL-distributions of graphon-signals indeed define color refinements in the sense of (66). Skipping all the technical details, we get directly to the homomorphism expressivity of this $k$-WL test. Notably, the realizing functions $t^{\mathbb{F}}$ of terms $\mathbb{F} \in \langle \mathcal{F}^k \rangle$ can be elegantly related to signal-weighted homomorphism densities via the terms from (77):

**Proposition D.9** (cf. Böker (2023, Corollary 43))**.** *Let $k \geq 1$, $(W, f) \in \mathcal{WS}_r$, and $\mathbb{F} \in \langle \mathcal{F}^k \rangle$ with $[\![\mathbb{F}]\!] = (F, (1, \dots, k), \varnothing, \boldsymbol{d}) \in \mathcal{M}^{k,0}$. Then, we have*

$$t((F, \boldsymbol{d}), (W, f)) = \int_{\mathbb{P}^k} t^{\mathbb{F}} \, \mathrm{d}\nu_{(W,f)}^{k\text{-WL}}. \tag{82}$$

This leads to the following result, which generalizes Böker (2023, Theorem 5, (1) ⇔ (2)):

**Theorem D.10** ($k$-WL for graphon-signals, formal)**.** *Let $r > 0$ and $(W, f), (V, g) \in \mathcal{WS}_r$ be two graphon-signals. Then, $\nu_{(W,f)}^{k\text{-WL}} = \nu_{(V,g)}^{k\text{-WL}}$, i.e. the graphon-signals are $k$-WL indistinguishable, if and only if*

$$t((F, \boldsymbol{d}), (W, f)) = t((F, \boldsymbol{d}), (V, g)) \tag{83}$$

*for all multigraphs $F$ of treewidth $\leq k - 1$ and $\boldsymbol{d} \in \mathbb{N}_0^{v(F)}$.*

The arguments of Böker (2023) transfer almost verbatim to this setting. The crux is that in the $k$-WL test, *both* the distribution of the graphon values $W$ and the signal $f$ are captured already in the first step (79) and, therefore, higher-order moments need to be considered.

# E    INVARIANT GRAPHON NETWORKS

## E.1    PROOF OF THEOREM 4.2

**Theorem 4.2.** *Let $k, \ell \in \mathbb{N}_0$. Then, $\mathsf{LE}_{k \to \ell}$ is a finite-dimensional vector space of dimension*

$$\dim \mathsf{LE}_{k \to \ell} = \sum_{s=0}^{\min\{k,\ell\}} s! \binom{k}{s} \binom{\ell}{s} \leq \mathsf{bell}(k + \ell). \tag{8}$$

We will need some more preparations for the proof, in which we consider any $T \in \mathsf{LE}_{k \to \ell}$ only on step functions using regular intervals first, which will allow us to use the existing results for the discrete case.

**Lemma E.1** (Fixed points of measure preserving functions). *If $k \in \mathbb{N}_0$ and $U \in L^2[0,1]^k$ such that $U^\varphi = U$ for all $\varphi \in \overline{S}_{[0,1]}$, then $U$ is constant. All involved equalities are meant $\lambda^k$-almost everywhere.*

Although the proof is somewhat tedious, it is based on elementary measure theory. The aspect that warrants closer attention, however, is that $\varphi$ acts uniformly across all coordinates.

*Proof.* Let $U \in L^2[0,1]^k$ such that $U$ is invariant under all measure preserving functions, and suppose that $U$ is not constant $\lambda^k$-almost everywhere. Then, there exist $a < b$ such that $A := U^{-1}((-\infty, a])$, $B := U^{-1}([b, \infty))$ have positive Lebesgue measure $\lambda^k(A), \lambda^k(B) > 0$. Seeing $U$ as a random variable on the probability space $([0,1]^k, \lambda^k)$, the conditional distributions

$$\mathbb{P}(\cdot | U \leq a) = \frac{\lambda^k(\cdot \cap A)}{\lambda^k(A)} \quad \text{and} \quad \mathbb{P}(\cdot | U \geq b) = \frac{\lambda^k(\cdot \cap B)}{\lambda^k(B)} \tag{84}$$

are well-defined. For $n \in \mathbb{N}$, let $I_j^{(n)} := [\frac{j-1}{n}, \frac{j}{n})$ for $j \in \{1, \ldots, n-1\}$ and $I_n^{(n)} := [\frac{n-1}{n}, 1]$ be a partition of $[0,1]$ into regular intervals, and set $\mathcal{P}_k^{(n)} := \{I_{j_1}^{(n)} \times \cdots \times I_{j_k}^{(n)} \mid j_1, \ldots, j_k \in \{1, \ldots, n\}\}$. First, note that we have

$$\mathbb{P}(Q | U \leq a) \neq \mathbb{P}(Q | U \geq b) \tag{85}$$

for some $m \in \mathbb{N}$ and $Q \in \mathcal{P}_k^{(m)}$. Otherwise, equality in (85) would also hold for all hyperrectangles with rational endpoints, which is a $\cap$-stable generator of the Borel $\sigma$-algebra $\mathcal{B}([0,1]^k)$. Consequently, equality would hold for all sets in $\mathcal{B}([0,1]^k)$ and thus, $1 = \mathbb{P}(A | U \leq a) = \mathbb{P}(A | U \geq b) = \lambda^k(\varnothing)/\lambda^k(B) = 0$, which is a contradiction. W.l.o.g., assume $\mathbb{P}(Q | U \leq a) > \mathbb{P}(Q | U \geq b)$ in (85). As

$$\sum_{S \in \mathcal{P}_k^{(m)}} \mathbb{P}(S | U \leq a) = \sum_{S \in \mathcal{P}_k^{(m)}} \mathbb{P}(S | U \geq b) = 1, \tag{86}$$

there must be another $R \in \mathcal{P}_k^{(m)}$ such that $\mathbb{P}(R | U \leq a) < \mathbb{P}(R | U \geq b)$. Set

$$\Delta_k := \{\boldsymbol{x} \in [0,1]^k \mid |\{x_1, \ldots, x_k\}| < k\}, \quad \Delta_k^{(n)} := \{Q \in \mathcal{P}_k^{(n)} \mid Q \cap \Delta \neq \varnothing\} \tag{87}$$

to be the union of all diagonals on $[0,1]^k$ and the elements of $\mathcal{P}_k^{(n)}$ overlapping with $\Delta_k$ respectively for $n \in \mathbb{N}$. As $\lambda^k(\bigcup_{Q \in \Delta_k^{(n)}} Q) \to \lambda^k(\Delta_k) = 0$ as $n \to \infty$, there must exist $m^* \geq m \in \mathbb{N}$ such that there are $Q \supseteq Q^* \in \mathcal{P}_k^{(m^*)} \setminus \Delta_k^{(m^*)}$, $R \supseteq R^* \in \mathcal{P}_k^{(m^*)} \setminus \Delta_k^{(m^*)}$ satisfying

$$\mathbb{P}(Q^* | U \leq a) > \mathbb{P}(Q^* | U \geq b), \quad \mathbb{P}(R^* | U \leq a) < \mathbb{P}(R^* | U \geq b). \tag{88}$$

Since $Q^*$ and $R^*$ do not overlap with any diagonal, we can now construct $\varphi \in S_{[0,1]}$ such that $\varphi^{\otimes k}$, which clearly defines a measure preserving function from $[0,1]^k$ to itself, sends $Q^*$ to $R^*$. By invariance of $U$ under all measure preserving functions, we get

$$\lambda^k(R^* \cap A) = \lambda^k\left((\varphi^{\otimes k})^{-1}(R^* \cap A)\right) = \lambda^k(Q^* \cap A), \tag{89}$$

$$\lambda^k(R^* \cap B) = \lambda^k\left((\varphi^{\otimes k})^{-1}(R^* \cap B)\right) = \lambda^k(Q^* \cap B), \tag{90}$$

which contradicts (88). Hence, $U$ must be $\lambda^k$-a.e. constant. $\qquad \square$

Let $k, \ell \in \mathbb{N}_0$ and $n \in \mathbb{N}$. Let $I_j^{(n)} := [\frac{j-1}{n}, \frac{j}{n})$ for $j \in \{1, \ldots, n-1\}$ and $I_n^{(n)} := [\frac{n-1}{n}, 1]$ be a partition of $[0,1]$ into regular intervals. Let $\mathcal{A}_n := \sigma\left(\{I_1^{(n)}, \ldots, I_n^{(n)}\}\right)$ denote the $\sigma$-algebra generated by this partition and let

$$\mathcal{F}_k^{(n)} := \{U \in L^2[0,1]^k \,|\, U \text{ is } \mathcal{A}_n^{\otimes k}\text{-measurable}\}; \tag{91}$$

define $\mathcal{F}_\ell^{(n)}$ similarly. Intuitively, this is just the set of *regular $k$-dimensional $L^2$ step kernels*.

**Lemma E.2** (Invariance under discretization)**.** *Any $T \in \mathsf{LE}_{k \to \ell}$ is invariant under discretization, which means that for any $n \in \mathbb{N}$, $T(\mathcal{F}_k^{(n)}) \subseteq \mathcal{F}_\ell^{(n)}$, where the inclusion should be understood up to sets of measure zero.*

*Proof.* Let $T \in \mathsf{LE}_{k \to \ell}$ and let $U \in \mathcal{F}_k^{(n)}$. Then, if $\varphi \in \overline{S}_{[0,1]}$ such that

$$\varphi(I_j^{(n)}) \subseteq I_j^{(n)} \tag{92}$$

for any $j \in \{1, \ldots, n\}$, we have $U^\varphi = U$ and hence also $T(U)^\varphi = T(U)$ $\lambda^\ell$-almost everywhere. Take any hypercube $Q = I_{j_1}^{(n)} \times \cdots \times I_{j_\ell}^{(n)}$ with $j_1, \ldots, j_\ell \in \{1, \ldots, n\}$ and any measure-preserving function $\varphi : [0, 1/n) \to [0, 1/n)$. We replicate $\varphi$ on the unit interval as

$$\varphi^*(x) := x \operatorname{div} 1/n + \varphi(x \bmod 1/n), \tag{93}$$

which clearly satisfies (92), and thus $T(U)^{\varphi^*} = T(U)$ almost everywhere. Since now

$$T(U)\big|_Q = T(U)^{\varphi^*}\big|_Q = \left(T(U)\big|_Q\right)^\varphi, \tag{94}$$

where we identify $\varphi$ with $\varphi^*\big|_{I_j^{(n)}}$ (which define measure preserving functions on $I_j^{(n)}$), we can use translation invariance and scale equivariance of the Lebesgue measure to conclude by Lemma E.1 that $T(U)\big|_Q$ is constant $\lambda^\ell$-almost everywhere. As $Q$ was chosen arbitrarily, this implies the statement of the lemma. $\square$

Equipped with Lemma E.2, we are now ready to show that the dimension of linear equivariant layers in $[0, 1]$ is finite. Central to the argument is the observation that we can consider the action of any $T \in \mathsf{LE}_{k \to \ell}$ on step functions, and apply the characterization from Maron et al. (2019b) to a sequence of nested subspaces, which fixes the operator on the entire space $L^2[0, 1]^k$.

*Proof of Theorem 4.2.* Let $n \in \mathbb{N}$ and $T \in \mathsf{LE}_{k \to \ell}$. By Lemma E.2, we know that $T(\mathcal{F}_k^{(n)}) \subseteq \mathcal{F}_\ell^{(n)}$. Since $\mathcal{F}_k^{(n)} \cong (\mathbb{R}^n)^{\otimes k} \cong \mathbb{R}^{n^k}$, we can regard $T\big|_{\mathcal{F}_k^{(n)}} : \mathcal{F}_k^{(n)} \to \mathcal{F}_\ell^{(n)}$ as a linear operator $\mathbb{R}^{n^k} \to \mathbb{R}^{n^\ell}$. Taking for any $\sigma \in S_n$ a measure-preserving transformation $\varphi_\sigma \in S_{[0,1]}$ with $\varphi_\sigma(I_j) = I_{\sigma(j)}$, we can see that $T\big|_{\mathcal{F}_k^{(n)}}$ is also permutation equivariant, and we can use the characterization of the basis elements from Cai & Wang (2022) (see § B.2).

Note that for any $n, m \in \mathbb{N}$ we have $\mathcal{F}_k^{(n)} \subseteq \mathcal{F}_k^{(nm)}$ and the canonical basis elements $\{T_\gamma\}_{\gamma \in \Gamma_{k+\ell}}$ under the identification $\mathcal{F}_k^{(n)} \cong \mathbb{R}^{n^k}, \mathcal{F}_k^{(m)} \cong \mathbb{R}^{m^k}$ are compatible in the sense that

$$T_\gamma^{(nm)}\big|_{\mathcal{F}_k^{(n)}} = T_\gamma^{(n)}. \tag{95}$$

Hence, the coefficients of $T\big|_{\mathcal{F}_k^{(n)}}$ w.r.t. the canonical basis $\{T_\gamma^{(n)}\}_{\gamma \in \Gamma_{k+\ell}}$ do not depend on the specific $n \in \mathbb{N}$. W.l.o.g., assume that $T$ restricted to some $\mathcal{F}_k^{(n)}$ is a canonical basis function $T_{\gamma^*}^{(n)}$ (where $\gamma^* \in \Gamma_{k+\ell}$ does not depend on $n$).

We now take a closer look at the partition $\gamma^*$ and its induced function $T_{\gamma^*}^{(n)}$ described by the steps **Selection**, **Reduction**, **Alignment**, and **Replication**. Partition $\gamma^*$ into the 3 subsets $\gamma_1^* := \{A \in \gamma^* \,|\, A \subseteq [k]\}$, $\gamma_2^* := \{A \in \gamma^* \,|\, A \subseteq k + [\ell]\}$, $\gamma_3^* := \gamma^* \setminus (\gamma_1^* \cup \gamma_2^*)$. For the constant $U \equiv 1 \in \mathcal{F}_k^{(1)} \subseteq L^2[0,1]^k$, $T_{\gamma^*}^{(n)}(U) \neq 0$ must also be constant a.e. by compatibility with discretization, so

the partition $\gamma^*$ cannot correspond to a basis function whose images are supported on a diagonal. This is precisely equivalent to $|A \cap (k + [\ell])| \leq 1$ for all $A \in \gamma^*$. Now suppose that the input only depends on a diagonal. Denoting the restriction of the constant $U \equiv 1$ to the diagonal under discretization of $[0, 1]$ into $n$ pieces by $U_{\gamma^*}^{(n)}$,

$$T_{\gamma^*}^{(n)}(U_{\gamma^*}^{(n)}) = T_{\gamma^*}^{(n)}(U) = T_{\gamma^*}^{(1)}(U) \neq 0 \tag{96}$$

is constant, but $\|U_{\gamma^*}^{(n)}\|_2 \to 0$ for $n \to \infty$, which contradicts boundedness (i.e., continuity) of the operator $T \in \mathcal{L}(L^2[0,1]^k, L^2[0,1]^\ell)$. Hence, $\gamma^*$ must correspond to a basis function for which the selection step ① is trivial, i.e., $|A \cap [k]| \leq 1$ for all $A \in \gamma^*$.

This leaves us with only the partitions $\gamma \in \Gamma_{k+\ell}$ whose sets $A \in \gamma$ contain at most one element from $[k]$ and $k + [\ell]$ respectively. In the following Lemma E.3 we will check that for all of these partitions, the **Reduction/Alignment/Replication**-procedure (with averaging in the sense of integration over $[0,1]$) indeed yields a valid operator $T_\gamma \in \mathcal{L}(L^2[0,1]^k, L^2[0,1]^\ell)$ which agrees with $T_\gamma^{(n)}$ on $\mathcal{F}_k^{(n)}$.

If we can now show that $\bigcup_{n \in \mathbb{N}} \mathcal{F}_k^{(n)} \subseteq L^2[0,1]^k$ is dense w.r.t. $\|\cdot\|_2$, we can conclude that $T = T_{\gamma^*}$, as $T$ is continuous and agrees with $T_{\gamma^*}$ on a dense subset. However, this follows by a simple application of the martingale convergence theorem: Considering $[0,1]^k$ with Lebesgue measure $\lambda^k$ as a probability space and $U \in L^2[0,1]^k$ as a random variable, we have $\mathbb{E}[U|\mathcal{A}_n^{\otimes k}] \in \mathcal{F}_k^{(n)}$. Also, $\sigma\left(\bigcup_{n \in \mathbb{N}} \mathcal{A}_n^{\otimes k}\right) = \mathcal{B}([0,1]^k)$ is the entire Borel $\sigma$-Algebra as $\mathcal{A}_n$ contains all intervals with rational endpoints, so $\mathbb{E}[U|\mathcal{A}_n^{\otimes k}] \to \mathbb{E}[U|\mathcal{B}([0,1]^k)] = U$ in $L^2$.

Define now
$$\widetilde{\Gamma}_{k,\ell} := \{\gamma \in \Gamma_{k+\ell} \,|\, \forall A \in \gamma : |A \cap [k]| \leq 1, |A \cap k + [\ell]| \leq 1\} . \tag{97}$$
It is straightforward to show that

$$\left|\widetilde{\Gamma}_{k,\ell}\right| = \sum_{s=0}^{\min\{k,\ell\}} s! \binom{k}{s}\binom{\ell}{s}, \tag{98}$$

which can be seen as follows: Any partition on the l.h.s. can contain $s \in \{0, \ldots, \min\{k,\ell\}\}$ sets of size 2. Fixing some $s$, any of these sets can only contain one element from $[k]$ and one from $k + [\ell]$. For the elements occuring in sets of size 2, there are $\binom{k}{s}\binom{\ell}{s}$ options, and there are $s!$ ways to match the $s$ selected elements in $[k]$ with the $s$ elements in $k + [\ell]$, leaving us with the formula on the right. This concludes the proof. $\qquad\square$

Note the resemblance of the basis elements $T_\gamma$ (10) to the definition of *graphon-signal operators* from § D.1. Loosely speaking, all operators $T_\gamma$ are variants of (60), where the roles of $\boldsymbol{a}$ and $\boldsymbol{b}$ are switched, w.r.t. an *empty* graph $F$.

### E.2 CONTINUITY OF LINEAR EQUIVARIANT LAYERS

We note that the choice $p = 2$ in Definition 4.1 is somewhat arbitrary, and $T_\gamma$ can indeed be seen as an operator $L^p \to L^p$ for any $p \in [1, \infty]$, with $\|T_\gamma\|_{p \to p} = 1$. The case case $p = 2$ is technically still needed to complete the proof of Theorem 4.2.

**Lemma E.3** (Continuity of Linear Equivariant Layers w.r.t. $\|\cdot\|_p$)**.** *Fix $k, \ell \in \mathbb{N}_0$. Let $T \in \mathsf{LE}_{k \to \ell}$ and $p \in [1, \infty]$. Then, $T$ can also be regarded as a bounded linear operator $L^p[0,1]^k \to L^p[0,1]^\ell$. Furthermore, all of the canonical basis elements from the proof of Theorem 4.2 have operator norm $\|T\|_{p \to p} = 1$.*

*Proof.* If suffices to show boundedness of all canonical basis elements. Just as in (10), take $\gamma \in \widetilde{\Gamma}_{k,\ell}$ containing $s$ sets of size 2 $\{i_1, j_1\}, \ldots, \{i_s, j_s\}$ with $i_1, \ldots, i_s \in [k]$, $j_1, \ldots, j_s \in k + [\ell]$, and set $\boldsymbol{a} = (i_1, \ldots, i_s), \boldsymbol{b} = (j_1, \ldots, j_s)$. Write $T_\gamma$ as

$$T_\gamma(U) := \left[ [0,1]^\ell \ni \boldsymbol{y} \mapsto \int_{[0,1]^{k-s}} U(\boldsymbol{x}_{\boldsymbol{a}}, \boldsymbol{x}_{[k]\setminus \boldsymbol{a}}) \, \mathrm{d}\lambda^{k-s}(\boldsymbol{x}_{[k]\setminus \boldsymbol{a}}) \bigg|_{\boldsymbol{x}_{\boldsymbol{a}} = \boldsymbol{y}_{\boldsymbol{b}-k}} \right] . \tag{99}$$

Consider at first $p < \infty$. Clearly, (99) is also well-defined for $U \in L^p[0,1]^k$ and

$$\|T_\gamma(U)\|_p^p = \int_{[0,1]^\ell} \left| \int_{[0,1]^{k-s}} U(\boldsymbol{x_a}, \boldsymbol{x}_{[k]\backslash\boldsymbol{a}}) \, \mathrm{d}\lambda^{k-s}(\boldsymbol{x}_{[k]\backslash\boldsymbol{a}}) \right|_{\boldsymbol{x_a}=\boldsymbol{y_{b-k}}}^p \mathrm{d}\lambda^\ell(\boldsymbol{y}) \tag{100}$$

$$\leq \int_{[0,1]^\ell} \int_{[0,1]^{k-s}} \left| U(\boldsymbol{x_a}, \boldsymbol{x}_{[k]\backslash\boldsymbol{a}}) \right|^p \, \mathrm{d}\lambda^{k-s}(\boldsymbol{x}_{[k]\backslash\boldsymbol{a}}) \bigg|_{\boldsymbol{x_a}=\boldsymbol{y_{b-k}}} \mathrm{d}\lambda^\ell(\boldsymbol{y}) \tag{101}$$

$$= \int_{[0,1]^s} \int_{[0,1]^{k-s}} \left| U(\boldsymbol{x_a}, \boldsymbol{x}_{[k]\backslash\boldsymbol{a}}) \right|^p \, \mathrm{d}\lambda^{k-s}(\boldsymbol{x}_{[k]\backslash\boldsymbol{a}}) \, \mathrm{d}\lambda^s(\boldsymbol{x_a}) = \|U\|_p^p, \tag{102}$$

with Jensen's inequality being applied in the second step. Note that equality holds, e.g., for $U \equiv 1$, so $\|T_\gamma\|_{p\to p} = 1$. For $p = \infty$, we also see

$$\|T_\gamma(U)\|_\infty = \operatorname*{ess\,sup}_{\boldsymbol{y}\in[0,1]^\ell} \left| \int_{[0,1]^{k-s}} U(\boldsymbol{x_a}, \boldsymbol{x}_{[k]\backslash\boldsymbol{a}}) \, \mathrm{d}\lambda^{k-s}(\boldsymbol{x}_{[k]\backslash\boldsymbol{a}}) \right|_{\boldsymbol{x_a}=\boldsymbol{y_{b-k}}} \tag{103}$$

$$\leq \operatorname*{ess\,sup}_{\boldsymbol{y}\in[0,1]^\ell} \int_{[0,1]^{k-s}} \underbrace{\left| U(\boldsymbol{x_a}, \boldsymbol{x}_{[k]\backslash\boldsymbol{a}}) \right|}_{\leq \|U\|_\infty \text{ a.e.}} \, \mathrm{d}\lambda^{k-s}(\boldsymbol{x}_{[k]\backslash\boldsymbol{a}}) \bigg|_{\boldsymbol{x_a}=\boldsymbol{y_{b-k}}} \tag{104}$$

$$\leq \|U\|_\infty, \tag{105}$$

again with equality for $U \equiv 1$. $\qquad\square$

### E.3 ASYMPTOTIC DIMENSION OF LINEAR EQUIVARIANT LAYERS

We briefly analyze the asymptotic differences in dimension between $\mathsf{LE}^{[n]}_{k\to\ell}$, the linear equivariant layer space of discrete IGNs, and $\mathsf{LE}_{k\to\ell} = \mathsf{LE}^{[0,1]}_{k\to\ell}$, of IWNs. Recall that

$$\dim \mathsf{LE}^{[n]}_{k\to\ell} = \mathsf{bell}(k+\ell), \tag{106}$$

$$\dim \mathsf{LE}^{[0,1]}_{k\to\ell} = \sum_{s=0}^{\min\{k,\ell\}} s! \binom{k}{s}\binom{\ell}{s}. \tag{107}$$

For a comparison of the dimensions for the first few pairs $(k, \ell)$, see Table 2.

Table 2: Dimensions of $\mathsf{LE}^{[n]}_{k\to\ell}$ and $\mathsf{LE}^{[0,1]}_{k\to\ell}$.

| $\dim \mathsf{LE}^{[n]}_{k\to\ell}$ | 0 | 1 | 2 | 3 | 4 | $\dim \mathsf{LE}^{[0,1]}_{k\to\ell}$ | 0 | 1 | 2 | 3 | 4 |
|---|---|---|---|---|---|---|---|---|---|---|---|
| **0** | | 1 | 1 | 2 | 5 | 15 | **0** | 1 | 1 | 1 | 1 | 1 |
| **1** | | 1 | 2 | 5 | 15 | 52 | **1** | 1 | 2 | 3 | 4 | 5 |
| **2** | | 2 | 5 | 15 | 52 | 203 | **2** | 1 | 3 | 7 | 13 | 21 |
| **3** | | 5 | 15 | 52 | 203 | 877 | **3** | 1 | 4 | 13 | 34 | 73 |
| **4** | | 15 | 52 | 203 | 877 | 4140 | **4** | 1 | 5 | 21 | 73 | 209 |

**The case of bounded $k$ or $\ell$.** Immediately visible from Table 2 is the vastly different behavior of the two expressions as long as one of the variables $k$, $\ell$ is bounded: In the discrete case, whenever $k \to \infty$ or $\ell \to \infty$, we have $\mathsf{bell}(k + \ell) \to \infty$ superexponentially. However, for the case of $[0, 1]$, suppose w.l.o.g. that only $k \to \infty$ and $\ell = \mathcal{O}(1)$ remains constant. Then, the corresponding dimension growth is bounded by

$$\dim \mathsf{LE}^{[0,1]}_{k\to\ell} = \dim \mathsf{LE}^{[0,1]}_{\ell\to k} = \mathcal{O}(k^\ell), \tag{108}$$

as (107) is dominated by $\binom{k}{\ell}$ in this case.

**The case of $k \sim \ell$.** We will now consider the worst case, i.e., when $k$ grows roughly as fast as $\ell$. For simplicity, assume $k = \ell$, and thus

$$\dim \mathsf{LE}^{[n]}_{k \to k} \; = \; \mathsf{bell}(2k), \quad \dim \mathsf{LE}^{[0,1]}_{k \to k} \; = \; \sum_{s=0}^{k} s! \binom{k}{s}^2. \tag{109}$$

The bell numbers grow superexponentially, as can be seen by one of its asymptotic formulas (e.g., refer to Weisstein, Equation 19):

$$\mathsf{bell}(n) \; \sim \; \frac{1}{\sqrt{n}} \left( \frac{n}{W(n)} \right)^{n+1/2} \exp\left( \frac{n}{W(n)} - n - 1 \right), \tag{110}$$

where $W$ denotes the Lambert W-function, i.e., the inverse of $x \mapsto x \exp(x)$, or a simpler characterization due to Grunwald & Serafin (2025, Proposition 4.7), which is not strictly asymptotically tight but suffices in our case:

$$\left( \frac{1}{e} \frac{n}{\log n} \right)^n \; \leq \; \mathsf{bell}(n) \; \leq \; \left( \frac{3}{4} \frac{n}{\log n} \right)^n, \tag{111}$$

as long as $n \geq 2$. Therefore, the dimension of linear equivariant layers in the discrete case can be bounded as

$$\dim \mathsf{LE}^{[n]}_{k \to k} \; \geq \; \left( \frac{1}{e} \frac{2k}{\log 2k} \right)^{2k}. \tag{112}$$

We will now provide bounds on the dimension in the continuous case. First note that by only considering the last addend,

$$\dim \mathsf{LE}^{[0,1]}_{k \to k} \; \geq \; k! \; \geq \; \mathsf{bell}(k) \tag{113}$$

still grows superexponentially. A well-known bound on the factorial (see, e.g., Knuth (1997, § 1.2.5, Ex. 24)) is

$$\frac{n^n}{e^{n-1}} \; \leq \; n! \; \leq \; \frac{n^{n+1}}{e^{n-1}}, \tag{114}$$

for $n \in \mathbb{N}$. For a rough upper bound on the dimension, we consider just an even tensor order $k$:

$$\dim \mathsf{LE}^{[0,1]}_{k \to k} \; = \; \sum_{s=0}^{k} s! \binom{k}{s}^2 \tag{115}$$

$$\leq \; (k+1)k! \binom{k}{k/2}^2 \; = \; (k+1) \frac{k!^3}{(k/2)!^4} \tag{116}$$

$$\overset{(114)}{\leq} \; (k+1) \frac{k^{3k+3}}{e^{3k-3}} \frac{e^{2k-4}}{(k/2)^{2k}} \; = \; \frac{1}{e}(k+1)k^3 \left( \frac{4}{e} k \right)^k, \tag{117}$$

which still grows significantly slower than (112).

### E.4 PROOF OF PROPOSITION 4.4

**Proposition 4.4.** *Any IWN from (11) is an instance of IGN-small (Cai & Wang, 2022).*

*Proof.* Let $\mathcal{N}^{\mathrm{IWN}}$ be an IWN as in Definition 4.3. By invariance under discretization (Lemma E.2), we can see that any linear equivariant layer in $\mathcal{N}^{\mathrm{IWN}}$ fulfills a basis representation stability condition, and under the identification $\mathcal{F}^{(n)}_k \cong \mathbb{R}^{n^k}$, which is captured by grid-sampling, this directly implies that grid-sampling commutes with application of the IWN and its discrete equivalent. □

### E.5 PROOF OF LEMMA 4.5

**Lemma 4.5.** *Let $\mathcal{N}^{\mathrm{IWN}} : \mathcal{WS}_r \to \mathbb{R}$ be an IWN with Lipschitz continuous nonlinearity $\varrho$. Then, $\mathcal{N}^{\mathrm{IWN}}$ is Lipschitz continuous w.r.t. $\delta_p$ for each $p \in [1, \infty]$.*

*Proof of Lemma 4.5.* Let

$$\mathcal{N}^{\text{IWN}} = \mathfrak{T}^{(S)} \circ \varrho \circ \cdots \circ \varrho \circ \mathfrak{T}^{(1)} \tag{118}$$

be an IWN as in Definition 4.3. By Lemma E.3, each $T \in \mathsf{LE}_{k \to \ell}$ is Lipschitz continuous w.r.t. $\|\cdot\|_p$ on the respective input and output space, and this immediately carries over to all $\mathfrak{T}^{(s)}$, $s \in [S]$. Hence, it suffices to check that the pointwise application of the nonlinearity, i.e., $L^p[0,1]^k \ni U \mapsto \varrho(U)$, is Lipschitz continuous for every $k \in \mathbb{N}_0$, where $\varrho$ is applied elementwise. If $C_\varrho$ is the Lipschitz constant of $\varrho$, we have

$$\|\varrho(U) - \varrho(W)\|_p^p = \int_{[0,1]^k} |\varrho(U) - \varrho(W)|^p \, \mathrm{d}\lambda^k \leq \int_{[0,1]^k} C_\varrho^p |U - W|^p \, \mathrm{d}\lambda^k = C_\varrho^p \|U - W\|_p^p, \tag{119}$$

for $p < \infty$, and the claim follows similarly for $\|\cdot\|_\infty$. □

### E.6 PROOF OF THEOREM 4.6

We show that IWNs can approximate signal-weighted homomorphism densities w.r.t. graphs of size up to their order. For this, we model the product in the homomorphism densities explicitly while tracking which linear equivariant layers are being used by the IWN. The final result then follows by using a tree decomposition.

**Theorem 4.6** (Approximation of signal-weighted homomorphism densities). *Let $r > 0$, $1 < k \in \mathbb{N}$, $\varrho : \mathbb{R} \to \mathbb{R}$ Lipschitz continuous and non-polynomial, and $F$ be a multigraph of treewidth $k - 1$, $\boldsymbol{d} \in \mathbb{N}_0^{v(F)}$. Fix $\varepsilon > 0$. Then there exists an IWN $\mathcal{N}^{\text{IWN}}$ of order $k$ such that for all $(W, f) \in \mathcal{WS}_r$*

$$|t((F, \boldsymbol{d}), (W, f)) - \mathcal{N}^{\text{IWN}}(W, f)| \leq \varepsilon. \tag{13}$$

To establish this result, we first show that IWNs can approximate signal-weighted homomorphism densities for graphs of size up to their order. To facilitate the final step of combining these approximations via a tree decomposition, we refrain from integrating over all nodes of the pattern graph immediately. Instead, we keep some *labeled* nodes to serve as connection points—akin to the construction of signal-weighted homomorphism densities from *tri-labeled graphs* in § D.1.

**Lemma E.4.** *Let $r > 0$, $R > 0$, $\varrho : \mathbb{R} \to \mathbb{R}$ Lipschitz continuous and non-polynomial, and $k > 1$. Let $\boldsymbol{F} = (F, \boldsymbol{a}, \boldsymbol{b}, \boldsymbol{d}) \in \mathcal{M}^{\ell,k}$ be a tri-labeled graph with $V(F) = [k]$ and $\ell \leq k$. Write $\iota_k$ for the function that embeds any $U \in L^2[0,1]^m$ for $m \leq k$ as a $k$-tensor $\iota_k(U) := [[0,1]^k \ni \boldsymbol{x} \mapsto U(x_1, \ldots, x_m)]$. Consider now the map $\Phi_{\boldsymbol{F}}$ that implements*

$$\Phi_{\boldsymbol{F}} : \left(L^2[0,1]^k\right)^3 \to \left(L^2[0,1]^k\right)^3, \quad \begin{bmatrix} \iota_k(W) \\ \iota_k(f) \\ U \end{bmatrix} \mapsto \begin{bmatrix} \iota_k(W) \\ \iota_k(f) \\ (\iota_k \circ T_{\boldsymbol{F} \to (W,f)})U \end{bmatrix} \tag{120}$$

*for any $(W, f) \in \mathcal{WS}_r$ and $U \in L^2[0,1]^k$ (technically, this is only partially specified in the first two coordinates). Then, extending notation informally, for any $\varepsilon > 0$, there exists a $k$-order EWN $\mathcal{N}_{\boldsymbol{F}}^{\text{EWN}}$ with nonlinearity $\varrho$ such that uniformly on $(W, f) \in \mathcal{WS}_r$ and $U \in L_R^\infty[0,1]^k$*

$$\|\Phi_{\boldsymbol{F}}(W, f, U) - \mathcal{N}_{\boldsymbol{F}}^{\text{EWN}}(W, f, U)\|_{\infty,[0,1]^k} \leq \varepsilon. \tag{121}$$

For the proof, we model the product in the graphon-signal operators explicitly while tracking which linear equivariant layers are being used by the EWN.

We start with a few further preparations for the proofs. Namely, our first goal will be to show that if an EWN of a fixed order can approximate a set of functions, it can also approximate their product (Lemma E.6). First, we show how we can approximate the identity with an EWN:

**Lemma E.5.** *Let $k > 1$ and $R > 0$. Let $\varrho : \mathbb{R} \to \mathbb{R}$ be differentiable at least at one point with nonzero derivative. For any $\varepsilon > 0$, there exist $T^{(1)}, T^{(2)} \in \mathsf{LE}_{k \to k}$ such that for $\mathcal{N}^{\text{EWN}} := T^{(2)} \circ \varrho \circ T^{(1)}$ we have*

$$\|\mathcal{N}^{\text{EWN}}(U) - U\|_{\infty,[0,1]^k} \leq \varepsilon \tag{122}$$

*uniformly for all $U \in L_R^\infty[0,1]^k$.*

*Proof.* We simply approximate the identity function using the nonlinearity $\varrho$. Let $x_0 \in \mathbb{R}$ be a point at which $\varrho$ is differentiable with $\varrho'(x_0) \neq 0$. Fix $\varepsilon > 0$. There exists some constant $\delta > 0$ such that $|x| \leq \delta$ implies

$$|\varrho(x_0 + x) - \varrho(x_0) - \varrho'(x_0)x| \leq \varepsilon |x|. \tag{123}$$

Set

$$\mathrm{id}_\varrho(x) := \frac{R}{\delta \varrho'(x_0)} \left( \varrho \left( x_0 + \frac{\delta}{R}x \right) - \varrho(x_0) \right). \tag{124}$$

Then, for any $x \in [-R, R]$, $\left|\frac{\delta}{R}x\right| \leq \delta$ and hence

$$|\mathrm{id}_\varrho(x) - x| = \left| \frac{R}{\delta \varrho'(x_0)} \right| \left| \varrho \left( x_0 + \frac{\delta}{R}x \right) - \varrho(x_0) - \varrho'(x_0)\frac{\delta}{R}x \right| \tag{125}$$

$$\leq \left| \frac{R}{\delta \varrho'(x_0)} \right| \varepsilon \left| \frac{\delta}{R}x \right| \leq \frac{R}{|\varrho'(x_0)|}\varepsilon. \tag{126}$$

We can now simply set $\mathcal{N}^{\text{EWN}} := \mathrm{id}_\varrho$, acting on the function values of an input independently. Hence, for any $U \in L_R^\infty[0,1]^k$, we get

$$\|\mathcal{N}^{\text{EWN}}(U) - U\|_{\infty,[0,1]^k} \leq \frac{R}{|\varrho'(x_0)|}\varepsilon \tag{127}$$

directly by (126). Since the r.h.s. can be made arbitrarily small, the proof is complete. $\qquad\square$

**Lemma E.6.** *Let $k > 1$ and $R > 0$. Let $\mathcal{N}_1^{\text{EWN}}, \ldots, \mathcal{N}_m^{\text{EWN}}$ be $m \in \mathbb{N}$ EWNs of order $k \in \mathbb{N}$ and Lipschitz continuous nonlinearity $\varrho$, each with constant orders of $k$, the same input dimension $d$, and output dimension of $1$, i.e., returning a function $[0,1]^k \to \mathbb{R}$. Fix $\varepsilon > 0$. Then, there exists an EWN $\mathcal{N}^{\text{EWN}*}$ of order $k$, such that for all $U \in (L_R^\infty[0,1]^k)^d$*

$$\left\| \prod_{\ell=1}^m \mathcal{N}_\ell^{\text{EWN}}(U) - \mathcal{N}^{\text{EWN}*}(U) \right\|_{\infty,[0,1]^k} \leq \varepsilon. \tag{128}$$

The proof is partially analogous to Keriven & Peyré (2019). A crucial difference is that we do not rely on modeling multiplication by increasing the tensor orders.

*Proof.* We will exploit a property of $\cos$ that allows us to express products as sums. Namely, it is well-known that for $x_1, \ldots, x_m \in \mathbb{R}$ we have

$$\prod_{j=1}^m \cos(x_j) = \frac{1}{2^m} \sum_{\sigma \in \{\pm 1\}^m} \cos \left( \sum_{j=1}^m \sigma_j x_j \right). \tag{129}$$

Fix $\varepsilon > 0$. At first, we will describe how to approximate any $\mathcal{N}_\ell^{\text{EWN}}$ using $\cos$ as a nonlinearity. By the classical universal approximation theorem (see for example Pinkus (1999, Theorem 3.1)), we can approximate $\varrho : \mathbb{R} \to \mathbb{R}$ on compact sets arbitrarily well by a feedforward neural network $\varrho_{\cos}$ with one hidden layer, using the cosine function as nonlinearity. For all $\ell \in [m]$, the set of values of all intermediate computations in the evaluation of $\mathcal{N}_\ell^{\text{EWN}}(U)$ for $U \in (L_R^\infty[0,1]^k)^d$ is bounded, and we define $M_1 \in [0, \infty)$ to be the supremum of this set. Note that this means that $\varrho$ is only ever evaluated on the compact set $[-M_1, M_1]$ in $\{\mathcal{N}_\ell^{\text{EWN}}\}_\ell$. We can see that replacing each occurence of $\varrho$ in $\mathcal{N}_\ell^{\text{EWN}}$ by $\varrho_{\cos}$ and absorbing the linear factors into the linear equivariant layers again yields valid EWNs. Call these EWNs $\mathcal{N}_{\ell,\cos}^{\text{EWN}}$, $\ell = 1, \ldots, m$. The underlying feedforward neural network $\varrho_{\cos}$ can now be chosen such that for all $\ell \in [m]$, $U \in (L_R^\infty[0,1]^k)^d$

$$\left\| \mathcal{N}_{\ell,\cos}^{\text{EWN}}(U) - \mathcal{N}_\ell^{\text{EWN}}(U) \right\|_{\infty,[0,1]^k} \leq \varepsilon. \tag{130}$$

Note that each $\mathcal{N}_{\ell,\cos}^{\text{EWN}}$ might have different numbers of layers. Hence, we invoke Lemma E.5 to equalize the number of layers, and add one more layer of the identity. We obtain EWNs $\widetilde{\mathcal{N}_{\ell,\cos}^{\text{EWN}}}$ such that

$$\left\| \mathrm{id}_{\cos}(\widetilde{\mathcal{N}_{\ell,\cos}^{\text{EWN}}}(U)) - \mathcal{N}_{\ell,\cos}^{\text{EWN}}(U) \right\|_{\infty,[0,1]^k} \leq \varepsilon \tag{131}$$

for all $U \in (L_R^\infty[0,1]^k)^d$. Using (129) and setting $\mathrm{id}_{\cos}(x) = c \cdot \cos(ax+b) + d$ for some $a, b, c, d \in \mathbb{R}$, we can now write

$$\prod_{\ell=1}^m \mathrm{id}_{\cos}(\widetilde{\mathcal{N}_{\ell,\cos}^{\mathrm{EWN}}}(U)) = \prod_{\ell=1}^m \left( c \cdot \cos(a \cdot \widetilde{\mathcal{N}_{\ell,\cos}^{\mathrm{EWN}}}(U) + b) + d \right) \tag{132}$$

$$= \sum_{A \subseteq [m]} c^{|A|} d^{m-|A|} \left( \prod_{\ell \in A} \cos(a \cdot \widetilde{\mathcal{N}_{\ell,\cos}^{\mathrm{EWN}}}(U) + b) \right) \tag{133}$$

$$= \sum_{A \subseteq [m]} \frac{c^{|A|} d^{m-|A|}}{2^{|A|}} \sum_{\sigma \in \{\pm 1\}^A} \cos \underbrace{\left( \sum_{\ell \in A} \sigma_\ell (a \cdot \widetilde{\mathcal{N}_{\ell,\cos}^{\mathrm{EWN}}}(U) + b) \right)}_{(*)}, \tag{134}$$

which can be represented by an EWN of order $k$ and one layer more compared to $\{\widetilde{\mathcal{N}_{\ell,\cos}^{\mathrm{EWN}}}\}_\ell$, stacking the expressions from $(*)$, applying the nonlinearity $\cos$ and aggregating the outputs as determined by the weighted sum. Let

$$M_2 := \max_{\ell \in [m]} \sup_U \|\mathcal{N}_\ell^{\mathrm{EWN}}(U)\|_{\infty,[0,1]^k}. \tag{135}$$

Since for any $U \in (L_R^\infty[0,1]^k)^d$

$$\left\| \prod_{\ell=1}^m \mathrm{id}_{\cos}(\widetilde{\mathcal{N}_{\ell,\cos}^{\mathrm{EWN}}}(U)) - \prod_{\ell=1}^m \mathcal{N}_\ell^{\mathrm{EWN}}(U) \right\|_{\infty,[0,1]^k} \tag{136}$$

$$\leq \sum_{\ell=1}^m \left\| \left( \prod_{i>\ell} \mathrm{id}_{\cos}(\widetilde{\mathcal{N}_{\ell,\cos}^{\mathrm{EWN}}}(U)) \prod_{i<\ell} \mathcal{N}_\ell^{\mathrm{EWN}}(U) \right) \left( \mathrm{id}_{\cos}(\widetilde{\mathcal{N}_{\ell,\cos}^{\mathrm{EWN}}}(U)) - \mathcal{N}_\ell^{\mathrm{EWN}}(U) \right) \right\|_{\infty,[0,1]^k} \tag{137}$$

$$\leq \sum_{\ell=1}^m \left\| \prod_{i>\ell} \mathrm{id}_{\cos}(\widetilde{\mathcal{N}_{\ell,\cos}^{\mathrm{EWN}}}(U)) \prod_{i<\ell} \mathcal{N}_\ell^{\mathrm{EWN}}(U) \right\|_{\infty,[0,1]^k} \left\| \mathrm{id}_{\cos}(\widetilde{\mathcal{N}_{\ell,\cos}^{\mathrm{EWN}}}(U)) - \mathcal{N}_\ell^{\mathrm{EWN}}(U) \right\|_{\infty,[0,1]^k} \tag{138}$$

$$\leq \sum_{\ell=1}^m (M_2 + 2\varepsilon)^{m-\ell} M_2^{\ell-1} 2\varepsilon, \tag{139}$$

and since (139) goes to zero as $\varepsilon \to 0$, this shows the claim for $\cos$. Another application of the universal approximation theorem yields the claim for $\varrho$. $\qquad\square$

The proof of Lemma E.4 now boils down to an application of Lemma E.6.

*Proof of Lemma E.4.* Fix a tri-labeled graph $\boldsymbol{F} = (F, \boldsymbol{a}, \boldsymbol{b}, \boldsymbol{d})$ with $V(F) = [k]$, $\boldsymbol{a}$ and $\boldsymbol{b}$ $\ell$- and $k$-tuples over $[k]$ respectively. Let $(W, f) \in \mathcal{WS}_r$ and $U \in L_R^\infty[0,1]^k$. Define

$$f_i : [0,1]^k \to \mathbb{R}, \ \boldsymbol{x} \mapsto f(x_i) \qquad \text{for } i \in V(F), \tag{140}$$

$$W_{\{i,j\}} : [0,1]^k \to \mathbb{R}, \ \boldsymbol{x} \mapsto W(x_i, x_j) \qquad \text{for } \{i,j\} \in E(F). \tag{141}$$

Clearly, $[\iota_k(W), \iota_k(f), U]^\top \mapsto f_i$, $[\iota_k(W), \iota_k(f), U]^\top \mapsto W_{\{i,j\}}$, and $[\iota_k(W), \iota_k(f), U]^\top \mapsto U_{\boldsymbol{b}} := [\boldsymbol{x} \mapsto U(\boldsymbol{x}_{\boldsymbol{b}})]$ can be exactly represented by EWNs of order $k$ with one layer (i.e., no application of the nonlinearity at all). By Lemma E.6, the product

$$\begin{bmatrix} \iota_k(W) \\ \iota_k(f) \\ U \end{bmatrix} \mapsto \left( \prod_{i \in V(F)} f_i^{d_i} \right) \left( \prod_{\{i,j\} \in E(F)} W_{\{i,j\}} \right) U_{\boldsymbol{b}} \tag{142}$$

can be approximated by an EWN of order $k$ arbitrarily well in $\|\cdot\|_\infty$. However, setting $D := \sum_i d_i$,

$$T_{\boldsymbol{F} \to (W,f)} U = \int_{[0,1]^{k-\ell}} \left( \prod_{i \in V(F)} f_i^{d_i} \right) \left( \prod_{\{i,j\} \in E(F)} W_{\{i,j\}} \right) U_{\boldsymbol{b}} \, \mathrm{d}\lambda^{[k] \setminus \boldsymbol{a}} \in L_{Rr^D}^\infty[0,1]^\ell, \tag{143}$$

and combined with applying Lemma E.5 again on $W$ and $f$, one concludes that

$$\begin{bmatrix} \iota_k(W) \\ \iota_k(f) \\ U \end{bmatrix} \mapsto \begin{bmatrix} \iota_k(W) \\ \iota_k(f) \\ (\iota_k \circ T_{\boldsymbol{F} \to (W,f)})U \end{bmatrix} \tag{144}$$

can be approximated arbitrarily well by an EWN for $(W, f) \in \mathcal{WS}_r$ and $U \in L_R^\infty[0,1]^k$. $\qquad\square$

With these preparations in place, we are now ready to prove Theorem 4.6. The proof proceeds by constructing a signal-weighted homomorphism density step by step using $\mathcal{F}^k$-terms of tri-labeled graphs, as outlined in § D.1.

*Proof of Theorem 4.6.* We first consider the operators $\Phi_{\boldsymbol{F}_{\mathrm{atom}}}$ from Lemma E.4 for the atomic graphs $\boldsymbol{F}_{\mathrm{atom}}$ from Definition D.2. The restriction we imposed on the tri-labeled graphs $\boldsymbol{F} = (F, \boldsymbol{a}, \boldsymbol{b}, \boldsymbol{d}) \in \mathcal{M}^{\ell,k}$ to have $V(F) = [k]$ is straightforwardly fulfilled by the adjacency graphs $\boldsymbol{A}_{\{i,j\}}^{(k)}$ and signal graphs $\boldsymbol{S}_{\boldsymbol{d}}^{(k)}$. Any neighborhood graph can be implemented by two such operators $\Phi_{\boldsymbol{F}}$, where only the labels in the introduce graph have to be permuted to account for the trailing dimension caused by our use of the $k$-order embedding $\iota_k$. In total, this means that $\Phi_{\boldsymbol{F}_{\mathrm{atom}}}$ for all atomic tri-labeled graphs in $\mathcal{F}^k$ can be written as the composition of at most two of the functions in Lemma E.4. By precisely this lemma, we know that $\Phi_{\boldsymbol{F}_{\mathrm{atom}}}$ can be approximated by some EWN $\mathcal{N}_{\boldsymbol{F}_{\mathrm{atom}}}^{\mathrm{EWN}}$ up to arbitrary precision in $\|\cdot\|_\infty$ and on some restriction on the values of the input tensor $U$. Note that the composition of such functions can again be trivially approximated by an EWN (where the first and last linear equivariant layers can be merged). It is crucial here that the set of all tensor values is bounded and that any EWN as well as $\Phi_{\boldsymbol{F}}$ is $L^\infty$-Lipschitz.

Now, for any term $\mathbb{F} \in \langle \mathcal{F}^k \rangle$, it is possible to approximate $\Phi_{[\![\mathbb{F}]\!]}$ with arbitrary precision by an EWN. To start, simply take $\Phi_{\boldsymbol{1}^{(k)}}(W,f) := [\iota_k(W), \iota_k(f), \mathbb{1}_{[0,1]^k}]^\top$, which can clearly be implemented by an EWN. As already mentioned, the composition of two approximable $\Phi_{\boldsymbol{F}}$ can be clearly also approximated by an EWN, and the product in the third "register" acting on $U$ as well by Lemma E.6. By the definition of $\langle \mathcal{F}^k \rangle$ and (61)/(62), this implies that $\Phi_{[\![\mathbb{F}]\!]}$ can indeed be approximated up to arbitrary precision for any $\mathcal{F}^k$-term $\mathbb{F}$. Letting $[\![\mathbb{F}]\!] = (F, \boldsymbol{a}, \varnothing, \boldsymbol{d}) \in \mathcal{M}^{k,0}$, we get by (63) that

$$(W, f) \overset{\mathcal{N}_{[\![\mathbb{F}]\!]}^{\mathrm{EWN}} \approx \Phi_{[\![\mathbb{F}]\!]}}{\mapsto} \begin{bmatrix} \iota_k(W) \\ \iota_k(f) \\ T_{[\![\mathbb{F}]\!] \to (W,f)}(1) \end{bmatrix} \mapsto \int_{[0,1]^k} T_{[\![\mathbb{F}]\!] \to (W,f)}(1) \, \mathrm{d}\lambda^k \;=\; t((F, \boldsymbol{d}), (W, f)), \quad (145)$$

and the mapping on the l.h.s. can be collapsed to an IWN $\mathcal{N}_{[\![\mathbb{F}]\!]}^{\mathrm{IWN}}$. Hence, the r.h.s. in (145) can also be approximated uniformly by an IWN. The claim of Theorem 4.6 follows, as $\mathcal{F}^k$-terms parametrize all signal-weighted homomorphism densities up to treewidth $k-1$ by Theorem D.3. $\qquad\square$

### E.7 Proof of Corollary 4.7

Having shown Theorem 4.6, $k$-WL expressivity can be obtained effectively by definition.

**Corollary 4.7** ($k$-WL expressivity). *$\mathfrak{F}_\varrho^{k\text{-IWN}}$ is at least as expressive as the $k$-WL test at distinguishing graphon-signals.*

*Proof.* Let $(W, f), (V, g) \in \mathcal{WS}_r$ be distinguishable by $k$-WL. By Theorem D.10, this means that there exists a multigraph $F$ of treewidth at most $k-1$ and $\boldsymbol{d} \in \mathbb{N}_0^{v(F)}$ for which

$$t((F, \boldsymbol{d}), (W, f)) \;\neq\; t((F, \boldsymbol{d}), (V, g)). \tag{146}$$

Let $\varepsilon := |t((F, \boldsymbol{d}), (W, f)) - t((F, \boldsymbol{d}), (V, g))| > 0$. By Theorem 4.6, take a $k$-order IWN $\mathcal{N}^{\mathrm{IWN}}$ such that

$$\sup_{(U,h) \in \mathcal{WS}_r} |\mathcal{N}^{\mathrm{IWN}}(U, h) - t((F, \boldsymbol{d}), (U, h))| \leq \varepsilon/3. \tag{147}$$

We obtain

$$|\mathcal{N}^{\mathrm{IWN}}(W, f) - \mathcal{N}^{\mathrm{IWN}}(V, g)| \tag{148}$$

$$\geq |t((F, \boldsymbol{d}), (W, f)) - t((F, \boldsymbol{d}), (V, g))| \tag{149}$$

$$- |\mathcal{N}^{\mathrm{IWN}}(W, f) - t((F, \boldsymbol{d}), (W, f)))| - |t((F, \boldsymbol{d}), (V, g)) - \mathcal{N}^{\mathrm{IWN}}(V, g)|$$

$$\geq \varepsilon - \varepsilon/3 - \varepsilon/3 \;=\; \varepsilon/3 > 0, \tag{150}$$

which yields the claim. □

## E.8 PROOF OF COROLLARY 4.8

Note that Theorem 4.6 tells us that IWNs of *arbitrary order* can approximate any signal-weighted homomorphism density. As these separate points in $\widetilde{\mathcal{WS}}_r$ by Theorem 3.2, it is straightforward to obtain universality on any set which is compact w.r.t. a distance on $\widetilde{\mathcal{WS}}_r$ under which the signal-weighted homomorphism densities are continuous.

**Corollary 4.8** ($\delta_p$-Universality of IWNs)**.** *Let $r > 1$, $p \in [1, \infty)$, $\varrho : \mathbb{R} \to \mathbb{R}$ Lipschitz continuous and non-polynomial. For any compact $K \subset (\widetilde{\mathcal{WS}}_r, \delta_p)$, $\mathfrak{F}_\varrho^{\mathrm{IWN}}$ is dense in the continuous functions $C(K)$ w.r.t. $\|\cdot\|_\infty$.*

*Proof.* We show this statement by applying the Stone-Weierstrass theorem. In essence, the main part of the proof already lies in establishing the approximation of the signal-weighted homomorphism densities (Theorem 4.6). Fix a compact subset $K \subset (\widetilde{\mathcal{WS}}_r, \delta_p)$. Consider the space

$$\mathcal{D} := \operatorname{span}\{t((F, \boldsymbol{d}), \cdot) \mid F \text{ multigraph}, \boldsymbol{d} \in \mathbb{N}_0^{v(F)}\} \subseteq C(K, \mathbb{R}). \tag{151}$$

Clearly, $\mathcal{D}$ is a linear subspace, and $\mathcal{D}$ contains a non-zero constant function as we can take a homomorphism density of a graph $F$ with no edges and $\boldsymbol{d} = 0$. Also, it is straightforward to see that $\mathcal{D}$ is a subalgebra, as for any two multigraphs $F_1, F_2$, $\boldsymbol{d}_1 \in \mathbb{N}_0^{v(F_1)}$, $\boldsymbol{d}_2 \in \mathbb{N}_0^{v(F_2)}$,

$$t((F_1, \boldsymbol{d}_1), \cdot) \cdot t((F_2, \boldsymbol{d}_2), \cdot) = t((F_1 \sqcup F_2, \boldsymbol{d}_1 \| \boldsymbol{d}_2), \cdot) \in \mathcal{D}, \tag{152}$$

i.e., the product of homomorphism densities w.r.t. two simple graphs can be rewritten as the homomorphism density w.r.t. their disjoint union. By Theorem 3.2, $\mathcal{D}$ also separates points, and we can apply Stone-Weierstrass (note that $K$ is a metric space and, thus, particularly Hausdorff) to conclude that $\mathcal{D} \subseteq C(K, \mathbb{R})$ is dense. However, by Theorem 4.6, any element of $\mathcal{D}$ can be approximated with arbitrary precision by $\mathfrak{F}_\varrho^{\mathrm{IWN}}$, and thus

$$C(K, \mathbb{R}) = \overline{\mathcal{D}} \subseteq \overline{\mathfrak{F}_\varrho^{\mathrm{IWN}}} \subseteq C(K, \mathbb{R}). \tag{153}$$

This concludes the proof. □

## F  REFINEMENT-BASED GRAPHON NEURAL NETWORKS AND $k$-WL

In this section, we extend the notion of a $k$–order GNN to the graphon–signal setting. In analogy to IWNs, we call such networks *higher-order graphon neural networks (WNNs)*. Moreover, we show that—similarly as signal-weighted homomorphism densities in (82)—such WNNs can also be written as continuous functions on the space of $k$-WL measures, and related back to their original formulation via the graphon-signal's $k$-WL *distribution* (see § D.3). In the following, we will make this precise.

### F.1  DEFINITION OF GRAPHON NEURAL NETWORKS

In the following, we extend higher-order graph neural networks that mimic the $k$-WL test (§ D) to graphon-signals.

**Definition F.1** ($k$–order graphon neural network). *Fix $k > 1$. An $S$-layer $k$–**order graphon neural network (WNN)** is a function $\mathcal{N}^{k\text{-WNN}} : \mathcal{WS}_r \to \mathbb{R}$ that uses the atomic types from Definition D.8 as initialization and updates its embeddings $\boldsymbol{h}^{(s)} : [0,1]^k \to \mathbb{R}^{d_s}$ iteratively for an input $(W, f) \in \mathcal{WS}_r$ and $\boldsymbol{x} \in [0,1]^k$ as*

$$\boldsymbol{h}^{(0)}(\boldsymbol{x}) := \left( \left( W(x_i, x_j) \right)_{\{i,j\} \in \binom{[k]}{2}}, \left( f(x_j) \right)_{j \in [k]} \right), \tag{154}$$

$$\boldsymbol{h}^{(s)}(\boldsymbol{x}) := \mathcal{N}^{(s)}_{\text{UPDATE}} \left( \boldsymbol{h}^{(s-1)}(\boldsymbol{x}), \left( \int_{[0,1]} \mathcal{N}^{(s)}_j (\boldsymbol{h}^{(s-1)}(\dots, x_{j-1}, y, x_{j+1}, \dots)) \, \mathrm{d}\lambda(y) \right)_{j \in [k]} \right), \tag{155}$$

*and*

$$\mathcal{N}^{k\text{-WNN}}(W, f) := \mathcal{N}_{\text{READOUT}} \left( \int_{[0,1]^k} \boldsymbol{h}^{(S)} \, \mathrm{d}\lambda^k \right). \tag{156}$$

*Here, $d_0 = \binom{k}{2} + k$, and $\mathcal{N}^{(s)}_j : \mathbb{R}^{d_{s-1}} \to \mathbb{R}^{\widetilde{d_s}}$, $\mathcal{N}^{(s)}_{\text{UPDATE}} : \mathbb{R}^{d_s} \times (\mathbb{R}^{\widetilde{d_s}})^k \to \mathbb{R}^{d_{s+1}}$, $\mathcal{N}_{\text{READOUT}} : \mathbb{R}^{d_S} \to \mathbb{R}$ are implemented by MLPs.*

Under suitable assumptions on the MLPs, i.e., (Lipschitz) continuity of the nonlinearity, it is straight-forward to show that such a WNN is (Lipschitz) continuous in any $L^p$ norm. By invariance w.r.t. measure preserving maps, this extends easily to continuity in the $\delta_p$ distances (cf. § C.2). It is immediately apparent that this mirrors the discrete version (Morris et al., 2019) and aligns with the $k$-WL distribution (Definition D.8).

### F.2  FORMULATION ON $k$–WL MEASURES

In the following, we show how the WNNs from Definition F.1 can be factorized as a function of the $k$-WL measure $\nu^{k\text{-WL}}_{(W,f)}$ for a graphon-signal $(W, f)$ (§ D.3). Effectively, this will allow us to regard $\mathcal{N}^{k\text{-WNN}}$ *not* as a function on the uncontrollable spaces $(\widetilde{\mathcal{WS}}_r, \delta_p)$, but as functions on the *compact* space $\mathcal{P}(\mathbb{M}^k)$ of probability measures over the *colors* produced by the $k$-WL test. In the discrete case, it is clear that any graph parameter which is *at most* $k$-WL expressive, could alternatively be seen as a function on the $k$-WL colors. In our setting, however, additional care is needed to ensure that the defined functions are well-defined and continuous w.r.t. the considered topologies. The idea of considering GNN outputs on the space of WL colors was already introduced by Böker et al. (2023) for MPNNs.

**Definition F.2** ($k$-order graphon neural network on $k$-WL measure). *Let $k > 1$ and fix $S \in \mathbb{N}$. Take the MLPs $\mathcal{N}^{(s)}_j$, $\mathcal{N}^{(s)}_{\text{UPDATE}}$, $\mathcal{N}_{\text{READOUT}}$ precisely from Definition F.1. Define the following embeddings $\boldsymbol{h}^{(s)} : \mathbb{M}^k_s \to \mathbb{R}^{d_s}$ on the space of $k$-WL measures (to be exact, their first coordinates) as follows:*

$$\boldsymbol{h}^{(0)}(\alpha) := \alpha \in \mathbb{R}^{d_0}, \tag{157}$$

$$\boldsymbol{h}^{(s)}(\alpha) := \mathcal{N}^{(s)}_{\text{UPDATE}} \left( \boldsymbol{h}^{(s-1)}(p_{s \to s-1}(\alpha)), \left( \int_{\mathbb{M}^k_{s-1}} \mathcal{N}^{(s)}_j \circ \boldsymbol{h}^{(s-1)} \mathrm{d}(\alpha_s)_j \right)_{j \in [k]} \right), \quad \forall s \in [S]. \tag{158}$$

*Finally, for any $\nu \in \mathcal{P}(\mathbb{M}^k)$, set*

$$\mathcal{N}^{k\text{-WNN}}(\nu) := \mathcal{N}_{\text{READOUT}} \left( \int_{\mathbb{M}^k} \boldsymbol{h}^{(S)} \circ p_{\infty \to S} \, \mathrm{d}\nu \right). \tag{159}$$

Measurability of all involved functions can be checked easily. We will demonstrate that $\mathcal{N}^{k\text{-WNN}}$ is indeed *continuous* w.r.t. the weak topology on $\mathcal{P}(\mathbb{M}^k)$:

**Lemma F.3** (Continuity of $k$-WNN). *Let $\mathcal{N}^{k\text{-WNN}} : \mathcal{P}(\mathbb{M}^k) \to \mathbb{R}$ be a WNN from Definition F.2. Suppose all involved MLPs are continuous. Then, $\boldsymbol{h}^{(S)} \in C(\mathbb{M}^k, \mathbb{R}^{d_S})$, and $\mathcal{N}^{k\text{-WNN}} \in C(\mathcal{P}(\mathbb{M}^k))$.*

*Proof.* For the first statement, proceed via induction. For $s = 0$, $\boldsymbol{h}^{(0)}$ is the identity, so there is nothing to show. Fix $s \in \mathbb{N}$ and suppose that $\boldsymbol{h}^{(s-1)} : \mathbb{M}^k_{s-1} \to \mathbb{R}^{d_s}$ is continuous. Let $(\alpha_n)_n$ be a sequence in $\mathbb{M}^k_s$ that converges to $\alpha \in \mathbb{M}^k_s$. By the induction hypothesis, $\boldsymbol{h}^{(s-1)}(p_{s \to s-1}(\alpha_n)) \to \boldsymbol{h}^{(s-1)}(p_{s \to s-1}(\alpha))$. The convergence

$$\int_{\mathbb{M}^k_{s-1}} \mathcal{N}_j^{(s)} \circ \boldsymbol{h}^{(s-1)} \mathrm{d}((\alpha_n)_s)_j \to \int_{\mathbb{M}^k_{s-1}} \mathcal{N}_j^{(s)} \circ \boldsymbol{h}^{(s-1)} \mathrm{d}(\alpha_s)_j \tag{160}$$

follows similarly as $\mathcal{N}_j^{(s)} \circ \boldsymbol{h}^{(s-1)} \in C(\mathbb{M}^k_{s-1}, \mathbb{R}^{d_s})$ and $\mathcal{P}(\mathbb{M}^k_{s-1}) \ni ((\alpha_n)_s)_j \xrightarrow{w} (\alpha_s)_j$. Continuity of $\mathcal{N}_{\text{UPDATE}}^{(s)}$ finally yields continuity of $\boldsymbol{h}^{(s)}$. Thus, the proof of the first statement is complete. The second part follows directly, as $\boldsymbol{h}^{(S)} \circ p_{\infty \to S} \in C(\mathbb{M}^k, \mathbb{R}^{d_S})$, $\mathcal{N}_{\text{READOUT}}$ is continuous, and we are considering the topology of weak convergence on $\mathcal{P}(\mathbb{M}^k)$. $\square$

In the following theorem, we will get to the objective of this section: Showing that the two formulations of $k$-order GNNs we have stated so far are indeed equivalent, in the sense that applying the $k$-WL measure version to the $k$-WL distribution of a graphon-signal (Definition F.2) indeed gives the same output as the original definition (Definition F.1).

**Theorem F.4** (Equivalence of formulations). *Let $(W, f) \in \mathcal{WS}_r$, and let $\mathcal{N}^{k\text{-WNN}}$ be an WNN. For the formulations from Definition F.1 and Definition F.2, we obtain that*

$$\mathcal{N}^{k\text{-WNN}}(W, f) = \mathcal{N}^{k\text{-WNN}}(\nu_{(W,f)}^{k\text{-WL}}). \tag{161}$$

*Proof.* We first show that

$$\boldsymbol{h}^{(s)}(\boldsymbol{x}) = \boldsymbol{h}^{(s)} \left( \mathfrak{C}_{(W,f)}^{k\text{-WL},(s)}(\boldsymbol{x}) \right) \tag{162}$$

for all $s \in \mathbb{N}_0$ and $\lambda^k$-almost all $\boldsymbol{x} \in [0,1]^k$. We proceed via induction. For $s = 0$ this is obvious, as $\mathcal{N}^{k\text{-WNN}}$ is initialized with the atomic types of the $k$-WL test. Consider some fixed but arbitrary $s \in \mathbb{N}$, and suppose that the equality (162) holds for $s - 1$. Then,

$$\boldsymbol{h}^{(s)} \left( \mathfrak{C}_{(W,f)}^{k\text{-WL},(s)}(\boldsymbol{x}) \right) = \mathcal{N}_{\text{UPDATE}}^{(s)} \Bigg( \boldsymbol{h}^{(s-1)} \left( \mathfrak{C}_{(W,f)}^{k\text{-WL},(s-1)}(\boldsymbol{x}) \right),$$

$$\left( \int_{\mathbb{M}^k_{s-1}} \mathcal{N}_j^{(s)} \circ \boldsymbol{h}^{(s-1)} \, \mathrm{d}\big( \mathfrak{C}_{(W,f)}^{k\text{-WL},(s-1)} \circ \boldsymbol{x}[\cdot/j] \big)_* \lambda \right)_{j \in [k]} \Bigg) \tag{163}$$

$$= \mathcal{N}_{\text{UPDATE}}^{(s)} \Bigg( \boldsymbol{h}^{(s-1)} \left( \mathfrak{C}_{(W,f)}^{k\text{-WL},(s-1)}(\boldsymbol{x}) \right),$$

$$\left( \int_{[0,1]^k} \mathcal{N}_j^{(s)} \circ \boldsymbol{h}^{(s-1)} \circ \mathfrak{C}_{(W,f)}^{k\text{-WL},(s-1)} \, \mathrm{d}\big( \boldsymbol{x}[\cdot/j] \big)_* \lambda \right)_{j \in [k]} \Bigg) \tag{164}$$

$$\stackrel{(*)}{=} \mathcal{N}_{\text{UPDATE}}^{(s)} \Bigg( \boldsymbol{h}^{(s-1)}(\boldsymbol{x}), \left( \int_{[0,1]^k} \mathcal{N}_j^{(s)} \circ \boldsymbol{h}^{(s-1)} \, \mathrm{d}\big( \boldsymbol{x}[\cdot/j] \big)_* \lambda \right)_{j \in [k]} \Bigg) \tag{165}$$

$$= \mathcal{N}_{\text{UPDATE}}^{(s)} \Bigg( \boldsymbol{h}^{(s-1)}(\boldsymbol{x}), \left( \int_{[0,1]} \mathcal{N}_j^{(s)}(\boldsymbol{h}^{(s-1)}(\boldsymbol{x}[\cdot/j])) \, \mathrm{d}\lambda \right)_{j \in [k]} \Bigg) \tag{166}$$

$$= \boldsymbol{h}^{(s)}(\boldsymbol{x}), \tag{167}$$

where we used the induction hypothesis in (165). This proves (162). For the final output $\mathcal{N}^{k\text{-WNN}}$, the equality follows just as easily by moving the pushforward $\nu_{(W,f)}^{k\text{-WL}} = (\mathfrak{C}_{(W,f)}^{k\text{-WL}})_* \lambda^k$ into the integrand.

$\square$

Note that—akin to signal-weighted homomorphism densities (§ D.2)—one could apply the Stone-Weierstrass theorem to show that the set of all $k$-order WNNs (or IWNs) is universal on $\mathbb{M}^k$ and, therefore, $k$-order WNNs are $k$-WL expressive. We omit the proof here.

Similar characterizations as in this section could also be derived for the *Folklore $k$-WL* test (see, e.g., Jegelka (2022)) and the corresponding $k$-FGNNs (Maron et al., 2019a), which achieve the same expressivity as $(k+1)$-GNNs. While extending the $k$-FWL test and FGNNs is straightforward, obtaining a characterization via signal-weighted homomorphism densities—derived using tri-labeled graphs (§ D.1)—would require slightly more effort.

## G CUT DISTANCE AND TRANSFERABILITY OF HIGHER-ORDER WNNS

### G.1 PROOF OF PROPOSITION 5.1

**Proposition 5.1.** *Let $\varrho : [0,1] \to \mathbb{R}$. Then, the assignment $\mathcal{W}_0 \ni W \mapsto \varrho(W) \in \mathcal{W}$, where $\varrho$ is applied pointwise, is continuous w.r.t. $\|\cdot\|_\square$ if and only if $\varrho$ is linear.*

*Proof.* We will show that the assignment

$$\mathcal{W}_0 \ni W \mapsto \varrho(W) \in \mathcal{W}, \tag{168}$$

where $\varrho$ is applied pointwise, is continuous if and only if $\varrho$ is linear. First, note that $\mathcal{W} \in W \mapsto \int_{[0,1]^2} W \, d\lambda^2$ is linear and continuous w.r.t. $\|\cdot\|_\square$, since

$$\left| \int_{[0,1]^2} W \, d\lambda^2 \right| \leq \sup_{S,T \subseteq [0,1]} \left| \int_{S \times T} W \, d\lambda^2 \right| = \|W\|_\square. \tag{169}$$

Let $\varrho : [0,1] \to \mathbb{R}$ such that $W \mapsto \varrho(W)$ is continuous. Then, also $W \mapsto \int_{[0,1]^2} \varrho(W) \, d\lambda^2$ is continuous. Let $p \in (0,1)$ and set $W_p := p$ to be a constant graphon. If we sample $G_p^{(n)} \sim \mathbb{G}_n(W_p)$, i.e., from an Erdős–Rényi model with edge probability $p$, $G_p^{(n)} \to W_p$ in the cut norm almost surely. But

$$\int_{[0,1]^2} \varrho\left(W_{G_p^{(n)}}\right) d\lambda^2 \to p \cdot \varrho(1) + (1-p) \cdot \varrho(0), \tag{170}$$

while $\int_{[0,1]^2} \varrho(W_p) \, d\lambda^2 = \varrho(p)$. This implies

$$\forall p \in (0,1) : \ \varrho(p) = p \cdot \varrho(1) + (1-p) \cdot \varrho(0), \tag{171}$$

i.e., $\varrho$ is a linear function. It is trivial to check that if $\varrho(x) = ax + b$ is a linear function, $W \mapsto \varrho(W)$ is indeed continuous. $\square$

### G.2 PROOF OF THEOREM 5.2

**Theorem 5.2** (Transferability). *Let $r > 1$. Let $\mathcal{N} : \widetilde{\mathcal{WS}}_r \to \mathbb{R}$ such that $\mathcal{N}$ is contained in the closure of*

$$\mathrm{span}\left\{t((F, \boldsymbol{d}), \cdot)\right\}_{F \ multigraph, \boldsymbol{d} \in \mathbb{N}_0^{v(F)}} \subseteq C_b(\widetilde{\mathcal{WS}}_r, \delta_1) \tag{16}$$

*w.r.t. uniform convergence. Then, for any $(W, f) \in \mathcal{WS}_r$ and $(G_n, \boldsymbol{f}_n), (G_m, \boldsymbol{f}_m) \sim \mathbb{G}_n(W, f), \mathbb{G}_m(W, f)$,*

$$\mathbb{E}\left|\mathcal{N}(G_n, \boldsymbol{f}_n) - \mathcal{N}(G_m, \boldsymbol{f}_m)\right| \to 0, \qquad n, m \to \infty. \tag{17}$$

*Proof.* Fix $\varepsilon > 0$. For such $\mathcal{N}$, there exists a linear combination of signal-weighted homomorphism densities, i.e., a finite collection of $\{\alpha_i\}_i \in \mathbb{R}$, multigraphs $\{F_i\}_i$, and exponents $\{\boldsymbol{d}_i\}_i$, $\boldsymbol{d}_i \in \mathbb{N}_0^{v(F_i)}$, such that

$$\left\| \mathcal{N} - \underbrace{\sum_i \alpha_i t((F_i, \boldsymbol{d}_i), \cdot)}_{=: \mathcal{N}_\varepsilon} \right\|_\infty \leq \varepsilon. \tag{172}$$

Set

$$\hat{\mathcal{N}}_\varepsilon(W, f) = \sum_i \alpha_i \cdot t(F_i^{\mathrm{simple}}, \boldsymbol{d}_i, (W, f)), \tag{173}$$

where $F_i \mapsto F_i^{\mathrm{simple}}$ removes parallel edges. $\hat{\mathcal{N}}_\varepsilon$ is Lipschitz continuous in the cut distance by § C.1 and, crucially, agrees with $\mathcal{N}_\varepsilon$ on $\{0,1\}$-valued graphons (since any monomial $x \mapsto x^d$ has 0 and 1 as fixed points), and, thus, on finite graph-signals. Let $M > 0$ be the $\delta_\square$-Lipschitz constant of $\hat{\mathcal{N}}_\varepsilon$.

Now, consider $(G_n, \boldsymbol{f}_n), (G_m, \boldsymbol{f}_m) \sim \mathbb{G}_n(W, f), \mathbb{G}_m(W, f)$. By the graphon-signal sampling lemma (Levie (2023); (4)), we can bound

$$\mathbb{E}\big[|\mathcal{N}(G_n, \boldsymbol{f}_n) - \mathcal{N}(G_m, \boldsymbol{f}_m)|\big] \tag{174}$$

$$\leq \underbrace{\mathbb{E}\big[|\mathcal{N}(G_n, \boldsymbol{f}_n) - \mathcal{N}_\varepsilon(G_n, \boldsymbol{f}_n)|\big]}_{\leq \varepsilon} + \mathbb{E}\big[|\hat{\mathcal{N}}_\varepsilon(G_n, \boldsymbol{f}_n) - \hat{\mathcal{N}}_\varepsilon(G_m, \boldsymbol{f}_m)|\big] \tag{175}$$

$$+ \underbrace{\mathbb{E}\big[|\mathcal{N}_\varepsilon(G_m, \boldsymbol{f}_m) - \mathcal{N}(G_m, \boldsymbol{f}_m)|\big]}_{\leq \varepsilon} \tag{176}$$

$$\leq 2\varepsilon + M \cdot \mathbb{E}\big[\delta_\square((G_n, \boldsymbol{f}_n), (G_m, \boldsymbol{f}_m))\big] \tag{177}$$

$$\leq 2\varepsilon + M \cdot \Big(\mathbb{E}\big[\delta_\square((G_n, \boldsymbol{f}_n), (W, f))\big] + \mathbb{E}\big[\delta_\square((W, f), (G_m, \boldsymbol{f}_m))\big]\Big) \tag{178}$$

$$\overset{(*)}{\leq} 2\varepsilon + 15M\left((\log n)^{-1/2} + (\log m)^{-1/2}\right), \tag{179}$$

where the sampling lemma was used in $(*)$. Hence,

$$0 \leq \limsup_{n,m\to\infty} \mathbb{E}\big[|\mathcal{N}(G_n, \boldsymbol{f}_n) - \mathcal{N}(G_m, \boldsymbol{f}_m)|\big] \leq 2\varepsilon \tag{180}$$

for all $\varepsilon > 0$, which completes the proof. $\qquad\square$

### G.3 PROOF OF COROLLARY 5.3

**Corollary 5.3** (Transferability of higher-order WNNs)**.** *The assumption of Theorem 5.2 holds for*

*(1) any IWN with continuous nonlinearity $\varrho$,*
*(2) any $\mathcal{N} : \widetilde{\mathcal{WS}}_r \to \mathbb{R}$ for which $\mathcal{N}(W, f) = \widetilde{\mathcal{N}}(\nu_{(W,f)}^{k\text{-WL}})$ for a continuous $\widetilde{\mathcal{N}} : \mathcal{P}(\mathbb{M}^k) \to \mathbb{R}$.*

*Proof.* **(1):** Let $\mathcal{N}^{\text{IWN}}$ be an IWN with nonlinearity $\varrho$, and fix $\varepsilon > 0$. Let $p : \mathbb{R} \to \mathbb{R}$ be a polynomial such that the IWN $\mathcal{N}_p^{\text{IWN}}$ which is obtained from $\mathcal{N}^{\text{IWN}}$ by replacing each occurrence of $\varrho$ with $p$ fulfills

$$\|\mathcal{N}_p^{\text{IWN}} - \mathcal{N}^{\text{IWN}}\|_\infty = \sup_{(W,f) \in \mathcal{WS}_r} |\mathcal{N}_p^{\text{IWN}}(W, f) - \mathcal{N}^{\text{IWN}}(W, f)| \leq \varepsilon. \tag{181}$$

Such a $p$ exists: We can approximate $\varrho : \mathbb{R} \to \mathbb{R}$ uniformly arbitrarily well on compact subsets of $\mathbb{R}$ by the standard Weierstrass theorem, and, as the domain of $\mathcal{N}^{\text{IWN}}$ only contains bounded functions $W$ and $f$, for *any* input $(W, f) \in \mathcal{WS}_r$, $\varrho$ is only ever considered on some fixed bounded set (which depends on the model parameters). The argument is the same as switching activation functions, as done on multiple occasions in § 4.3. Now, observe that $\mathcal{N}_p^{\text{IWN}}$ can be reduced to an integral over $[0, 1]^n$ of a polynomial in the variables $W(x_i, x_j)$ and $f(x_k)$, for $i, j, k \in [n]$ and some $n \in \mathbb{N}$. This implies that $\mathcal{N}_p^{\text{IWN}}$ is a linear combination of signal-weighted homomorphism densities.

**(2):** This follows immediately by combining Theorem D.7 with Proposition D.9, using that the algebra generated by functions of the form $\nu \mapsto \int f \, d\nu$ for $f \in C(\mathbb{M}^k)$ is dense in $C(\mathcal{P}(\mathbb{M}^k))$ by the Stone-Weierstrass theorem. Note that such a factorization specifically exists for $k$-WNNs by Theorem F.4. $\qquad\square$

### G.4 QUANTITATIVE TRANSFERABILITY RESULTS

**Theorem 5.4** (Quantitative transferability of IWNs, informal)**.** *Let $r > 1$. For any $\mathcal{N}^{\text{IWN}}$ in a universal class of 2-layer IWNs with real-analytic nonlinearity $\varrho$, there exists a constant $M_{\mathcal{N}^{\text{IWN}}} > 0$ such that for $(W, f) \in \mathcal{WS}_r$ and $(G_n, \boldsymbol{f}_n), (G_m, \boldsymbol{f}_m) \sim \mathbb{G}_n(W, f), \mathbb{G}_m(W, f)$ for large $n, m$,*

$$\mathbb{E}\,|\mathcal{N}^{\text{IWN}}(G_n, \boldsymbol{f}_n) - \mathcal{N}^{\text{IWN}}(G_m, \boldsymbol{f}_m)| \leq M_{\mathcal{N}^{\text{IWN}}}\left((\log n)^{-1/2} + (\log m)^{-1/2}\right). \tag{18}$$

For simplicity, we consider a simple two-layer IWN of the form

$$\mathcal{N}(W, f) = \int_{[0,1]^k} \varrho\left(\sum_{S \in \binom{[k]}{2}} a_S W(\boldsymbol{x}_S) + \sum_{t \in [k]} b_t f(x_t) + c\right) d\boldsymbol{x}, \tag{182}$$

where the input $(W, f) \in \mathcal{WS}_r$ is a graphon-signal, and $(a_S)_S, (b_t)_t, c$ are real-valued coefficients. Note that linear combinations of (182) of arbitrary order $k$ can distinguish any two graphon-signals that are not weakly isomorphic, simply by switching activation functions and showing that they can approximate any signal-weighted homomorphism density, akin to the arguments in § 4.3 and Keriven & Peyré (2019). This parametrization, however, is impractical, as it typically requires an intractable number of addends of the form (182), and is introduced mainly to illustrate the core idea of the argument. We note that

- $\mathcal{N}$ is generally *not* continuous in cut distance, as long as $\varrho$ is nonlinear (Proposition 5.1),
- Yet, $\mathcal{N}$ is still transferable (or *estimable*; see also Lovász (2012, § 15)), in the sense that $\mathcal{N}$ restricted to finite graphs is continuous in cut distance, and can be uniquely extended to a function $\hat{\mathcal{N}}$ which is cut distance continuous. Formally, from Theorem 5.2 we can conclude that

$$\left( \mathbb{E}_{(G_n, \boldsymbol{f}_n) \sim \mathbb{G}_n(W,f)}[\mathcal{N}(G_n, \boldsymbol{f}_n)] \right)_{n \in \mathbb{N}} \tag{183}$$

is Cauchy, and simply set

$$\hat{\mathcal{N}}(W, f) := \lim_{n \to \infty} \mathbb{E}_{(G_n, \boldsymbol{f}_n) \sim \mathbb{G}_n(W,f)}[\mathcal{N}(G_n, \boldsymbol{f}_n)]. \tag{184}$$

However, to make quantitative statements about the transferability of $\mathcal{N}$ when applied to graphs of different sizes, we would need a *Lipschitz* bound for $\hat{\mathcal{N}}$ in cut distance, similar as can be obtained for MPNNs. For multigraph homomorphism densities, it is straightforward to see that the cut distance continuous extension can be simply obtained by removing parallel edges. The latter admits a Lipschitz bound in cut norm/distance (by the graphon-signal counting lemma; see § C.1). As such, any graph parameter consisting of a (finite) linear combination of homomorphism densities is also Lipschitz in $\|\cdot\|_\square$. However more generally, for non-polynomial $\varrho$, (182) might depend on an infinite number of multigraph homomorphism densities. It is therefore not immediate to see that $\hat{\mathcal{N}}$ will generally still be Lipschitz continuous in cut norm/distance, even if $\mathcal{N}$ is w.r.t. $L^1$ (which holds as long as $\varrho$ is Lipschitz; see Lemma 4.5).

One general recipe to ensure that limits of Lipschitz functions stay Lipschitz is looking at their Lipschitz (semi-)norms: Let $\mathcal{X}$ be some metric space, and $(f_n)_n$ a sequence of real-valued Lipschitz functions on $\mathcal{X}$ such that $\|f_n\|_{\mathrm{Lip}} < \infty$ for all $n$. Supposing that $\sum_{n=1}^{\infty} f_n$ is defined on all of $\mathcal{X}$, one can obtain

$$\left\| \sum_{n=1}^{\infty} f_n \right\|_{\mathrm{Lip}} \leq \sum_{n=1}^{\infty} \|f_n\|_{\mathrm{Lip}}. \tag{185}$$

If the r.h.s. is still finite, this implies that the series $\sum_{n=1}^{\infty} f_n$ is still Lipschitz continuous as a function $\mathcal{X} \to \mathbb{R}$. To make bounding a potential Lipschitz constant of (182) tractable, we can try to write it as a series of homomorphism densities. For this, we assume that the nonlinearity $\varrho$ is real-analytic, i.e., it has a power series expansion $\varrho(x) = \sum_{\ell=0}^{\infty} \gamma_\ell x^\ell$ with convergence radius $R \in (0, \infty]$. While this is a much stronger assumption compared to Lipschitzness of $\varrho$ (on the compact domains of graphon-signal values we consider), it is one that is fulfilled by many common activation functions like Sigmoid, Softplus, Swish (each with $R = \pi$), or GELU ($R = \infty$). With the above definitions in place, we will now state a formal version of Theorem 5.4 and proceed with its proof.

**Theorem G.1** (Quantitative transferability of IWNs, formal). *Let $r > 1$ and $\mathcal{N}$ be an IWN of the form (182), with real-analytic nonlinearity $\varrho$ of convergence radius $R \in (0, \infty]$. If we have $r\|(\boldsymbol{a}, \boldsymbol{b}, c)\|_1 < R$ for the weights $(\boldsymbol{a}, \boldsymbol{b}, c)$ of $\mathcal{N}$, then there exists a constant $M_\mathcal{N} > 0$ such that for any two finite graph-signals $(G_1, \boldsymbol{f}_1), (G_2, \boldsymbol{f}_2) \in \mathcal{WS}_r$, identified with their step graphon-signals,*

$$|\mathcal{N}(G_1, \boldsymbol{f}_1) - \mathcal{N}(G_2, \boldsymbol{f}_2)| \leq M_\mathcal{N} \cdot \delta_\square((G_1, \boldsymbol{f}_1), (G_2, \boldsymbol{f}_2)). \tag{186}$$

The statement of Theorem 5.4 can be directly obtained from Theorem G.1, under the same conditions on the IWN and nonlinearity as well as $r > 1$, simply by taking expectations over random graphs and invoking the graphon-signal sampling lemma on the r.h.s. of (186). The resulting multiplicative constant is then the Lipschitz constant of the continuous IWN extension $\hat{\mathcal{N}}$, up to an absolute factor.

Note that, since Theorem 5.4 is merely stating Lipschitzness of $\mathcal{N}$ for finite graphs (or Lipschitzness of $\hat{\mathcal{N}}$ on $\widetilde{\mathcal{WS}_r}$), any other known bounds in cut distance can be directly applied to obtain quantitative rates for the statement of Theorem 5.4. Under sampling simple graphs $\mathbb{G}_n(W)$ from a stochastic block model with $k$ blocks, for example, a rate of $\mathcal{O}((\frac{k}{n \log k})^{1/2})$ can be achieved (Klopp & Verzelen, 2019), which significantly surpasses the generic rate of $\mathcal{O}((\log n)^{-1/2})$ typically given by the sampling lemmas. In fact, different discretization techniques have been used in the literature on convergence and transferability, such as grid sampling (Ruiz et al., 2020) or discretization of the shift operator (Le & Jegelka, 2023) (which is equivalent to grid averaging for graphons), typically under regularity assumptions like Lipschitzness and with rates of $\mathcal{O}(n^{-1/2})$ (these would, however, needed to be explicitly stated in cut distance to be applied to Theorem G.1).

*Proof of Theorem G.1.* Suppose we are given an IWN $\mathcal{N}$ as from (182) with real-analytic nonlinearity $\varrho$, and $\varrho$ is given by its series expansion. Given a graphon-signal $(W, f)$ and $\boldsymbol{x} \in [0,1]^k$, we can expand

$$
\varrho \left( \sum_{S \in \binom{[k]}{2}} a_S W(\boldsymbol{x}_S) + \sum_{t \in [k]} b_t f(x_t) + c \right) = \sum_{\ell=0}^{\infty} \gamma_\ell \left( \sum_{S \in \binom{[k]}{2}} a_S W(\boldsymbol{x}_S) + \sum_{t \in [k]} b_t f(x_t) + c \right)^\ell
\tag{187}
$$

$$
= \sum_{\ell=0}^{\infty} \gamma_\ell \sum_{m+n+p=\ell} \binom{\ell}{m,n,p} \left( \sum_S a_S W(\boldsymbol{x}_S) \right)^m \left( \sum_t b_t f(x_t) \right)^n c^p
\tag{188}
$$

$$
= \sum_{\ell=0}^{\infty} \gamma_\ell \sum_{m+n+p=\ell} \binom{\ell}{m,n,p} \sum_{\substack{S_1,\ldots,S_m \\ t_1,\ldots,t_n}} a_{S_1} \cdots a_{S_m} b_{t_1} \cdots b_{t_n} W(\boldsymbol{x}_{S_1}) \cdots W(\boldsymbol{x}_{S_m}) f(x_{t_1}) \cdots f(x_{t_n}) c^p
\tag{189}
$$

and (by absolute convergence),

$$
\mathcal{N}(W, f) = \sum_{\ell=0}^{\infty} \gamma_\ell \sum_{m+n+p=\ell} \binom{\ell}{m,n,p} \sum_{\substack{S_1,\ldots,S_m \\ t_1,\ldots,t_n}} a_{S_1} \cdots a_{S_m} b_{t_1} \cdots b_{t_n} c^p t((F_{S_1,\ldots,S_m}, \boldsymbol{d}_{t_1,\ldots,t_n}), (W, f)),
\tag{190}
$$

where $(F_{S_1,\ldots,S_m}, \boldsymbol{d}_{t_1,\ldots,t_n})$ is the multigraph on $k$ vertices with edges $S_1, \ldots, S_m$, and node features $(\boldsymbol{d}_{t_1,\ldots,t_n})_i := |\{t \in \{\!\{t_1, \ldots, t_n\}\!\} \mid t = i\}|$. With this expansion of $\mathcal{N}$ in (190), it is straightforward to see that the cut distance continuous extension

$$
\hat{\mathcal{N}}(W, f) = \sum_{\ell=0}^{\infty} \gamma_\ell \sum_{m+n+p=\ell} \binom{\ell}{m,n,p} \sum_{\substack{S_1,\ldots,S_m \\ t_1,\ldots,t_n}} a_{S_1} \cdots a_{S_m} b_{t_1} \cdots b_{t_n} c^p t((F^{\text{simple}}_{S_1,\ldots,S_m}, \boldsymbol{d}_{t_1,\ldots,t_n}), (W, f)),
\tag{191}
$$

can be simply obtained by removing parallel edges in all occuring multigraph homomorphism densities. Note that $e(F^{\text{simple}}_{S_1,\ldots,S_m}) \leq m$, and $D_{t_1,\ldots,t_n} := \sum_i (\boldsymbol{d}_{t_1,\ldots,t_n})_i = n$. By the graphon-signal counting lemma (§ C.1), we can bound

$$
\left\| t((F^{\text{simple}}_{S_1,\ldots,S_m}, \boldsymbol{d}_{t_1,\ldots,t_n}), (W, f)) \right\|_{\text{Lip}, \delta_\square}
\tag{192}
$$

$$
\leq \max\{4r^{D_{t_1,\ldots,t_n}} e(F^{\text{simple}}_{S_1,\ldots,S_m}), 2D_{t_1,\ldots,t_n} r^{D_{t_1,\ldots,t_n}-1}\}
\tag{193}
$$

$$
\leq \max\{4r^n m, 2nr^{n-1}\} \leq \max\{4\ell r^\ell, 2\ell r^{\ell-1}\} \overset{r>1}{\leq} 4\ell r^\ell,
\tag{194}
$$

and thus, applying (185),

$$\|\hat{\mathcal{N}}\|_{\text{Lip},\delta_\square} \leq \sum_{\ell=0}^{\infty} |\gamma_\ell| \, 4\ell r^\ell \bigg( \underbrace{\sum_S |a_S| + \sum_t |b_t| + |c|}_{=: \|(\boldsymbol{a},\boldsymbol{b},c)\|_1} \bigg)^\ell \tag{195}$$

$$= \sum_{\ell=0}^{\infty} |\gamma_\ell| \, 4\ell \, (r\|(\boldsymbol{a},\boldsymbol{b},c)\|_1))^\ell = 4r\|(\boldsymbol{a},\boldsymbol{b},c)\|_1 \cdot \tilde{\varrho}'(r\|(\boldsymbol{a},\boldsymbol{b},c)\|_1), \tag{196}$$

where $\tilde{\varrho}(x) := \sum_{\ell=0}^{\infty} |\gamma_\ell| \, x^\ell$ is the majorant of $\varrho$. This bound holds as long as

$$r\|(\boldsymbol{a},\boldsymbol{b},c)\|_1 \, < \, R \tag{197}$$

is within the radius of convergence of the series expansion of $\varrho$ (e.g., for GELU, this statement always holds). Hence, $\hat{\mathcal{N}}$ is Lipschitz w.r.t. $\delta_\square$. $\qquad\square$

A similar argument could plausibly be extended to multilayer IWNs; however, the resulting expansion quickly becomes unwieldy. The derived Lipschitz constant can blow up due to the dependence on $\tilde{\varrho}$, and is likely not tight. Note that by simply applying the triangle inequality to the Lipschitz norms, we lose any potential cancellation that might occur. Likewise, our estimate of the Lipschitz constants for the simple homomorphism densities—where, e.g., we ignore any tightening gained by dropping the parallel edges—is quite loose, chosen only to result in a simple final bound.

Indeed, one could show that under the assumptions of Theorem G.1,

$$\hat{\mathcal{N}}(W, f) \, = \, \mathbb{E}_{(G,\boldsymbol{f}) \sim \mathbb{G}_k(W,f)} \left[ \mathcal{N}(G, \boldsymbol{f}) \right], \tag{198}$$

that is, choosing $n = k$ in (184) already yields an unbiased estimator. This formulation could serve as a starting point for an alternative approach to deriving bounds on the transferability of $\mathcal{N}$: Disregarding the node signals, the difference of (198) for two different graphons $W, V$ can be bounded by a constant multiple (depending on $\varrho$ and the parameters $(\boldsymbol{a},\boldsymbol{b},c)$) of the *variation distance* between the random graph distributions $\mathbb{G}_k(W), \mathbb{G}_k(V)$, which in turn can be bounded by their cut distance (Lovász, 2012, § 10.1), though with an exponential dependence on $k$. To generalize such an argument to graphon-signals, one would have to define a suitable distance for random graph-signal distributions that is compatible with the graphon-signal cut distance. It appears that such a distance would have to metrize the weak topology on the probability measures over graph-signals, so the standard variation distance does not seem to work here. Further investigations of this approach and establishing tight and general bounds are left for future work.

### G.5 ADDITIONAL REMARKS

**Factorization of Higher-Order WNNs.** Since $\mathbb{P}^k \subset \mathbb{M}^k$ is closed, defining the factorization from Corollary 5.3 only on measures on $\mathbb{P}^k$ (see § D) would also suffice by the Tietze extension theorem. However, it is important to note that defining the extension *solely* on the $k$-WL measures $\{\nu_{(W,f)}^{k\text{-WL}}\}_{(W,f)}$ is a priori *not* enough, as we have not shown that this set is closed.

In other words, we can also look at these phenomena through the lens of different topologies on $\widetilde{\mathcal{WS}}_r$. The 1-WL test and MPNNs are continuous in the cut distance, which makes $\widetilde{\mathcal{WS}}_r$ a compact space, in which convergence is determined by signal-weighted homomorphism densities w.r.t. *simple* graphs (Corollary 3.3). For the $k$-WL test and higher-order graphon neural networks such as the ones we considered (IWNs, $k$-WNNs), we have seen that this does not hold. However, these models factorize as continuous functions on the *compact* space of $k$-WL colors $\mathbb{M}^k$, whose convergence is determined by *multigraph* signal-weighted homomorphism densities. Note that Böker (2023, § 4.7) combines all $k$-WL measures, $k > 1$, into a single object (which is again compact by Tychonoff's theorem), which describes precisely the topology of convergence of *all* multigraph homomorphism densities. A topology which is at least as fine is given by the $\{\delta_p\}_{p\in[1,\infty)}$ distances, as the multigraph homomorphism densities are continuous in these. This is illustrated in Figure 6.

We want to briefly elaborate on $(*)$ regarding relative compactness in the middle of Figure 6. For 1-WL via distributions of iterated degree measures defined on a color space $\mathbb{M}$, the assignment

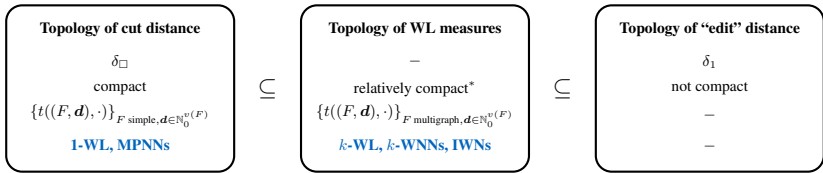

Figure 6: Relevant graphon-signal topologies.

$(W, f) \mapsto \nu_{(W,f)}$ is continuous in cut distance, and, as such, the above set can be interpreted as the continuous image of a compact set into a Hausdorff space. Hence, $\{\nu_{(W,f)}\}_{(W,f)} \subset \mathcal{P}(\mathbb{M})$ is again compact, and distinguishing between the two sets is not crucial, as remarked by Böker et al. (2023). Moreover, the pullback of this map yields a compact topology on the original graphon-signal space $\widetilde{\mathcal{WS}}_r$ which corresponds precisely to convergence of simple signal-weighted homomorphism densities w.r.t. trees. For $k$-WL this is slightly more subtle, as the map $(W, f) \mapsto \nu_{(W,f)}^{k\text{-WL}}$ is *not* continuous in $\delta_\square$ but only in $\delta_1$, and the space $(\widetilde{\mathcal{WS}}_r, \delta_1)$ is not compact. Therefore, one cannot apply the same argument to show that $\{\nu_{(W,f)}^{k\text{-WL}}\}_{(W,f)} \subset \mathcal{P}(\mathbb{M}^k)$ is closed.

The "natural" tool modeling IWNs and $k$-WL as continuous functions on a compact set would appear to be *probability graphons* (Lovász, 2012; Abraham et al., 2023), whose values are probability measures of the edge weights (i.e., $p$-Erdős–Rényi graphs would converge to a constant graphon of value $p\delta_1 + (1-p)\delta_0$, with $\delta_x$ indicating the Dirac measure at $x$ here).

**Simple $k$-WL (Böker, 2023).** Böker (2023) also proposes a *simple $k$-WL* test for graphons, which iteratively updates the colors using a pre-defined set of "shift-like" operators that explicitly parametrize *simple* homomorphism densities. They also remark that while simple vs. multigraph homomorphism densities yield the same notion of weak isomorphism (see also Theorem 3.2), simple $k$-WL is strictly weaker than $k$-WL for finite $k$. Analogous to § D and § F, it would be straightforward to extend the simple $k$-WL test to graphon-signals and to design higher-order WNNs based on the simple $k$-WL test which would be continuous in cut distance. We also note that the parametrization of operators given by Böker (2023, § 5.2) for simple $k$-FWL can be reduced from $(k+1)^2 2^k$ to $(k^2 + 1)2^k$ operators by resolving ambiguities. While both formulations are *asymptotically* equivalent, for small, feasible values of $k$, this would yield an improvement when implementing this method (20 vs. 36 operators to consider for $k = 2$). Yet, Böker (2023) contends that the definition of simple $k$-WL is rather complicated and "artificial". In this work, however, we argue that cut distance discontinuity can be *fixed* and is, indeed, of limited practical concern.

# H  EXPERIMENT DETAILS

In this section, we provide details for the toy experiments shown in Figure 1, Figure 2, and Figure 3. The purpose of these experiments is to empirically validate the findings from § 5 regarding the cut distance discontinuity and the transferability of IWNs.

## H.1  SETUP

**Data.** We keep the signal fixed at a constant value of 1 and look at the following 4 different graphons:

- **Erdős–Rényi (ER):** $W := 1/2$ constant.
- **Stochastic Block Model (SBM):** We take 5 blocks, each with intra-cluster edge probabilities of $p = 0.8$ and inter-cluster edge probabilites $q = 0.3$.
- **Triangular:** Here, $W(x, y) = (x + y)/2$ (the sets $\{W \geq z\}, z \in [0, 1]$ are triangles).
- **Narrow:** $W(x, y) = \exp\left(\sin\left(\frac{(x-y)^2}{\gamma}\right)^2\right)$, with $\gamma := 0.05$.

From each of these graphons, we sample 100 simple graphs of each of the sizes $\{200, 400, 600, 800, 1000\}$. We use a *weighted* graph of 1000 nodes sampled from each graphon as an approximation of the respective graphon itself (convergence of weighted graphs is typically much faster than of simple graphs).

**Models.** The following two models are compared:

- **MPNN**: A standard MPNN with mean aggregation over the *entire* node set, as used in the analyses of Levie (2023); Böker et al. (2023).
- **2-IWN**: An IWN of order 2, i.e., with a basis dimension of 7.

For both models, we use a simple setup of 2 layers, a hidden dimension of 16, and the sigmoid function as activation.

**Experiment.** In Figure 2, we plot the absolute errors of the model outputs for the sampled simple graphs in comparison to their graphon limits. Due to the cut distance continuity of MPNNs and the sampling lemma (Levie, 2023, Theorem 4.3), the MPNN outputs decrease as the graph size grows. While the convergence is slow, it is still significantly faster than the worst-case bound. This is expected as the considered graphons are fairly regular, and the convergence rates can be improved under additional regularity assumptions (Le & Jegelka, 2023; Ruiz et al., 2023). The IWN, however, is discontinuous in the cut distance (Proposition 5.1) and, as such, the errors do not decrease. In Figure 3, we further plot the difference between the 0.95 and 0.05 quantiles of the output distributions on simple graphs for each of the considered sizes. Notably, there are only minor differences visible between the MPNN and the IWN. This validates Theorem 5.2 and suggests that IWNs can have similar transferability properties as MPNNs (also beyond the worst-case bound), and $\delta_p$-continuity suffices for transferability.

**Code.** The code used for the toy experiment is provided under https://github.com/dan1elherbst/Higher-Order-WNNs.

## H.2 PSEUDOCODE OF MODELS

### H.2.1 MPNN

---
**Algorithm 1** MPNN forward pass
---
1: **procedure** $\mathcal{N}^{\text{MPNN}}(\boldsymbol{W} \in [0,1]^{n \times n}, \boldsymbol{X} \in \mathbb{R}^{n \times d_0})$
2: $\quad \boldsymbol{H} \leftarrow \boldsymbol{X}$
3: $\quad$ **for** $s \leftarrow 1$ to $S$ **do**
4: $\quad\quad \boldsymbol{H} \leftarrow 1/n \cdot \boldsymbol{W} \cdot \boldsymbol{H}$
5: $\quad\quad \boldsymbol{H} \leftarrow \text{LINEAR}^{(s)}_{d_{s-1} \rightarrow d_s}(\boldsymbol{H})$
6: $\quad\quad$ **if** $s < S$ **then**
7: $\quad\quad\quad \boldsymbol{H} \leftarrow \varrho(\boldsymbol{H})$
8: $\quad\quad$ **end if**
9: $\quad$ **end for**
10: $\quad$ **return** $1/n \cdot \sum_{i=1}^{n} \boldsymbol{H}_{i*}$
11: **end procedure**

---

### H.2.2 2-IWN

---
**Algorithm 2** 2-IWN forward pass
---
1: **procedure** $\mathcal{N}^{\text{2-IWN}}(\boldsymbol{W} \in [0,1]^{n \times n}, \boldsymbol{X} \in \mathbb{R}^{n \times d_0})$
2: $\quad \boldsymbol{H} \leftarrow \text{STACK}(\boldsymbol{W}_{**\varnothing}, \boldsymbol{X}_{*\varnothing*}) \in \mathbb{R}^{n^2 \times d_0}$
3: $\quad$ **for** $s \leftarrow 1$ to $S$ **do**
4: $\quad\quad \boldsymbol{H} \leftarrow \text{2-IWN-LINEAR}^{(s)}_{d_{s-1} \rightarrow d_s}(\boldsymbol{H})$
5: $\quad\quad$ **if** $s < S$ **then**
6: $\quad\quad\quad \boldsymbol{H} \leftarrow \varrho(\boldsymbol{H})$
7: $\quad\quad$ **end if**
8: $\quad$ **end for**
9: $\quad$ **return** $1/n^2 \cdot \sum_{i,j=1}^{n} \boldsymbol{H}_{ij*}$
10: **end procedure**

---

---
**Algorithm 3** 2-IWN linear operators
---
1: **procedure** $\text{2-IWN-LINEAR}^{(s)}_{d_{s-1} \rightarrow d_s}(\boldsymbol{H} \in \mathbb{R}^{n^2 \times d_{s-1}})$
2: $\quad$ # operators in $\mathsf{LE}_{2 \rightarrow 2}$
3: $\quad \boldsymbol{H}_1 \leftarrow 1/n \cdot \text{EINSUM}(\text{'ijd->id'}, \boldsymbol{H})_{*\varnothing*}$
4: $\quad \boldsymbol{H}_2 \leftarrow 1/n \cdot \text{EINSUM}(\text{'ijd->id'}, \boldsymbol{H})_{\varnothing**}$
5: $\quad \boldsymbol{H}_3 \leftarrow 1/n \cdot \text{EINSUM}(\text{'ijd->jd'}, \boldsymbol{H})_{*\varnothing*}$
6: $\quad \boldsymbol{H}_4 \leftarrow 1/n \cdot \text{EINSUM}(\text{'ijd->jd'}, \boldsymbol{H})_{\varnothing**}$
7: $\quad \boldsymbol{H}_5 \leftarrow \text{EINSUM}(\text{'ijd->ijd'}, \boldsymbol{H})$
8: $\quad \boldsymbol{H}_6 \leftarrow \text{EINSUM}(\text{'ijd->jid'}, \boldsymbol{H})$
9: $\quad \boldsymbol{H}_7 \leftarrow 1/n^2 \cdot \text{EINSUM}(\text{'ijd->d'}, \boldsymbol{H})_{\varnothing\varnothing*}$
10: $\quad$ # combine operators and return linear
11: $\quad \boldsymbol{H} \leftarrow \text{STACK}(\boldsymbol{H}_1, \ldots, \boldsymbol{H}_7) \in \mathbb{R}^{n^2 \times 7d_{s-1}}$
12: $\quad$ **return** $\text{LINEAR}^{(s)}_{7d_{s-1} \rightarrow d_s}(\boldsymbol{H})$
13: **end procedure**

---

