# OpenReview forum: "Higher-Order Graphon Neural Networks: Approximation and Cut Distance"
_ICLR.cc/2025/Conference — ICLR 2025 Spotlight_

### Official Review · Reviewer_p53n · 2024-10-28

**Soundness:** 4
**Presentation:** 2
**Contribution:** 3
**Rating:** 8
**Confidence:** 3

**Summary:**

This paper proposes the Invariant Graphon Network (IWN) architecture for graphons as an extension of the Invariant Graph Network GNN architecture for discrete graphs. The paper's results are framed as a more natural extension and generalization of the related work of Cai & Wong 2022. The authors provide an overview of the relevant graphon theory and an in-depth introduction to a novel signal-weighted homomorphism density concept, providing a comprehensive characterization of weak isomorphism of graphon-signals using this definition, as well as showing a convergence equivalence to cut distance convergence. A generalization of linear equivariant layers is then introduced, through measure-preserving functions rather than permutations in the discrete case. Several central theorems are shown about this function space, including invariance under discretization, construction of a canonical basis, and analysis of dimension. The IWN architecture is then built and proven to be an instance of an IGN-small network. Continuity with regard to $\delta_p$ is then shown, as well as lack of continuity for $\delta_{\square}$ unless the activation function is linear and thus expressivity is lost. Finally, expressivity of IWNs are shown in the sense that order $k+1$ IWNs are able to distinguish with the strength of $k$-WL for graphons.

**Strengths:**

- There is very thorough analysis of relation to previous work throughout this paper. This work's placement in the literature is clear.
- Original contributions are outlined clearly and presented in a cohesive framework.
- The ideas and results are very original, with several novel concepts defined. The definitions are shown to make sense through the strength of the theory, providing good evidence that these ideas are pointing toward a very useful framework.
- The proof sketch narration is generally clear, and the full proofs in the appendices are nicely written. It is clear there was a good deal of care taken toward ensuring the quality of presentation for these main results.
- In relation to the cited related works, the results shown in this paper are a significant contribution to the community.

**Weaknesses:**

- The discussion of the $\delta_{\square}$ discontinuity limiting transferability seems like it could be more complete. Especially as the definitions leading to this point are novel, it seems important to justify this not being a product of a definitional choice that could be reworked and rather a facet of the true underlying problem. Reading the appendix helped clear up my confusions here, but it would be ideal for the section itself to contain just a bit more of that insight.
- Due to the heavy theoretical nature, this paper requires a very high degree of engagement to fully understand the stated results. It would be much easier to digest the main ideas on an initial reading with more time, even a short section, dedicated to showcasing some of the concepts you develop, perhaps in some experiments. While convergence experiments such as those in Cai & Wong 2022 are not relevant here, I think there are great possibilities for a showcase of the continuity result and the expressivity result which would really elevate the overall impact of the paper.

One small typo:
- In Section 6, Paragraph 1, the phrase "...that IWNs, as a subset their of IGN-small, retain..." appears. Removing the "their" would fix this.

**Questions:**

- You state that IWNs being only continuous for the weaker $\delta_p$ distances, combined with the lack of compactness, prevent the approach from being applied to the entire space. Is there anything interesting to be said about limited regions of this space under the $\delta_p$ distance? Is there any hope for an interesting bound in the $\delta_{\square}$ distance regardless of discontinuity, or is the behavior fundamentally unworkable?
- Have you attempted to build some toy examples of the model, even without any sort of training?

---

> ### Author Response · Authors · 2024-11-21
> **Official Comment by Authors**
>
> We thank the reviewer for their insightful remarks and positive feedback on our work!
>
> > not being a product of a definitional choice that could be reworked and rather a facet of the true underlying problem
> >
>
> This is an excellent point! In the revision, we have clarified this and highlighted the close connection to $k$-WL. Please refer to the general comments for the full answer to this question.
>
> > short section, dedicated to showcasing some of the concepts you develop, perhaps in some experiments
>
> Refer to general comments.
>
> > Is there any hope for an interesting bound in the distance regardless of discontinuity, or is the behavior fundamentally unworkable?
>
> Thank you for raising this interesting question! Indeed, there is a strong indication that IWNs might be able to approximate sufficiently well-behaved functions on the entire space, and not just compact subsets (which are tiny compared to the entire graphon-signal space in the topology induced by $\delta_p$). Note that we can show such a result for signal-weighted homomorphism densities (Theorem 5.2), which are *bounded* continuous functions. Nevertheless, since the $\delta_p$-topologies are not compact, there might be unbounded continuous functions which can *impossibly* be approximated on the entire space by IWNs. A conjecture would be that all bounded $\delta_p$-continuous functions can be approximated arbitrarily well. While we have not checked this carefully, we agree this would also be an interesting further result.
>
> > toy examples of the model
>
> Refer to general comments.

---

> > ### Comment · Reviewer_p53n · 2024-11-27
> > **Increased Score**
> >
> > Thank you for the careful revision of the work and for your responses. I believe many of my concerns have been suitably addressed, and have raised my score.

---

> > > ### Author Response · Authors · 2024-11-28
> > > **Response by Authors**
> > >
> > > Thank you for taking the time to carefully review our work and for your thoughtful feedback. We are glad to hear that our comments and the revision have addressed your concerns. Should there be any remaining points where we can provide further clarification, feel free to let us know.

---

### Official Review · Reviewer_cdhd · 2024-10-29

**Soundness:** 3
**Presentation:** 4
**Contribution:** 3
**Rating:** 8
**Confidence:** 3

**Summary:**

This paper studied Invariant Graphon Networks, a generalization of Invariant Graph Networks (IGN) from Maron (2018) to graphons. The authors show that Invariant Graphon Networks can be characterized using a subset of the IGN basis functions, due to additional discretization invariance. The authors proved that Invariant Graphon Networks of order $k+1$ are at least as powerful as the $k$-dimensional Weisfeiler-Leman test for graphons proposed in Böker (2023). These expressivity results extend the convergence results in Cai and Wang (2022), by making use of signal-weighted homomorphism densities as a new tool for the analysis. The authors show that Invariant Graphon Networks are typically discontinuous with respect to cut distance, in contrast to the continuity results of MPNNs defined on graphons (Levie 2023). This implies a negative result on the size transferability of IGNs.

**Strengths:**

1. The paper makes novel contributions on extending IGNs to graphons and studying their expressivity and continuity properties.

2 .The paper is clearly written and well motivated.

3. The notion of signal-weighted homomorphism densitiy is an interesting combination of the graphon-signal space in Levie (2023) and homomorphism densities in Böker (2023).

**Weaknesses:**

Practical consequences: The authors remark that since Invariant Graphon Networks (IWNs) are discontinuous with respect to the cut distance, IWNs can assign vastly different values to similar graph inputs (line 96). It will be nice to substantiate this remark using empirical evidence, either through observation made in existing works using IGNs or original experiment studies.

**Questions:**

1. The definition of signal-weighted homomorphism densities (Defn 3.1) seems standard from the graphon literature, see for example homomorphism density on weighted graphs per equation 5-6 in [1], or various notions of homomorphism densities on multigraphs per chapter 17 in [2]. Can the authors remark the connections to these existing definitions?

2. Invariance under discretization: As stated in Lemma 4.3 and Lemma D.1, invariance under discretization implies the output $L(U)$ is constant almost everywhere. This also leads to a much smaller dimension of the space of linear equivariant layers in Theorem 4.2. Does it make sense to consider relaxing the notion of invariance under discretization (i.e., approximate invariance). Will the authors expect this leads to stronger expressivity results than those stated in Section 5?

3. Nontrivial IWN in Proposition 5.1: Proposition 5.1 (2) states that IWN is continuous with respect to the cut-distance if and only if the pointwise acitivation function is a linear function, where the authors treat it as "trivial" (Line 444). However, such linear IWN may not be trivial if the intermediate layers use higher-order tensors. Can the authors clarify this?

4. The effect of pooling: In the current definition of Invariant/Equivariant Graphon Networks (Defn 4.4), the network consists of linear equivariant maps interleaving with pointwise nonlinearity. In the original works by Maron et al. [3], [4], an invariant pooling operation is often used in practice. Can the authors comment on the role of pooling on the continuity property of IGNs?


References:

[1] Borgs, Christian, et al. "Counting graph homomorphisms." Topics in Discrete Mathematics: Dedicated to Jarik Nešetřil on the Occasion of his 60th Birthday. Berlin, Heidelberg: Springer Berlin Heidelberg, 2006. 315-371.

[2] Lovász, László. Large networks and graph limits. Vol. 60. American Mathematical Soc., 2012.

[3] Maron, Haggai, et al. "Invariant and Equivariant Graph Networks." International Conference on Learning Representations. 2018.

[4] Maron, Haggai, et al. "Provably powerful graph networks." Advances in neural information processing systems 32, 2019.

---

> ### Author Response · Authors · 2024-11-21
> **Official Comment by Authors**
>
> We thank the reviewer for their insightful remarks and positive feedback on our work!
>
> > It will be nice to substantiate this remark using empirical evidence, either through observation made in existing works using IGNs or original experiment studies.
>
> Please refer to the general comments.
>
> > Definition of signal-weighted homomorphism densities (Defn 3.1) seems standard from the graphon literature.
>
> We thank the reviewer for this comment. Indeed, the references [1] and [2] mentioned by the reviewer also define weighted graph homomorphism counts for node-featured graphs that would extend to our definition for graphons, which we were not fully aware of before the paper submission. Yet, to the best of our knowledge, there are currently no works that define and systematically analyze this concept for node-featured *graphons*, which can capture (a) weak isomorphism (Theorem 3.3) or (b) the k-WL test (Theorem 3.5) specifically under the topology that arises from the graphon-signal space [5]. We also note that there are numerous concepts of homomorphism counts used in the GNN literature, and, e.g., the definitions of [6] do not work for us as (a) not considering the signal moments only distinguishes graphs after twin reduction (e.g., signal values of zero can be removed), and (b) considering the same function for each node would result in the signal-weighted homomorphism densities not being closed under multiplication, which is essential for our proofs. As such, we see the task of defining a suitable concept as nontrivial. In the revision, we have made the connection to existing concepts clearer (cf. L219, L231-235).
>
> > Invariance under discretization
>
> We are not entirely sure that we fully understand the reviewer’s question. Invariance under discretization, intuitively meaning that (higher-order) step kernels are again mapped to step kernels w.r.t. the same regular partition of $[0,1]$, is not a condition that we impose on the network, but rather a *consequence* of framing IWNs as bounded linear operators—which is a reasonable assumption for any decently well-behaved network on graphons for theoretical analyses. Could the reviewer kindly elaborate on their question if our response does not fully address their concern?
>
> > Nontrivial IWN in Proposition 5.1
>
> In the mentioned proposition (now Proposition 5.5 in the revision), we precisely state that $W \mapsto \varrho(W)$ for any graphon $W$ is discontinuous in the cut norm, i.e., we do *not* consider an entire IWN, but only the pointwise application of the nonlinearity $\varrho$ (the integral of this would be a very simple IWN). If $\varrho$ is linear, this assignment is just a rescaling of $W$ and the addition of a constant, which is clearly cut distance continuous (as a homogenous function up to the additive constant). Even if one considers higher-order IWNs with a linear activation, one can derive that any parametrization (irrespective of the IWN’s order) will collapse to a function of the form  $W \mapsto a (\int_{[0,1]^2} W \mathrm{d}\lambda^2) + b$, which is clearly cut distance continuous, but of trivial expressivity.
>
> > The effect of pooling
>
> In our analysis, the pooling is implicitly defined by the last layer which maps to scalars, and, as such, is fixed to the mean (or integral for graphons). This is indeed crucial for our analysis, as other pooling operations such as $\mathrm{max}$ seem to not go well with the graphon space: One of the simplest such models would be essentially $W \mapsto \||W\||_\infty$, which is not continuous w.r.t. any of the $\delta_p$ distances, $p < \infty$. Therefore, we do not expect our analysis to extend to such a case. We kindly ask the reviewer to let us know should the question not be fully addressed.
>
> **References**
>
> [5] Ron Levie. A graphon-signal analysis of graph neural networks. Advances in Neural Information Processing Systems, 36:64482–64525, December 2023.
>
> [6]  Hoang Nguyen and Takanori Maehara. Graph Homomorphism Convolution. In Proceedings of the 37th International Conference on Machine Learning, pp. 7306–7316. PMLR, November 2020.

---

> > ### Comment · Reviewer_cdhd · 2024-11-24
> > **Clarification of Invariance under discretization/ follow-ups**
> >
> > I thank the authors for their detailed explanations. To clarify my question on invariance under discretization:
> > - Combining Theorem 4.2 and Lemma 4.3, we can see that the number of basis satisfying both permutation invariance and discretization invariance is less than the number of basis satisfying only permutation invariance (eqn 13).
> > - If one wants to retain more number of basis, does it make sense to consider approximate invariance under discretization?
> >
> > Relating to the general comment posted by the authors, I also wonder whether considering approximate invariance (permutation and/or discretization) provides another way to fix the cut-distance discontinuity.
> > I understand this is beyond the scope of the current work, but having some discussions on this may provide another interesting direction for future work.

---

> ### Author Response · Authors · 2024-11-25
> **Response by Authors**
>
> We thank the reviewer for elaborating on their question.
>
> Indeed, the dimension of the space of layers from Theorem 4.2/Eq. 15 satisfying both permutation and discretization invariance (or $\delta_p$-continuity) is smaller than the original IGN layer dimension of $\mathrm{bell}(k+\ell)$. In general, when designing such higher-order GNNs, the goal is to have a small and easily parametrizable basis. As we show that the basis from Theorem 4.2 achieves the *same* expressivity via the $k$-WL hierarchy (which also holds for finite graphs), we do not see any reason to keep the basis larger than necessary. Also, keeping the basis larger (and, e.g., discarding invariance under discretization) would not have any positive effect on the continuity properties of an IWN, as the resulting hypothesis class would just become a strict superset, still containing the same cut distance discontinuous functions (simply by setting the parameters of the added basis elements to zero). Even worse, when adding other basis elements for which continuity/invariance under discretization is not fulfilled, as mentioned in the introduction (L74-77) and further hinted at in § 4.1 (L326-327, L347-349), the resulting functions would not even be *well-defined* on the graphon-signal space, and, as such, impossible to analyze (which is why Cai & Wang (2022) introduced the “partition norm” vector). We also want to emphasize that the cut distance discontinuity does *not* arise from the linear layers, but from the pointwise application of the nonlinearity (cf. Proposition 5.5 in the updated script).
>
> Nevertheless, we agree that fixing the cut distance discontinuity and further examining its consequences would be an interesting avenue for further work (which we also mentioned in the conclusion, L535-537). This would entail developing a $k$-WL expressive architecture for which the edge weights are similarly processed as in the 1-WL, i.e., only acting through integral operators and not already being explicitly passed already in the first step of the test. With the IWN/IGN architecture, due to the pointwise application of nonlinearities, something like this is fundamentally unworkable (meaning that *no* further restriction of the basis would yield such a parametrization). As also mentioned in L536, Böker (2023) defines the *simple* $k$-WL test for this, which could potentially be used to design a higher-order *cut distance continuous* architecture (however, the parametrization would become slightly more complicated). Yet, in light of our result of the cut distance discontinuity’s fixability for transferability purposes, it is questionable if such a parametrization would yield meaningful benefits (what would be imaginable are better generalization guarantees, due to compactness of the graphon-signal space w.r.t. cut distance, as well as the possibility of extensions to sparse graph limits).

---

### Official Review · Reviewer_1FQG · 2024-11-01

**Soundness:** 4
**Presentation:** 4
**Contribution:** 3
**Rating:** 8
**Confidence:** 4

**Summary:**

This work proposes Invariant Graphon Networks (IWNs), which is an extension of Invariant Graph Networks (IGNs) defined on discrete graphs, to *graphons*. They study its separation power in relation to the $k$-WL test and its approximation capability for functions defined on graphons with vertex features.

They extend the notion of homomorphism densities (which, in turn, is an extension of *homomorphism counting*) from plain graphons to graphons with vertex features, and show that this notion can be used to determine the $k$-WL separation power of graphon neural networks. They then use it to show that their construction yields architectures at least powerful as the $k$-WL test, which in turn implies that it has universal approximation power of functions defined on vertex-featured fraphons.

Notably, their construction is more economic than that of the discrete IGN counterpart, due to a reduced basis for linear layers, leading to slower asymptotic growth in the number of basis elements required.

Finally, they expose a fundamental limitation of IWNs, and thus of IGNs, in that they are discontinuous with respect to the cut distance, which limits their applicability, and in particular their generalization capability, in certain learning scenarios. This is unlike MPNNs, which are continuous w.r.t. the cut distance.

**Strengths:**

I believe this work offers significant contribution to the community, in that it provides useful theoretical tools to analyze neural networks on vertex-featured graphons, and these tools are already shown useful in this work in their universal approximation and their discontinuity results. The theoretical analysis of IWNs' separation power with the $k$-WL test could guide future architectural innovations or alternative GNN designs on graphons.

**Weaknesses:**

The paper does not provide numerical experiments. Nonetheless, it provides valuable theoretical contribution that can potentially explain empirical results, motivate the design of architectures and also lead to new theoretical insights on GNNs. While numerical experiments could have provided practical insights into the theoretical results, the theoretical results themselves have high potential for practical impact.

**Questions:**

### Minor comments
1. Ln. 140-141: "Note that $\delta_{\square} \leq \delta_p$." - Does it follow from the L1-Lp inequality? This could benefit a short explanation or a citation.
2. Ln. 151: "If simple edges are further sampled..." - unclear definition of G(W,X).
3. Eq. (3), (4): Perhaps missing "as $n \to \infty$".


### Suggestions
1. Ln. 135-141: Some readers may not be familiar with the concept of cut distance. A brief explanation, perhaps as a short note or in a footnote, could benefit readers as well as help highlight your results on discontinuity.
2. An intuitive explanation for the cut-distance discontinuity, possibly with an illustrative example, could help comprehension.
3. Ln. 314 and the subsequent paragraph: To reassure the reader, I recommend stating explicitly that the dimension does not depend on $n$, and that the following theorem shows that this independence carries through to the case of graphons.

---

> ### Author Response · Authors · 2024-11-21
> **Official Comment by Authors**
>
> We thank the reviewer for their valuable suggestions and positive feedback on our work!
>
> > The paper does not provide numerical experiments.
>
> We added some proof-of-concept experiments; please refer to the general comments.
>
> > Minor comments:
>
> 1. It is a standard fact that the cut norm can be bounded by the $\mathcal{L}^1$ norm, and the inequality for arbitrary $p$ indeed follows straightforwardly (specifically for $[0,1]$).
> 2. $\mathbb{G}(W, \mathbf{X})$ is the definition of the simple random graph obtained by sampling edges $e_{ij}$ further from $W(\mathbf{X})$ via $e_{ij} \sim \mathrm{Bernoulli}(W(X_i, X_j))$.
> 3. Thanks for spotting this!
>
> > Suggestions
>
> 1. We have attempted to make this a bit clearer, and added a reference which explains the concept in more detail.
> 2. Please see the general comments, § 5.2, and Figures 1-3.
> 3. We agree that this adds to the readability, and have changed the sentence accordingly.
>
> We thank the reviewer for their helpful comments and suggestions, and we have addressed these points in the revision of the script.

---

### Official Review · Reviewer_fxhj · 2024-11-04

**Soundness:** 3
**Presentation:** 3
**Contribution:** 3
**Rating:** 8
**Confidence:** 3

**Summary:**

The paper extends IGNs to the graphon-signal space by introducing Invariant Graphon Networks (IWNs). It provides a generalization of linear equivariant layers to measure spaces, specifically the unit interval [0,1]. The authors introduce signal-weighted homomorphism densities to extend the concept of homomorphism densities to graphon-signals. They prove expressivity results for IWNs, showing that IWNs of order $k+1$ are at least as expressive as $k$-WL for graphons and that they are universal approximators on any compact subset of graphon-signals. This extends the standard results from Maron et al. from graphs to graphons. The paper also discusses the continuity properties of IWNs, highlighting that they are discontinuous wrt the cut distance.

**Strengths:**

- The paper seems sound and the theoretical results extend the results from Cai & Wang (2022), Böker et al. (2023), and Levie (2023)
- The authors connect their work to previous studies, such as those by Cai & Wang (2022) and Levie (2023), situating their contributions within the broader context of graph(on) neural networks

**Weaknesses:**

- It's not clear what the consequences for ML on graphs are
- The non-continuity of IWNs wrt to sampling is mentioned and proved, but it is not clear what consequences it has. Also, the authors mention it as a rather negative point, however, this is not clear to me. Do you have an example where this property hinders size generalization? Or is it rather a worst-case property that doesn't affect IGNs in practice?

**Questions:**

- Could you discuss your non-continuity results better? What do you think are the consequences?
- Also: how is it possible that the IWNs from Cai & Wang (2022) preserve convergence when sampling graphs from (sufficiently regular) graphon, but yours doesn't? Cai & Wang (2022)  also consider non-linearities, so is the reason that you consider graphon-signal?
- It appears that many results from Böker et al. (2023) and Levie (2023) were reproven using the same techniques for graphon-signal or IWNs. Could the authors discuss their technical contributions.


## Additional Comments
While the theoretical results appear to be correct (I didn't check all proofs) and generalize some results on graphon neural networks, it is not clear what the broader impact or significance is. I believe answering some of my questions would help to understand this as a reader. I am happy to increase my score if that is done.

---

> ### Author Response · Authors · 2024-11-21
> **Official Comment by Authors**
>
> We highly appreciate the reviewer’s thoughtful feedback and their positive remarks on our work. See below for detailed answers to the concerns and questions raised.
>
> > It's not clear what the consequences for ML on graphs are. […] Do you have an example where this property hinders size generalization?
>
> Please refer to the general comments.
>
> > IWNs from Cai & Wang (2022) preserve convergence when sampling graphs from (sufficiently regular) graphon, but yours doesn't?
>
> The IGNs as framed by Cai & Wang in [1] have similar convergence properties as IWNs: Cai & Wang show that IGNs applied to *weighted* graphs sampled from a graphon (”edge probability continuous model”) converge, but IGNs applied to *unweighted* graphs (”edge probability discrete model”, by [1] demonstrated through the model from Keriven et al. in [2]) do not. While the former only requires continuity of the model in $\delta_p$ distances, the latter would require continuity in the cut distance $\delta_\square$ (see, e.g., Eqs. (3) and (4)).
>
> > It appears that many results from Böker et al. (2023) and Levie (2023) were reproven using the same techniques for graphon-signal or IWNs.
>
> While the results of Böker [3] and Levie [4] are indeed foundational to our work, we did not reprove any results and we are convinced that our technical contributions go far beyond a straightforward adaptation of their theory to graphon-signals. One of our technical contributions is the introduction of signal-weighted homomorphism densities in § 3: For finite node-featured graphs there are various notions of homomorphism counts, such as enforcing that homomorphisms are stable w.r.t. node features, or different forms of weighting—for graphon analyses, however, node features have been mostly ignored. The trivial approach does not extend to graphon-signals, which can have node features that just attain a single value on null sets. While proof techniques do overlap with [3] and [5], originally finding a notion of weighted homomorphism that can be used for graphon-signals to elegantly capture (a) weak isomorphism (Theorem 3.3) or (b) the k-WL test (Theorem 3.5) is not straightforward, and was also mentioned as a challenge for future analyses in [3]. Our treatment of IWNs and linear equivariant layers, specifically the definition for general measure spaces and the derivation of a canonical basis in section 4, has a priori little to do with the aforementioned works and makes use of very different techniques, such as the representation stability argument for Theorem 4.2.
>
> > It is not clear what the broader impact or significance is
>
> Please refer to the general comments.
>
>
>
>
> **References**
>
> [1] Chen Cai and Yusu Wang. Convergence of Invariant Graph Networks. In Proceedings of the 39th International Conference on Machine Learning, pp. 2457–2484. PMLR, June 2022.
>
> [2] Nicolas Keriven, Alberto Bietti, and Samuel Vaiter. Convergence and stability of graph convolutional networks on large random graphs. Advances in Neural Information Processing Systems, 33:21512–21523, 2020.
>
> [3] Jan Böker, Ron Levie, Ningyuan Huang, Soledad Villar, and Christopher Morris. Fine-grained Expressivity of Graph Neural Networks. Advances in Neural Information Processing Systems, 36:46658–46700, December 2023.
>
> [4] Ron Levie. A graphon-signal analysis of graph neural networks. Advances in Neural Information Processing Systems, 36:64482–64525, December 2023.
>
> [5] László Lovász. Large Networks and Graph Limits. American Mathematical Soc., 2012.

---

> > ### Comment · Reviewer_fxhj · 2024-12-02
> >
> > Dear authors,
> >
> > thank you for the response and answering my questions. I decided to raise my score.
> >
> > Best,

---

### Official Review · Reviewer_MDWF · 2024-11-06

**Soundness:** 3
**Presentation:** 4
**Contribution:** 3
**Rating:** 8
**Confidence:** 1

**Summary:**

This paper introduces Invariant Graphon Networks (IWNs), an extension of Invariant Graph Networks (IGNs), to analyze graphons (limit objects of dense graphs) and investigate their expressivity and continuity properties. IWNs leverage a subset of the IGN framework focused on bounded linear operators, allowing them to approximate graphon functions with an expressivity at least as powerful as the k-Weisfeiler-Leman (k-WL) test.

**Strengths:**

This paper introduces Invariant Graphon Networks (IWNs) as an extension of Invariant Graph Networks (IGNs) specifically tailored for graphons, significantly enhancing their expressive power and providing new universal approximation properties for graphon signals. The work advances the theory in graph neural networks by rigorously defining IWNs and aligning them with theoretical results through the introduction of signal-weighted homomorphism densities.
- The paper presents a thorough comparison of IWNs with existing graph neural networks, including Message Passing Neural Networks (MPNNs) and other IGNs, addressing size transferability and continuity aspects. It establishes that IWNs maintain at least the same expressive power as the k-Weisfeiler-Leman (k-WL) test, a known benchmark in GNN expressivity.
- The authors have provided extensive mathematical proofs.

**Weaknesses:**

The absence of empirical experiments and practical application scenarios is the key weakness of this work. Without empirical validation, it is difficult to gauge how IWNs perform on real-world datasets, whether the theoretical advantages translate into meaningful results, and how they compare to established GNN models in practice. IWNs could be applied in scenarios where we need to capture dense graph structures or analyze graphs of varying sizes, which could benefit from their extended expressivity and approximation capabilities.
- It is unclear whether the mathematical assumptions hold across diverse real-world datasets. This could limit the applicability of IWNs to more constrained datasets where these assumptions are valid.
- IWNs are discontinuous with respect to the cut distance. This discontinuity implies that minor structural changes in the discrete graph could lead to significant output changes in the IWN, unlike Message Passing Neural Networks (MPNNs).
- Adapting IWNs to discrete graphs might involve computational overhead, as the model is designed for continuous graphon spaces rather than finite graphs.

**Questions:**

Please see the weaknesses part.

---

> ### Author Response · Authors · 2024-11-21
> **Official Comment by Authors**
>
> We thank the reviewer for their comments and positive feedback on our paper.
>
> > absence of empirical experiments and practical application scenarios
>
> See general comments.
>
> > unclear whether the mathematical assumptions hold across diverse real-world datasets
>
> Indeed, graphons are *dense* graph limits, while real-world datasets often exhibit sparser structures. Nevertheless, still any finite graph can be represented as a graphon, and our analyses do not impose any additional regularity assumptions on the graphon such as their continuity, which is *much* more restrictive.
>
> > computational overhead
>
> IWNs can be applied to any finite graph and have the same asymptotic complexity as IGNs or higher-order GNNs for any given tensor order. Also, our construction is smaller than the full IGN basis (see § F).

---

### Author Response · Authors · 2024-11-21
**General Comments by Authors**

We thank all reviewers for their thoughtful and constructive feedback and for **unanimously supporting our work**! We have uploaded a major revision of the script, addressing the reviewers’ feedback. See the highlighted parts for specific updates. Most importantly, we have made the following changes:

1. Throughout the paper, we tried to make the motivation and practical consequences of our work clearer, specifically by **giving a better intuition for the cut distance discontinuity**, as suggested by reviewers fxhj and 1FQG. The cut distance discontinuity is *not* visible when considering finite graphs up to a certain size (as all norms are equivalent there), but only when limits to a graphon are taken. Most prominently, the cut distance discontinuity can be observed when applying an IWN to simple graphs sampled from a graphon-signal. While the simple graphs converge to the underlying graphon-signal in the cut distance (Eq. 8), the function values of the IWN do not converge. This is similar to the result of [1] on non-convergence of IWNs under their “edge probability discrete model”. We have rewritten the section on continuity (now § 5.2) and have dedicated a paragraph (L454-459) to an intuitive explanation.
2. An intriguing point brought up by reviewer p53n is that it is “important to justify [that cut distance discontinuity is not] a **product of a definitional choice that could be reworked** and rather a facet of the true underlying problem”. In the updated version, we have made clearer that the cut distance discontinuity is **inherently tied to the definition of the $k$-WL test**: The $k$-WL test considers multigraph homomorphism densities, which are cut distance discontinuous, and therefore, *any* ML model defined on graphon-signals which is $k$-WL expressive (e.g., also the higher-order models from [3]) would exhibit this discontinuity. This comes from a fundamental difference in the way that 1-WL and $k$-WL consider weighted edges, which are treated as actual weights by 1-WL and only act through the shift operator of the graph(on), while $k$-WL (and IWNs) see them as edge features, capturing their entire *distribution*. We have clarified this in § 5.2 (L460-467). For this discussion, we also extend the $k$-WL test, for which we previously just used the definition for *graphons* from [2], to graphon-signals (see Theorem 3.5 and Appendix C.4).
3. We **significantly refine the discussion of the transferability of IWNs**: As the cut distance discontinuity implies that IWNs do not converge to their graphon limits, this means that the standard approach to prove transferability by invoking the triangle inequality does *not* apply to IWNs. However, it turns out that this **discontinuity can be “fixed”**, in the sense that any IWN can be approximated arbitrarily well by a cut distance continuous function, for which transferability can be obtained. As such, IWNs are just as transferable as MPNNs in the worst case. See Theorem 5.6 for the formal statement, and Appendix E.6 for its proof. We believe that **this observation makes our contribution even stronger**: For practical purposes typically the actual transferability is of greater interest than the convergence to the graphon limit, which is usually just an intermediate step.
4. While we primarily view the work as a theoretical contribution, we agree with the reviewers that **experiments** can improve understanding of the concepts. We added a toy experiment in Figures 1-3, showing the continuity and transferability of IWNs vs. MPNNs, and provide details in Appendix G.
5. To make space for the experiments and discussion of transferability, we had to move some of the proof sketches (Theorem 4.2 and Theorem 5.2) to the appendix.

See below for individual responses to the reviewers.

**References**

[1] Chen Cai and Yusu Wang. Convergence of Invariant Graph Networks. In Proceedings of the 39th International Conference on Machine Learning, pp. 2457–2484. PMLR, June 2022.

[2] Jan Böker. Weisfeiler-Leman Indistinguishability of Graphons. The Electronic Journal of Combinatorics, 30(4):P4.35, December 2023. ISSN 1077-8926. doi: 10.37236/10973.

[3] Christopher Morris, Martin Ritzert, Matthias Fey, William L. Hamilton, Jan Eric Lenssen, Gaurav Rattan, and Martin Grohe. Weisfeiler and leman go neural: Higher-order graph neural networks. In Proceedings of the AAAI conference on artificial intelligence, volume 33, pp. 4602–4609, 2019.

---

### Meta-Review · Area_Chair_eUDb · 2024-12-20

**Metareview:**

The authors propose Invariant Graphon Networks (IWNs) to generalized Invariant Graph Networks (IGNs) to graphons. The authors prove that IWNs of order (k+1) is at least as powerful as k-dimensional Weisfeiler-Leman (WL) test for graphons. The derived expressivity extend results in Cai and Wang (2022). Additionally, the authors show that IGNs are discontinuous with respect to cut distance, in contrast to the continuity results of MPNNs on graphons (Levie 2023). The Reviewers agree that it is a good submission for ICLR'2025. However, the lack of empirical supporting evidence remains as a concern for the Reviewers. We urge the Authors to incorporate the Reviewers' comments and discussions in the rebuttal to the updated version.

**Additional Comments On Reviewer Discussion:**

The authors clarify the notion of signal-weighted homomorphism densities, and the technical contribution to extend results in Böker et al (2023) and Levie 2023 in the rebuttal.

---

### Decision · Program_Chairs · 2025-01-22

Accept (Spotlight)